# The Tell-Tale Norm: $\ell_2$ Magnitude as a
# Signal for Reasoning Dynamics in Large Language Models

**Jinyang Zhang** [1 2 3 *]  **Hongxin Ding** [1 2 3 *]  **Yue Fang** [1 2 3 *]  **Weibin Liao** [1 2 3 *]
**Muyang Ye** [4]  **Junfeng Zhao** [1 3 ✉]  **Yasha Wang** [2 3 5 ✉]

## Abstract

Recent work has sought to understand Large Language Models (LLMs) reasoning, yet a principled, model-intrinsic signal that captures its *layer-wise reasoning dynamics* remains underexplored. We bridge this gap by demonstrating that **the $\ell_2$ norm of hidden states serves as an endogenous signal of the model's reasoning intensity.** Using Sparse Autoencoders (SAEs) as a diagnostic probe, we observe that LLMs' internal reasoning is marked by a sharp increase in reasoning feature activations concentrated in late layers. Motivated by this pattern, we establish a formal link between reasoning intensity and the model's latent geometry and theoretically prove that the $\ell_2$ norm of hidden states bounds the activation strength of SAE reasoning features. Empirical correlation analysis and causal interventions further prove $\ell_2$ norm as a faithful indicator, where heightened norms consistently correspond to critical reasoning steps. We then introduce three test-time scaling techniques guided by $\ell_2$ norms: (i) Adaptive Layer-wise Reasoning Recursion, (ii) Endogenous Reasoning State Steering, and (iii) $\ell_2$-guided Response Selection, which requires no additional training or data and is compatible with advanced inference engines. Experiments across model architectures and benchmarks show that $\ell_2$-norm-based techniques significantly im-

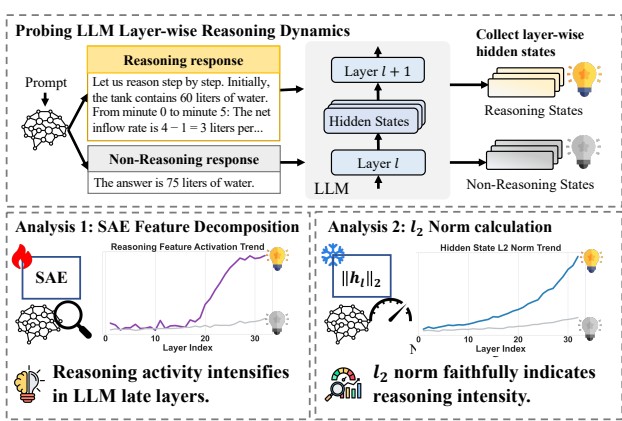

*Figure 1.* Analysis on LLM layer-wise reasoning dynamics shows that reasoning activity intensifies in the later layers, which is reflected by SAE feature activations and hidden state $\ell_2$ norms.

prove reasoning performance, offering a principled yet simple lens to perceive and control LLM latent reasoning dynamics. Our codes are available at https://github.com/zjy1298/The-Tell-Tale-Norm.

## 1. Introduction

Reasoning abilities in recent Large Language Models (LLMs), such as DeepSeek-R1 (Guo et al., 2025), GPT-4 o1 (Achiam et al., 2023) and Qwen-3 (Yang et al., 2025) have propelled model performance to unprecedented heights across complex reasoning tasks. These models exihibit explicit *reasoning behaviors* at the output level, generating extended Chain-of-Thought (CoT) sequences (Wei et al., 2022; Feng et al., 2023; Liao et al., 2025). Despite these successes, the internal dynamics and mechanisms underlying LLM reasoning remain largely opaque. In particular, how reasoning emerges and evolves within the LLM's hidden space across layers and during generation has received little systematic investigation. Understanding these properties is crucial for both interpreting model behavior and designing interventions to enhance reasoning performance.

---

*Equal contribution  [1]School of Computer Science, Peking University  [2]National Engineering Research Center of Software Engineering, Peking University  [3]Key Laboratory of High Confidence Software Technologies, Ministry of Education  [4]Zhejiang University, Hangzhou, China  [5]Peking University Information Technology Institute (Tianjin Binhai). Correspondence to: Jinyang Zhang <jinyangzhang25@stu.pku.edu.cn>, Junfeng Zhao <zhaojf@pku.edu.cn>, Yasha Wang <wangyasha@pku.edu.cn>.

*Proceedings of the 43rd International Conference on Machine Learning*, Seoul, South Korea. PMLR 306, 2026. Copyright 2026 by the author(s).

Prior work has explored LLM reasoning mechanisms from two perspectives. One line of research focuses on the output space, identifying "reasoning tokens" through heuristics (Galichin et al., 2025; Tang et al., 2025; Zhang et al., 2025a) or metrics like entropy (Wang et al., 2025) and mutual-information with correctness (Qian et al., 2025). While informative, these methods are inherently retrospective: they can only identify and intervene reasoning after decoding, failing to reveal the intermediate computations or dynamics. Another line leverages mechanistic interpretability tools, such as Sparse Autoencoders (SAEs) (Bricken et al., 2023; Cunningham et al., 2023; Shu et al., 2025). By reconstructing activations under sparsity constraints, SAEs identify features causally linked to reasoning within LLM hidden space across layers (Zhang et al., 2025b; Fang et al., 2026c; Li et al., 2025b; Chen et al., 2025b; Venhoff et al., 2025). However, SAE features are not endogenous properties of the model, but merely proxies whose identification requires nontrivial, model- and data-specific training, which limits their generality and practicality. Moreover, most SAE research adopts a localized view, focusing on interpreting and manipulating the extracted features themselves, overlooking the significance of *how such features are distributed across the model's depth with varying sparsity and activation strength, which provides insights into the model's* **layer-wise reasoning dynamics**.

To mitigate this gap, we delve into the model's layer-wise reasoning behavior and take a step toward answering the two fundamental research questions:

- *RQ1: Do LLMs exhibit distinct layer-wise dynamics when performing reasoning?*
- *RQ2: Can such reasoning activity be characterized by perceivable, model-intrinsic properties, independent of external probes?*

To address **RQ1**, we leverage established SAE methodologies to identify reasoning features within the LLM's hidden space. Our investigation reveals a clear **layer-wise reasoning trajectory**: when the LLM is engaged in reasoning, SAE reasoning feature activations in late layers exhibit significantly high magnitudes. This observation suggests that LLMs enter a distinct reasoning state at intermediate depths. However, while SAEs provide a lens into these dynamics, they are not endogenous to the model; their training is computationally expensive and data-specific, limiting their utility as a general-purpose diagnostic tool.

This limitation leads us to **RQ2**: we seek a training-free and model-intrinsic indicator of reasoning. We theoretically demonstrate that under the conditions of faithful SAE reconstruction and orthogonal reasoning and semantic features, the $\ell_2$ norm of hidden states provides two-sided bounds on the activation strength of SAE reasoning features. Intuitively, reasoning induces feature shifts in hidden space that significantly inflate activation magnitudes, leading to higher $\ell_2$ norms. Our empirical correlation analysis confirms that layer-wise $\ell_2$ norm uniquely aligns with SAE-identified reasoning features and model output entropy, outperforming other statistical metrics. Causal interventions that suppress high-norm tokens in reasoning-heavy layers lead to a more drastic performance drop compared to random suppression, further validating the conclusion. These findings establish the *layer-wise $\ell_2$ norm as a robust, training-free indicator for thinking intensity and reasoning activeness in LLMs.*

Building on these insights (Figure 1), we explore a new direction for test-time reasoning control. We propose a suite of $\ell_2$-norm-based techniques that adaptively amplify reasoning activity during intermediate computations, without any additional data or training. Specifically, we introduce: (i) **Adaptive Layer-wise Reasoning Recursion**, allocating additional compute to critical reasoning layers, (ii) **Endogenous Reasoning State Steering**, guiding hidden states toward high-intensity subspaces to amplify reasoning, and (iii) $\ell_2$-**guided Response Selection**, where a prediction with the highest reasoning intensity is selected among candidates. These techniques yield *+4.51%* average gain across benchmarks and *+9.13%* on challenging reasoning tasks, highlighting their effectiveness.

**Our contributions are as follows:**

- **Insightfully**, we reveal LLMs' layer-wise reasoning dynamics through SAEs, and identify the intrinsic layer $\ell_2$ norm as a faithful indicator that captures reasoning activeness and intensity.
- **Theoretically**, we prove that the $\ell_2$ norm bounds SAE reasoning feature activation, supported by extensive correlation and causal analysis.
- **Practically**, we propose three training-free, data-free, adaptive techniques for amplifying reasoning and boosting model performance at test time.
- **Experimentally**, we demonstrate consistent improvements across multiple mathematical and general reasoning benchmarks across diverse LLM architectures.

## 2. Probing LLM Layer-wise Reasoning Dynamics via Sparse Autoencoders

In this section, to answer **RQ1**, we analyze internal representations in LLMs using SAEs, to understand how reasoning unfolds within the model's hidden space.

## 2.1. Preliminaries: Sparse Autoencoders as Interpretability Probes

The primary challenge in interpreting LLM hidden states is neuron polysemanticity and superposition, where meaningful features are embedded in overlapping subspaces. SAEs address this by decomposing hidden activations $h \in \mathbb{R}^d$ into a sparse, overcomplete set of interpretable features.

An SAE typically consists of an encoder and a decoder. The encoder maps $h$ to high-dimensional latent space $f(h) = \text{ReLU}(W_{enc}(h - b_{dec}) + b_{enc})$, where $f \in \mathbb{R}^M$ ($M \gg d$). The decoder reconstructs the original state: $\hat{h} = W_{dec}f(h) + b_{dec}$. Columns of the decoder matrix are denoted by $\{\mathbf{d}_i\}_{i=1}^M$, form a set of basis vectors and are commonly interpreted as *SAE features*, where each $\mathbf{d}_i$ corresponds to a distinct direction in the hidden space. To ensure the features are interpretable, the SAE is trained by minimizing a reconstruction loss alongside an $l_1$ penalty:

$$\mathcal{L} = \|h - \hat{h}\|_2^2 + \lambda \|f(h)\|_1$$

By enforcing sparsity, SAEs disentangle the polysemantic states into monosemantic latent features, making it possible to isolate features that correspond exclusively to reasoning.

## 2.2. Layer-wise Reasoning Dynamics Revealed by SAE Reasoning Feature Activations

**Reasoning Feature Identification.** For a given LLM, for each transformer layer $l$ in the model, we train an independent SAE on hidden states of that layer collected during inference on the MMLU-pro (Wang et al., 2024) dataset. To identify reasoning-related features, we generate two paired responses: (i) a *thinking* response that contains an explicit chain-of-thought, and (ii) a *non-thinking* response that tries to directly answer the question. Given a feature $\mathbf{d}_i$ in the layer $l$ SAE, let $z_{l,i,t}$ denote its activation on token $t$. We compute the differential activation of feature $\mathbf{d}_i$ at layer $l$:

$$\Delta z_{l,i} = \frac{1}{|\mathcal{T}_{\text{think}}|} \sum_{t \in \mathcal{T}_{\text{think}}} z_{l,i,t} - \frac{1}{|\mathcal{T}_{\text{non}}|} \sum_{t \in \mathcal{T}_{\text{non}}} z_{l,i,t}. \quad (1)$$

where $\mathcal{T}_{\text{think}}$ and $\mathcal{T}_{\text{non}}$ denote token sets in the thinking and non-thinking responses, respectively. This contrast isolates reasoning activity by holding the semantics fixed: features that activate strongly only in the thinking response therefore reflect reasoning-specific internal states rather than answer content. We therefore identified the **Top-$k$** features with the highest $\Delta z_{l,i}$ as reasoning features in each layer. Qualitative inspection further validates that these features capture genuine reasoning behavior: tokens eliciting high activations for these features (visualized in Figure 15b) predominantly correspond to reasoning-critical words (e.g., logical transitions and intermediate conclusions).

**Layer-wise Reasoning Dynamics** We visualize the normalized mean activation magnitudes of top-5 reasoning features across LLM layers during thinking mode generation. As shown in the top row of Figure 2, across Qwen-3 variants (visualizations and discussions for more models are in Appendix C.2 and F.), we observe a consistent pattern: **SAE reasoning features exhibit substantially higher activation magnitudes in late layers than in early-to-middle layers**. This layer-wise heterogeneity suggests that reasoning is not uniformly distributed throughout the network. Instead, early layers primarily process lexical and semantic information, while late layers increasingly concentrate reasoning-related computations, entering a distinct high-intensity thinking state. Notably, this transition precedes explicit token generation, enabling reasoning to be amplified or steered directly in the hidden space for efficient test-time scaling, without relying on expanding long CoT at the output level.

While SAEs effectively expose the internal reasoning dynamics and offer an opportunity to efficiently control model reasoning, they remain external proxies. The requirement to train SAEs for every LLM is computationally prohibitive and lacks generality and practicality. Naturally, this raises a fundamental question: *if reasoning truly induces distinct internal states within the model, does it also manifest in an intrinsic property of the model's latent geometry, which can be perceived and controlled without relying on external probes?* In the following section, we move beyond external probes to explore how this reasoning activity is intrinsically reflected in the $\ell_2$ norm of the hidden states.

## 3. $\ell_2$-Norm as an Intrinsic Signal for Reasoning Activity

In this section, to answer $RQ2$, we go beyond external probes and establish a formal link between reasoning and the model's endogenous $\ell_2$ norm of hidden states.

### 3.1. Theoretical Foundations: Linking SAE Activations to Hidden-State $\ell_2$

We prove that when an LLM enters reasoning states, resulting shifts in its feature-space representation manifest as a measurable expansion in the hidden state's $\ell_2$ magnitude.

#### 3.1.1. FOUNDATIONAL ASSUMPTIONS

To formalize the relationship between hidden states $\mathbf{h} \in \mathbb{R}^d$ and SAE features $\mathbf{z} \in \mathbb{R}^m$, we rely on the following priors:

- **Assumption 1 (Approximate Orthonormality)**: The decoder basis $\{\mathbf{d}_i\}_{i=1}^m$ satisfies $\langle \mathbf{d}_i, \mathbf{d}_j \rangle = \delta_{ij} + \mathcal{O}(\epsilon_{\text{orth}})$.
- **Assumption 2 (Reconstruction Quality)**: Let $\hat{h}$ be the SAE reconstructed hidden states, we assume faithful reconstruction where $\|\mathbf{h} - \hat{\mathbf{h}}\|_2 \leq \epsilon_{\text{recon}}$.
- **Assumption 3 (Semantic Orthogonality)**: We de-

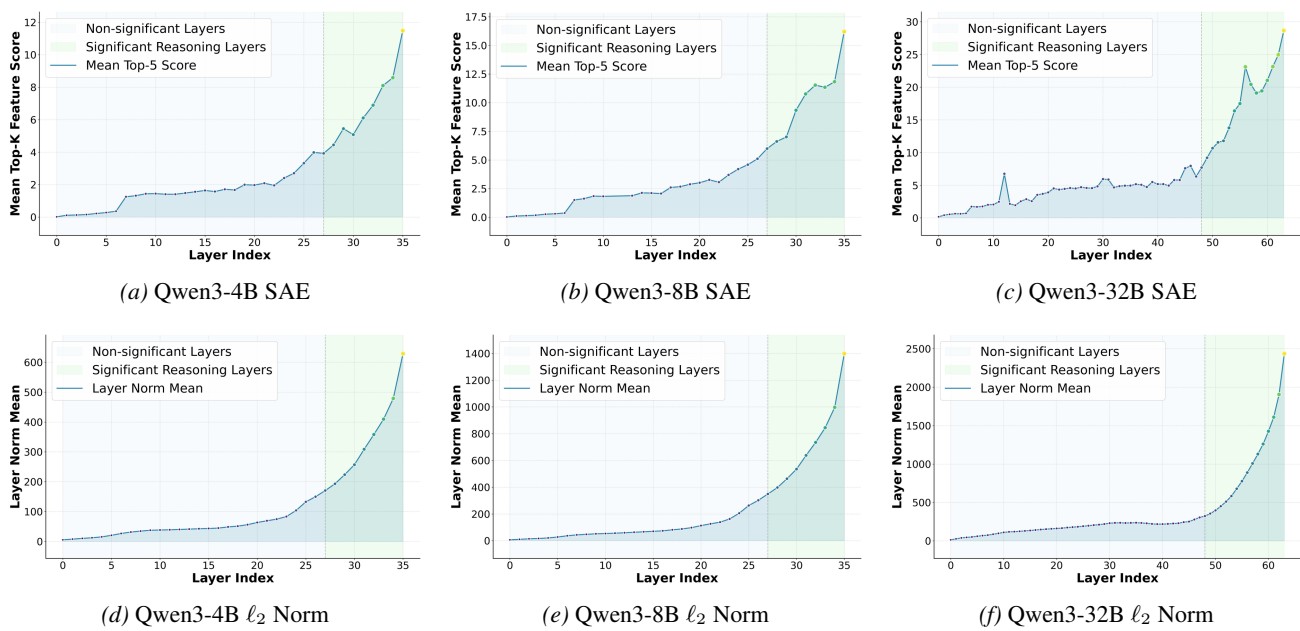

*Figure 2.* Layer-wise activations of top-5 SAE reasoning features (top row) and hidden state $\ell_2$ norms (bottom row) across Qwen-3 model family. Both SAE activations and $\ell_2$ norms rise and peak in the last quarter of layers with similar trends.

compose the average hidden states into $\bar{\mathbf{h}}_{\text{think}} = \mathbf{h}_{\text{base}} + \mathbf{h}_{\text{reasoning}}$ and $\bar{\mathbf{h}}_{\text{non}} = \mathbf{h}_{\text{base}} + \boldsymbol{\epsilon}_{\text{noise}}$, where $\epsilon_{\text{noise}} \ll \|\mathbf{h}_{\text{reasoning}}\|_2$. We assume reasoning dynamics are approximately orthogonal to base semantics: $|\langle \mathbf{h}_{\text{base}}, \mathbf{h}_{\text{reasoning}} \rangle| \le \beta \|\mathbf{h}_{\text{base}}\|_2 \|\mathbf{h}_{\text{reasoning}}\|_2$ for $\beta \ll 1$.

- **Assumption 4 (Feature Disentanglement)**: The SAE partitions the latent space into disjoint sets $\mathcal{I}_{\text{base}}$ and $\mathcal{I}_{\text{reasoning}}$, where $\mathcal{I}_{\text{reasoning}}$ captures features predominantly induced by reasoning.

### 3.1.2. HIDDEN-STATE $\ell_2$ NORM AS LATENT ENERGY

We first establish that the squared $\ell_2$ norm of $\mathbf{h}$ serves as a faithful aggregator of its active latent feature energies.

**Lemma 1 (**

$$\ell_2$$

**Energy Decomposition).** Under Assumption 1 and 2, the squared $\ell_2$ norm of $\mathbf{h}$ satisfies:

$$\left| \|\mathbf{h}\|_2^2 - \sum_{i=1}^{m} z_i^2 \right| \le \mathcal{R}(\epsilon_{\text{recon}}, \epsilon_{\text{orth}}), \quad (2)$$

where $\mathcal{R} = 2\epsilon_{\text{recon}} \|\mathbf{h}\|_2 + 3\epsilon_{\text{recon}}^2 + \mathcal{O}(\epsilon_{\text{orth}} \|\mathbf{z}\|_2^2)$.

*Proof Sketch.* We expand $\|\hat{\mathbf{h}}\|_2^2 = \langle \sum z_i \mathbf{d}_i, \sum z_j \mathbf{d}_j \rangle$. Under Assumption 1, this yields $\sum z_i^2$ plus a term $\mathcal{O}(\epsilon_{\text{orth}} \|\mathbf{z}\|_2^2)$. Applying Cauchy-Schwarz to the reconstruction residual $\mathbf{r} = \mathbf{h} - \hat{\mathbf{h}}$ completes the bound. $\square$

*Intuition.* This result implies that changes in the hidden-state norm directly reflect changes in total latent feature energy, up to controlled approximation error.

### 3.1.3. MAIN THEOREM: $\ell_2$ NORM AS AN INTENSITY SIGNAL

Our main theorem formally shows that the differential activation of SAE reasoning features, is directly reflected in the observable shift in the hidden state's magnitude.

**Theorem 1 (Reasoning Intensity Bounds).** Let $\Delta z_i = z_i^{\text{think}} - z_i^{\text{non}}$. Under Assumptions 1–4 and Lemma 1, define

$$\Delta_{\text{obs}} := \|\bar{\mathbf{h}}_{\text{think}}\|_2^2 - \|\bar{\mathbf{h}}_{\text{non}}\|_2^2.$$

Then there exist constants $c_1, c_2 > 0$ depending only on $\beta$, $\epsilon_{\text{orth}}$, and SAE sparsity such that

$$c_1 \sqrt{\Delta_{\text{obs}} - \epsilon_{\text{total}}} \le \sum_{i \in \mathcal{I}_{\text{reasoning}}} \Delta z_i \le c_2 \sqrt{\Delta_{\text{obs}} + \epsilon_{\text{total}}}. \quad (3)$$

In particular, when $\epsilon_{\text{total}} \ll \Delta_{\text{obs}}$ and $\beta \ll 1$,

$$\sum_{i \in \mathcal{I}_{\text{reasoning}}} \Delta z_i = \Theta\left( \sqrt{\|\bar{\mathbf{h}}_{\text{think}}\|_2^2 - \|\bar{\mathbf{h}}_{\text{non}}\|_2^2} \right). \quad (4)$$

**Intuition.** Theorem 1 formalizes a direct quantitative link between the observable $\ell_2$ norm expansion in hidden states and the cumulative activation of reasoning-specific features. Intuitively, when the model enters a reasoning mode, additional latent components in $\mathcal{I}_{\text{reasoning}}$ become activated on top of semantic representations. Due to their approximate orthogonality to non-reasoning feature, these activations add constructive "energy" to the hidden state, causing a noticeable increase in $|\mathbf{h}|_2^2$. This justifies using the $\ell_2$

norm as a **training-free signal**: we can monitor the "expansion" of the model's latent space to quantify reasoning intensity without deploying expensive SAE probes at every layer. Detailed assumptions, extended justifications, and the complete formal proofs for all lemmas and theorems are provided in Appendix B.

What's more, a complementary theoretical perspective, which links the $\ell_2$-norm of hidden states to the optimization pressure exerted by the cross-entropy loss on hard and reasoning-critical tokens, is presented in Appendix B.3.

### 3.2. Empirical Correlation Analysis

We then conduct a comprehensive multi-granular empirical analysis and further validate the link between $\ell_2$ norm and the model's latent reasoning intensity.

**Validity of theoretical assumptions.** We empirically validate the reliability of the main theoretical assumptions underlying our analysis, including decoder near-orthogonality, reconstruction fidelity, semantic–reasoning separation, and feature-level disentanglement. Across models and layers, the results consistently support these assumptions to a sufficient approximation for our geometric interpretation. Detailed evidence and quantitative results are provided in Appendix C.1.

**Layer-wise Trend Alignment.** We first visualize the average $\ell_2$ norm of hidden states for thinking response tokens across the model's depth. As shown in the bottom row of Figure 2, a pronounced two-stage pattern that is consistent across all models is observed. The activation magnitudes remain relatively low and stable in early-to-middle layers while a sharp increase occurs in approximately the final 25% of layers. A vertical comparison with SAE trends within each model reveals that the $\ell_2$ norm closely mirrors the layer-wise behavior of SAE reasoning feature activations, suggesting that **the late-stage separation of reasoning features is coupled with a global expansion of the hidden state energy**. More detailed analysis and visualizations for other models are provided in Appendix C.2

**Quantitative Correlation with SAE Activations and Final Output Entropy.** To quantitatively evaluate the alignment between $\ell_2$ norm and reasoning-related behavior, we compute Spearman correlations (Spearman, 1961) between several intrinsic, layer-wise metrics and two reasoning signals: (i) SAE reasoning feature activations, and (ii) final output entropy, which is associated with reasoning processes, as high-entropy output tokens typically correspond to critical reasoning steps where multiple plausible continuations compete (Wang et al., 2025).

Besides $\ell_2$, we consider additional layer-wise intrinsic metrics: layer hidden entropy, layer output entropy, and atten-

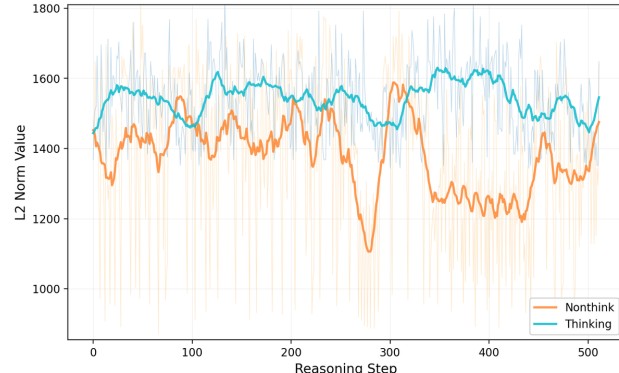

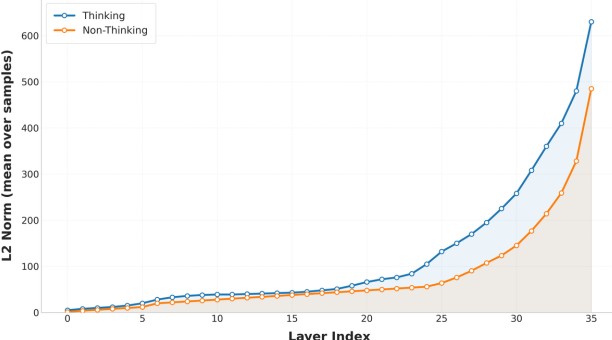

*Figure 3.* Hidden-state $\ell_2$ norm comparison between thinking and non-thinking generations. **Top:** Token-wise evolution of hidden-state $\ell_2$ norms at Layer 34 of Qwen3-8B, during thinking vs. non-thinking generations. $\ell_2$ norm remains consistently higher for the thinking mode. **Bottom:** Layer-wise evolution of hidden-state $\ell_2$ norms of Qwen3-4B, during thinking vs. non-thinking generations. $\ell_2$ norm remains consistently higher for the thinking mode.

tion entropy, as calculated in Appendix C.4. As summarized in Table 1, $\ell_2$ norm consistently exhibits strong and positive correlations with both SAE activations and final output entropy. In contrast, alternative metrics show substantially weaker and unstable correlations. These results demonstrate that $\ell_2$ norm uniquely aligns with established reasoning signals and more faithfully tracks internal reasoning dynamics than other statistical properties.

**Thinking vs. Non-thinking Contrast.** We further compare hidden-state $\ell_2$ norms between thinking and non-thinking generations. As shown in Figure 3, the thinking mode consistently exhibits larger norms in both token-wise and layer-wise views, with the gap becoming more pronounced in later layers/tokens. This suggests that reasoning states are associated with a clearer high-energy signature.

**Token-wise Semantic Evidence.** We examine tokens with high $\ell_2$ norms and compare them with those identified by high SAE activations. As visualized in Figure 15, we observe a similar and overlapped depth-dependent shift in focus: tokens related to control-flow markers that structure multi-step reasoning (e.g., "if", "but") exhibit higher

*Table 1.* Spearman correlation between layer-wise intrinsic statistics and (i) SAE reasoning feature activations (across all layers), and (ii) final output entropy (across the last quarter of layers as these layers are closer to the output).

| Metric | Qwen3-1.7B | | Qwen3-8B | | Qwen3-14B | | LLaMA3-8B | |
|---|---|---|---|---|---|---|---|---|
| | **SAE** | **Final Entropy** | **SAE** | **Final Entropy** | **SAE** | **Final Entropy** | **SAE** | **Final Entropy** |
| $\ell_2$ Norm | **86.47** | **63.52** | **85.15** | **63.25** | **87.62** | **66.65** | **84.11** | **52.10** |
| Layer Hidden Entropy | -41.58 | -8.62 | -19.94 | -13.67 | -47.07 | -15.76 | -69.97 | -31.59 |
| Layer Output Entropy | -61.89 | -22.97 | -65.35 | -31.82 | -69.97 | -27.54 | -72.15 | -34.50 |
| Attention Entropy | -51.42 | 27.59 | -54.37 | 32.39 | -52.76 | 23.46 | 1.80 | 36.65 |

saliency in late-layers (e.g., Layer 30) than those in middle layers (e.g., Layer 20). This transition suggests that the late-layer amplification of the $\ell_2$ norm specifically signals the emergence of explicit reasoning control. While tokens identified are not identical, we note that this is because $\ell_2$ norm provides a coarse "energy-like" indicator of where the model expends representational capacity. On the other hand, SAE features capture more selective, directional subspaces aligned with reasoning. Nevertheless, the partial overlap and systematic co-variation across depth suggest that high-$\ell_2$ tokens mark locations where the model allocates substantial computation toward reasoning control. A detailed comparative analysis of content-vs-control dynamics is provided in Appendix C.3. Case studies of critical tokens marked by $\ell_2$ are in Appendix M.

### 3.3. Causal Intervention Analysis

To rigorously verify whether the observed $\ell_2$ norm peaks are functionally necessary for reasoning, we perform a controlled intervention study that directly manipulates hidden states during inference. We focus on *significant reasoning layers*, defined as the final 25% of layers ($l \geq 0.75L$), where both SAE activations and $\ell_2$ norms exhibit sharp increases. Given a hidden state $h_l$ at layer $l$, if its $\ell_2$ norm exceeds an *adaptive layer-wise threshold* $\tau_\ell$ (Algorithm 1), we suppress the state by scaling it as: $h'_l = \alpha \cdot h_l$, with $\alpha = 0.1$, as detailed in Algorithm 2. This intervention selectively weakens high-magnitude hidden states, which we hypothesize correspond to moments of intensive reasoning. As a baseline, we implement *Random Suppression*, where the same proportion of tokens is randomly selected and suppressed by the same factor.

Figure 4 visualizes the intervention results on Qwen3-14B. Across all suppression levels, targeting high-$\ell_2$-norm states consistently causes severe performance degradation, even at low intervention ratios, while random suppression leads to a substantially milder decline. These results provide strong causal evidence that the model's reasoning activity is concentrated in the high-magnitude subspace, and that $\ell_2$ norm is functionally tied to the model's reasoning process. Complete numerical results, visualizations for other models, more analysis, and case studies are in Appendix D.

## 4. Application: $\ell_2$ Norm Guided Test-Time Reasoning Control

Drawing insights from our previous analysis, which reveals that the $\ell_2$ serves as a reliable signal for model reasoning intensity, we explore three complementary directions that leverage $\ell_2$ norm dynamics for test-time reasoning control:

- *Computation Scaling:* Identifying critical layers where standard processing may be insufficient and dynamically assigning more computation to deepen understanding.
- *State Intervention:* Guiding latent representations toward high-energy subspaces to amplify reasoning.
- *Result Evaluation:* Leveraging internal signals to rank candidate outputs without external labels.

An overview of these techniques is illustrated in Figure 5. Pseudo codes are provided in Appendix L.

### 4.1. $\ell_2$ Norm Peak Detection

Efficient intervention requires pinpointing *when* and *where* reasoning is most intensive. While the $\ell_2$-norm reflects reasoning intensity, identifying these states during test-time decoding is non-trivial. Naively applying a fixed global threshold is infeasible for two reasons. First, determining which states belong to the "top" percentile is non-causal, as such global statistics are only available after the entire sequence is generated. Second, as demonstrated in Section 3, the average $\ell_2$-norm varies significantly between "thinking" and "non-thinking" regimes. Since benchmarks with different difficulty require varying levels of thinking intensity, a threshold inferred from one dataset cannot generalize to others. Such cross-benchmark tuning would also violate the data-free objective. To overcome these challenges, we propose an adaptive $\ell_2$ **norm peak detection** mechanism. By detecting outlier magnitudes relative to the local context within the current decoding sequence, we capture high-norm pivotal reasoning moments.

Let $\{m_t\}_{t=1}^T$ be the sequence of $\ell_2$ norm scores of target hidden state residual stream vectors at decoding step $t$. Following the peak definition (Tukey, 1977), we define the inter-quartile range (IQR) as $\mathrm{IQR} = Q_3 - Q_1$, where $Q_1$ and $Q_3$ are the 25th and 75th percentiles of the sequence

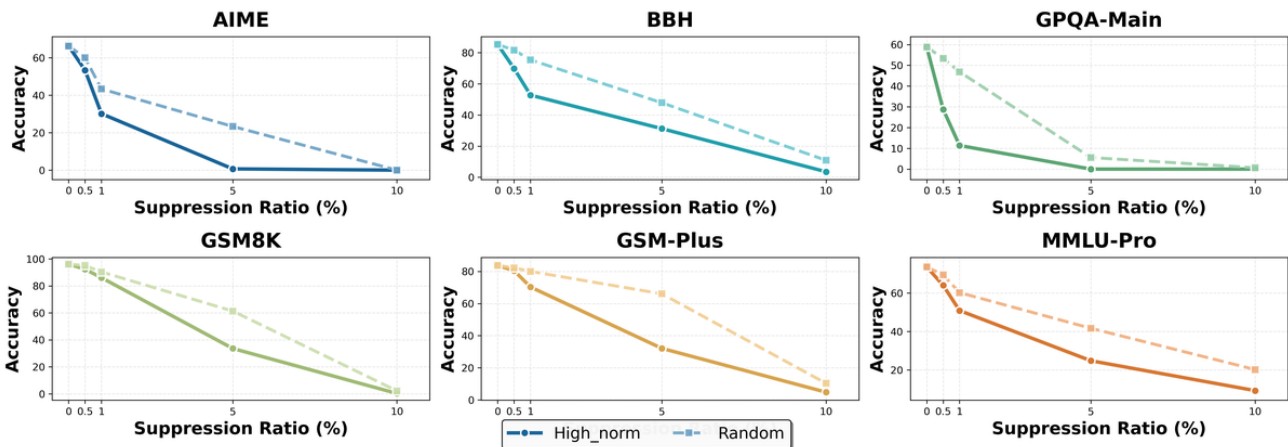

*Figure 4.* Causal intervention results on Qwen3-14B across reasoning benchmarks under different suppression ratios. We compare suppressing high-$\ell_2$-norm hidden states against Random Suppression.

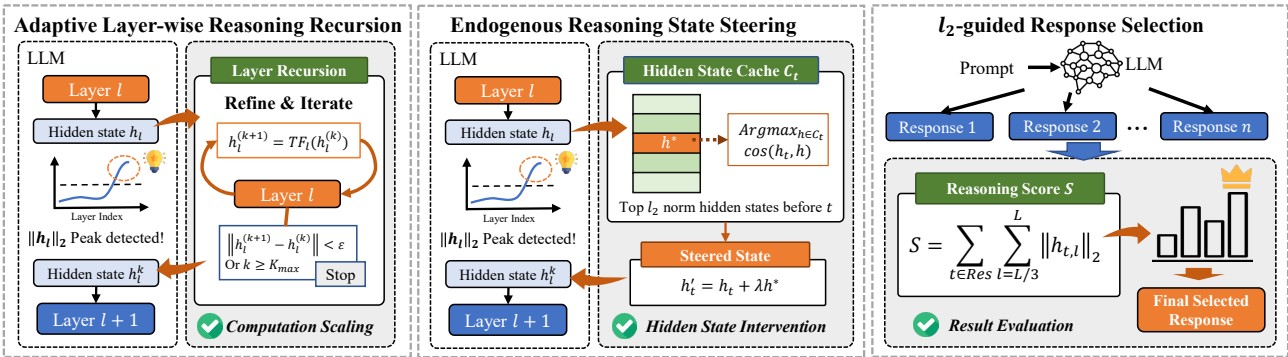

*Figure 5.* Practical applications based on $\ell_2$ norm dynamics.

$\{m_t\}$. We identify the set of $\ell_2$ norm peaks $\mathcal{P}_{\ell_2}$ as:

$$\mathcal{P}_{\ell_2} = \{\, t \mid m_t \geq Q_3 + \tau \cdot \text{IQR}(m) \,\}, \qquad (5)$$

where $\tau$ (typically set to 1.5 (Tukey, 1977)) is a scale factor controlling the strictness of the peak threshold. Thisclas approach is fully adaptive and prior-free: it automatically calibrates to the difficulty of the prompt and the model's internal state, ensuring that interventions are triggered at moments of relatively high-intensity thinking.

### 4.2. Adaptive Layer-wise Reasoning Recursion

Motivated by the goal of *Computation Scaling*, we propose **Adaptive Layer-wise Reasoning Recursion (ALRR)** to deepen reasoning at layers where $\ell_2$ peaks occur. These positions often encode complex, pivotal reasoning elements, such as intermediate conclusions or variable bindings. When a peak is detected at layer $l$, rather than directly passing the state to layer $l + 1$ following standard transformer information propagating, we perform an in-place refinement by iteratively re-injecting and reprocessing the state in the same layer:

$$h_l^{(0)} = h_l, \quad h_l^{(k+1)} = \text{TF}_l\big(h_l^{(k)}\big) \qquad (6)$$

The recursion terminates when $\|h_l^{(k+1)} - h_l^{(k)}\|_2 < \epsilon$ or $k$ reaches a limit $K_{max}$. This procedure allocates additional computation precisely where reasoning activity is most concentrated, effectively increasing the depth of the reasoning chain, enabling the model to "ponder" and encouraging local convergence in representation space.

### 4.3. Endogenous Reasoning State Steering

For *State Intervention*, we introduce **Endogenous Reasoning State Steering (ERSS)** to amplify ongoing reasoning and guide latent paths toward more intensified regions in the hidden space. Unlike existing steering methods that rely on handcrafted lexical triggers or externally computed control vectors, ERSS is entirely self-contained.

During decoding, we maintain a cache $\mathcal{C}_t$ of historical hidden states $h_i, i < t$ with top-percentile $\ell_2$ norms, representing highly activated reasoning states encountered so far. When a new $\ell_2$ peak is detected at step $t$, we retrieve the cached state $h^*$ most semantically aligned with the current hidden representation $h_t$:

$$h^* = \arg\max_{h_i \in \mathcal{C}_t} \cos(h_t, h_i). \qquad (7)$$

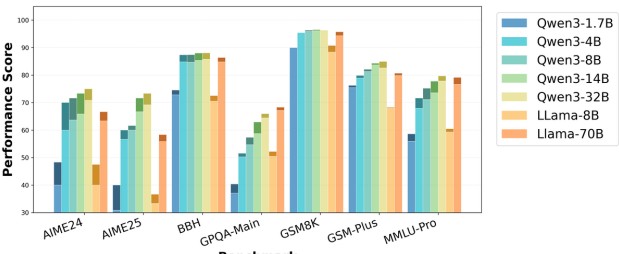

*Figure 6.* Performance across benchmarks using Adaptive Layer-wise Recursion (ALRR).

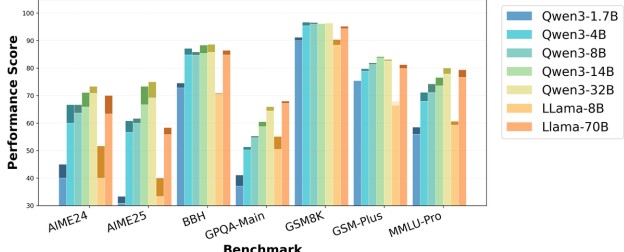

*Figure 7.* Performance across benchmarks using Endogenous Reasoning State Steering (ERSS).

The current state is then steered via additive injection:

$$h'_t = h_t + \lambda h^* \qquad (8)$$

Crucially, because $h^*$ is drawn from the same decoding trajectory and selected via maximum cosine similarity, the steering remains contextually grounded, without introducing external semantic drift or logical incoherence. The mathematical elegance of this selection lies in its self-reinforcing nature. Based on the law of cosines:

$$\|h_t+h^*\|^2 = \|h_t\|^2+\|h^*\|^2+2\|h_t\|\|h^*\|\cos(h_t, h^*) \quad (9)$$

Since $h^*$ is a high-norm state by definition, maximizing $\cos(h_t, h^*)$ inherently maximizes the $\ell_2$-norm of $h'_t$. Consequently, this process reinforces high-activation dynamics in the hidden space while preserving contextual consistency, effectively amplifying the model's latent reasoning.

### 4.4. $\ell_2$-guided Response Selection

For *Result Evaluation*, we propose $\ell_2$-**guided Response Selection (LRS)** to rank candidate outputs in an unsupervised setting. We hypothesize that a reasoning path's quality is reflected by the sustained intensity of its internal activations, particularly in the later stages of the model where high-level semantic integration occurs. For each candidate response, we compute a reasoning score $S$:

$$S = \sum_{t \in \text{output}} \sum_{l=L/3}^{L} |h_{l,t}|_2 \qquad (10)$$

This metric aggregates $\ell_2$ magnitudes from middle-to-late layers, prioritizing deep semantic processing over shallow token prediction. The candidate with the highest score $S$ is selected as the final prediction. This approach requires no external supervision and relies solely on internal representational dynamics, providing a simple yet effective mechanism for unsupervised response evaluation.

### 4.5. Experiment Results and Analysis

#### 4.5.1. MAIN RESULTS
We evaluate proposed techniques across Qwen3 series and the DeepSeek-R1-Distill-Llama family on a diverse suite

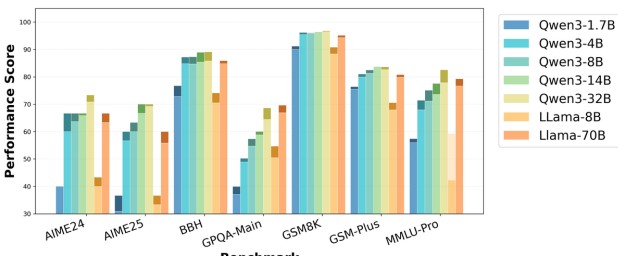

*Figure 8.* Performance across benchmarks using $\ell_2$-guided Response Selection (LRS).

of mathematical and general reasoning benchmarks. Evaluation settings are reported in Appendix E.

As illustrated in Figure 6, 7, 8, across all evaluated models and benchmarks, all three $\ell_2$-guided intervention strategies, Adaptive Layer-wise Recursion (ALRR), Endogenous State Steering (ERSS), and $\ell_2$-guided Selection (LRS), yield consistent performance gains. On average, our methods achieve a **4.51% relative improvement across all benchmarks**, and deliver **substantially larger gains of 9.13% on challenging reasoning tasks** such as AIME. Detailed numerical results and analysis are provided in Appendix G, H, I. Specifically, ALRR and ERSS successfully enhance the internal reasoning of models during the generation process, while LRS provides a robust unsupervised reranking mechanism that outperforms average performance. These results demonstrate that the $\ell_2$-norm is an effective control handle for test-time optimization.

#### 4.5.2. CROSS-BENCHMARK $\ell_2$ NORM DISTRIBUTION

To further understand $\ell_2$ norm dynamics across tasks, we visualize late-layers' average total $\ell_2$ magnitudes per sequence for each benchmark on Qwen3-8B. As shown in Figure 9, $\ell_2$ profiles exhibit substantial heterogeneity depending on the task difficulty. For challenging reasoning benchmarks such as AIME and GPQA, $\ell_2$ norms are consistently much higher than those observed on simpler and more general benchmarks like MMLU-Pro, which aligns closely with our earlier observations in Section 3 that more intensified reasoning exhibits significantly larger total norms. Furthermore, this disparity also justifies our $\ell_2$

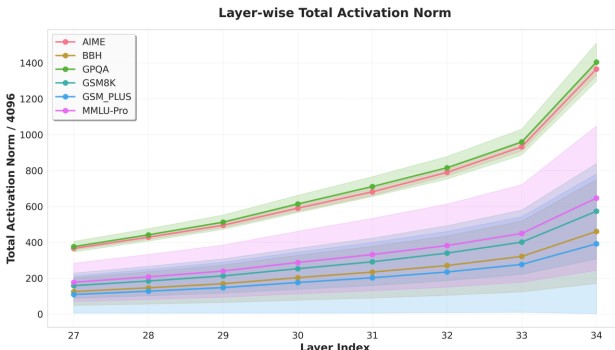

*Figure 9.* The average total $\ell_2$ norm per sequence distribution across reasoning benchmarks on Qwen3-8B per sequence (For ease of visualization, we divide them by 4096). Solid lines represent mean values, while shaded regions indicate standard deviations.

norm peak detection. The fact that a "high" norm in GSM-8k might be considered "low" in GPQA confirms that a universal cutoff for locating reasoning step is ineffective. Instead, our approach remains benchmark-agnostic and fully data-free, ensuring that interventions are consistently targeted at the model's true internal reasoning surges. Appendix K provides a detailed analysis of the relationship between different norm-based statistics and both task domain (e.g., mathematical vs. general) and task difficulty, together with our findings.

In addition, we provide a broader sensitivity and boundary-condition analysis in Appendix J, including ablations of different hyperparameters and design choices in three methods; robustness under different SAE configurations; compatibility across different test-time enhancement methods; and analyses of failure modes and boundary conditions on non-reasoning benchmarks.

## 5. Discussion

**Insight.** Recent work on enhancing LLM reasoning mainly follows two paradigms. Explicit CoT expands reasoning in the output space but incurs substantial token and latency overhead. Latent CoT internalizes this process, yet often requires new token types or architectural modifications, limiting generality. More broadly, explicit CoT can be seen as scaling computation along the output dimension by generating more reasoning tokens, while latent-space scaling has also been shown possible by Looped Transformers. This shifts the key question to where extra computation should be allocated across layers. Our work addresses this question and identifies hidden-state $\ell_2$ norm as a training-free signal of reasoning-critical locations.

Current reasoning control is also dominated by decode-stage signals, such as entropy-based criteria for CoT generation or token-level signals used in reinforcement learning

and reranking. While useful, these signals mainly operate at the output level and provide limited access to fine-grained layer-wise dynamics, which is especially inconvenient for studying or controlling latent thought. In this sense, we hope to encourage further exploration of simple intrinsic signals already present inside the model. Our work provides one such example: token-wise and layer-wise variations in hidden-state $\ell_2$ norm offer a lightweight way to probe internal reasoning states without additional training or architectural changes.

**Interpretation.** The $\ell_2$ norm of a hidden representation admits an intuitive physical analogy: it measures representational energy, and peaks where the model allocates computation most intensively. This intuition is supported by prior work in computer vision, which shows that for cross-entropy-trained models, in-distribution samples tend to have larger feature norms than OOD samples because increasing the correct logit often requires increasing feature magnitude. We argue that a similar mechanism holds in autoregressive LLMs: tokens that are critical for a correct continuation require sharper logit separation, which induces larger hidden-state norms. Under this view, $\ell_2$ norm peaks mark computation-critical tokens in the reasoning process.

At the same time, our benchmark analyses suggest that the relation between $\ell_2$ norm and reasoning is not captured by a single explanation. Both task difficulty and task domain are associated with different norm statistics, indicating that $\ell_2$ norm is not a complete or one-dimensional characterization of reasoning. Rather, it likely reflects a richer mixture of computational factors whose mechanism remains to be understood. We therefore view our results not as a complete account of latent reasoning, but as evidence that simple endogenous signals such as $\ell_2$ norm can reveal meaningful internal structure and provide a useful starting point for further study.

## 6. Conclusion

We show that hidden-state $\ell_2$ norm provides a simple endogenous signal of reasoning intensity in LLMs. Theoretically and empirically, it tracks layer-wise reasoning activity and bounds the activation of SAE reasoning features. Based on this observation, we develop three training-free inference-time methods that improve reasoning performance across multiple benchmarks. More broadly, our results suggest that simple internal signals can provide useful access to latent computation, offering both practical tools for reasoning enhancement and a promising direction for mechanistic interpretability.

## Acknowledgement

This work is supported by the National Natural Science Foundation of China (No.62576013, No.U23A20468).

## Impact Statement

This work explores the internal reasoning dynamics in LLMs. We identify hidden-state $l_2$ norms as reliable indicators of reasoning intensity and leverage them for training-free test-time reasoning enhancement, contributing to the development of more transparent and computationally efficient AI. The proposed method does not require additional data and optimization and does not introduce additional ethical or privacy concerns. While these techniques empower users to amplify reasoning capabilities, we emphasize that they measure the model's internal deliberation intensity rather than absolute factual correctness, encouraging a more nuanced and responsible deployment of LLMs in critical decision-making tasks.

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

# A. Related Work

## A.1. LLM Reasoning Behaviors and Test-Time Scaling

Recent advances in Large Language Models have demonstrated remarkable reasoning capabilities, particularly in specialized reasoning-oriented models such as DeepSeek-R1 (Guo et al., 2025; Shao et al., 2024), OpenAI's o1 (Achiam et al., 2023) and Qwen-3 model family (Yang et al., 2025). These models utilize extended Chain-of-Thought (CoT) (Wei et al., 2022; Chen et al., 2025b) sequences to solve complex problems. Recent studies have focused on the characteristics of these reasoning trajectories, demonstrating that allocating more compute during inference, either through searching multiple paths (Yao et al., 2023), generating longer sequences (Chen et al., 2025a; Ding et al., 2025b), majority voting (Fu et al., 2025) or inserting special tokens (Zhao et al., 2025; Muennighoff et al., 2025), directly correlates with improved accuracy. A key focus within this area is the identification of critical reasoning tokens at the output level, leveraging rollout sampling (Lin et al., 2024; Fang et al., 2026b), output entropy (Wang et al., 2025) or mutual information with correctness (Qian et al., 2025). While these works successfully highlight where reasoning occurs in the output sequence, they remain largely descriptive and retrospective, failing to explore the internal dynamics of how reasoning is computationally represented in the model's hidden space. This lack of mechanistic insight limits the ability to control reasoning behaviors more efficiently. Instead of waiting for a "thinking token" to be decoded, our work seeks to identify reasoning activity as it unfolds across hidden layers.

## A.2. Mechanistic Interpretability and Reasoning Control

To move beyond output-level analysis, recent work has turned to mechanistic interpretability tools to probe the internal representations underlying reasoning. Sparse Autoencoders (SAEs) are utilized to decompose polysemantic neural activations into sparse, interpretable features (Cunningham et al., 2023; Bricken et al., 2023; Liao et al., 2026). Recent studies isolate features specifically triggered during reflection, backtracking, and logical deduction (Galichin et al., 2025; Zhang et al., 2025b; Ding et al., 2025a). These insights have enabled representation engineering, where identified reasoning directions are used as steering vectors to induce longer CoT or improve performance by intervening directly in the activation space (Li et al., 2025b; Venhoff et al., 2025; Chen et al., 2025b; Fang et al., 2026a). However, existing interpretability and control frameworks face two major limitations. First, most studies focus primarily on identifying and manipulating individual reasoning features, rather than systematically characterizing how reasoning activity itself evolves across model depth. Second, they are heavily probe-dependent. Identifying and steering reasoning features requires training expensive, model-specific SAEs and often relies on labeling features based on token-level heuristics (Fang et al., 2026c; Galichin et al., 2025). This makes the methods difficult to generalize across different architectures or scales without significant overhead. Our work addresses these gaps by analyzing LLM layer-wise reasoning dynamics and identifying the $\ell_2$ norm as an endogenous, probe-free indicator of reasoning intensity. Unlike SAE-based steering, our approach senses the model's internal "thinking" state adaptively, offering a more universal and efficient mechanism for understanding and controlling reasoning.

# B. Theoretical Analysis Between $\ell_2$ Norm and SAE Reasoning Feature Activation

After observing significant layer-wise variations in SAE activations across models, we note that training a separate sparse autoencoder (SAE) for every layer of every model incurs prohibitive computational costs and suffers from poor scalability. To address this, we seek simpler and more general-purpose proxies that can effectively capture the presence of reasoning-related features in hidden states.

Theoretically, we find that the magnitude of SAE feature activations is inherently constrained by the $\ell_2$ norm of the hidden state. Below, we present our derivation.

## B.1. Preliminaries and Notation

**Sparse Autoencoder (SAE).** A Sparse Autoencoder consists of an encoder $f_{\text{enc}} : \mathbb{R}^d \to \mathbb{R}^m$ and a decoder $f_{\text{dec}} : \mathbb{R}^m \to \mathbb{R}^d$, where $m \gg d$ represents the overcomplete feature dimension. Let $\mathbf{h}_t \in \mathbb{R}^d$ denote the hidden activation at layer $\ell$ and token position $t$ in the input sequence. The encoder maps the hidden state to the **overcomplete SAE feature** $\mathbf{z}_t \in \mathbb{R}^m$ as:

$$\mathbf{z}_t = f_{\text{enc}}(\mathbf{h}_t) = \text{ReLU}(\mathbf{W}_{\text{enc}}\mathbf{h}_t + \mathbf{b}_{\text{enc}}), \tag{11}$$

where $\mathbf{W}_{\text{enc}} \in \mathbb{R}^{m \times d}$ is the encoding matrix and $\mathbf{b}_{\text{enc}} \in \mathbb{R}^m$ is the bias vector. The decoder reconstructs the original hidden state from the SAE feature as:

$$\hat{\mathbf{h}}_t = f_{\text{dec}}(\mathbf{z}_t) = \mathbf{W}_{\text{dec}}\mathbf{z}_t = \sum_{i=1}^{m} z_{i,t}\mathbf{d}_i, \tag{12}$$

where $\mathbf{d}_i$ denotes the $i$-th column of $\mathbf{W}_{\text{dec}} \in \mathbb{R}^{d \times m}$ and $z_{i,t}$ is the $i$-th dimension of the SAE feature $\mathbf{z}_t$.

**Training Objective.** SAE is trained to minimize the following loss function, which jointly enforces reconstruction fidelity and feature sparsity:

$$\mathcal{L}_{\text{SAE}} = \mathbb{E}_{\mathbf{h}}\left[\|\mathbf{h} - \hat{\mathbf{h}}\|_2^2 + \lambda\|\mathbf{z}\|_1\right], \tag{13}$$

where the first term is the $\ell_2$ reconstruction loss to preserve the original hidden state information, and the second term is the $\ell_1$ regularization term with coefficient $\lambda > 0$ to promote sparsity of the SAE feature $\mathbf{z}$.

**Differential Feature Activation.** For a given reasoning problem, we generate two response sequences: a **thinking version** and a **non-thinking version**. Let $\mathbf{z}_t^{\text{think}} \in \mathbb{R}^m$ and $\mathbf{z}_t^{\text{non}} \in \mathbb{R}^m$ denote the SAE features at token position $t$ for the thinking and non-thinking response sequences, respectively. We first compute the **average activation** of the $i$-th SAE feature dimension across all token positions for each version, then define the **differential feature activation** as the difference between these two averages:

$$\Delta z_i = \mathbb{E}_t\left[z_{i,t}^{\text{think}}\right] - \mathbb{E}_t\left[z_{i,t}^{\text{non}}\right], \quad i = 1, \ldots, m, \tag{14}$$

where $z_{i,t}^{\text{think}}$ and $z_{i,t}^{\text{non}}$ are the $i$-th dimension of $\mathbf{z}_t^{\text{think}}$ and $\mathbf{z}_t^{\text{non}}$, respectively, and $\mathbb{E}_t[\cdot]$ denotes the expectation (average) over all token positions $t$ in the response sequence.

### B.1.1. FUNDAMENTAL ASSUMPTIONS

**Assumption B.1** (Decoder Approximate Orthonormality). The decoder basis vectors $\{\mathbf{d}_i\}_{i=1}^m$ satisfy approximate orthonormality(Bricken et al., 2023):

$$\langle\mathbf{d}_i, \mathbf{d}_j\rangle = \delta_{ij} + \mathcal{O}(\epsilon_{\text{orth}}), \quad \|\mathbf{d}_i\|_2 = 1 + \mathcal{O}(\epsilon_{\text{orth}}), \tag{15}$$

where $\epsilon_{\text{orth}} \ll 1$ is a small constant representing deviation from exact orthonormality.

**Assumption B.2** (Reconstruction Quality). The SAE achieves good reconstruction quality, i.e., there exists a small constant $\epsilon_{\text{recon}} > 0$ such that:

$$\|\mathbf{h}_t - f_{\text{dec}}(f_{\text{enc}}(\mathbf{h}_t))\|_2 \le \epsilon_{\text{recon}}, \quad \forall t. \tag{16}$$

**Assumption B.3** (Semantic Inclusion Property at Response Level). For a given reasoning problem, we generate two complete response sequences: a *thinking response* $\mathbf{H}_{\text{think}} = \{\mathbf{h}_t^{\text{think}}\}_{t=1}^{T_{\text{think}}}$ and a *non-thinking response* $\mathbf{H}_{\text{non}} = \{\mathbf{h}_t^{\text{non}}\}_{t=1}^{T_{\text{non}}}$, where $T_{\text{think}}$ and $T_{\text{non}}$ denote the token lengths of the two responses, and $\mathbf{h}_t^{\text{think}}, \mathbf{h}_t^{\text{non}} \in \mathbb{R}^d$ are the hidden state embeddings of the $t$-th token in each sequence respectively.

At the response level, the semantic information of $\mathbf{H}_{\text{think}}$ is assumed to contain the base semantic content of $\mathbf{H}_{\text{non}}$ plus additional reasoning-specific semantic information. This inclusion relationship is formalized based on the **expectation (average) of hidden state embeddings** over the entire response sequence, which avoids relying on local token alignment and reflects a global semantic constraint.

Formally, let $\bar{\mathbf{h}}_{\text{think}} = \mathbb{E}_t\left[\mathbf{h}_t^{\text{think}}\right] = \frac{1}{T_{\text{think}}}\sum_{t=1}^{T_{\text{think}}} \mathbf{h}_t^{\text{think}}$ denote the average hidden state embedding of the thinking response, and $\bar{\mathbf{h}}_{\text{non}} = \mathbb{E}_t\left[\mathbf{h}_t^{\text{non}}\right] = \frac{1}{T_{\text{non}}}\sum_{t=1}^{T_{\text{non}}} \mathbf{h}_t^{\text{non}}$ denote that of the non-thinking response. There exist two vectors $\mathbf{h}_{\text{base}} \in \mathbb{R}^d$ (global base semantic embedding) and $\mathbf{h}_{\text{reasoning}} \in \mathbb{R}^d$ (global reasoning-specific semantic embedding, $\mathbf{h}_{\text{reasoning}} \ne \mathbf{0}$) such that:

$$\bar{\mathbf{h}}_{\text{think}} = \mathbf{h}_{\text{base}} + \mathbf{h}_{\text{reasoning}}, \tag{17}$$

and the average hidden state of the non-thinking response satisfies:

$$\bar{\mathbf{h}}_{\text{non}} = \mathbf{h}_{\text{base}} + \boldsymbol{\epsilon}_{\text{noise}}, \tag{18}$$

where $\epsilon_{\text{noise}} \ll \|\mathbf{h}_{\text{reasoning}}\|_2$. This ensures that the deviation of the non-thinking response's average embedding from the base semantic embedding is negligible compared to the reasoning-specific semantic component.

*Justification.* While local token embeddings may vary due to context-dependent fluctuations (e.g., different wording or syntax), the average hidden state over the entire response sequence can effectively characterize the global core semantics of the response, filtering out local noise. A thinking response, which incorporates deliberate reasoning steps, does not replace the basic understanding of the problem but adds reasoning-specific content on top of it—this is reflected in the global average embedding by the additional $\mathbf{h}_{\text{reasoning}}$ component.

This assumption is supported by two lines of evidence: (i) Empirical observations show that removing thinking tokens degrades reasoning performance while preserving basic language understanding (Guo et al., 2025), indicating that thinking content is an incremental semantic component rather than a replacement for base semantics; (ii) Both responses target the same reasoning problem, so their global average embeddings must share a common base semantic foundation ($\mathbf{h}_{\text{base}}$), with the difference primarily arising from the presence or absence of reasoning-specific content (Li et al., 2025b; Venhoff et al., 2025). Using average embeddings to formalize the inclusion relationship also aligns with the statistical nature of SAE feature activation analysis (i.e., average activation of SAE features), ensuring consistency across theoretical assumptions and subsequent analyses.

**Assumption B.4** (Approximate Orthogonality of Reasoning Component)**.** For the global base semantic embedding $\mathbf{h}_{\text{base}}$ and global reasoning-specific semantic embedding $\mathbf{h}_{\text{reasoning}}$ (defined in Assumption B.3), there always exists at least one such decomposition where the two components are approximately orthogonal. Formally:

$$|\langle \mathbf{h}_{\text{base}}, \mathbf{h}_{\text{reasoning}} \rangle| \leq \beta \|\mathbf{h}_{\text{base}}\|_2 \|\mathbf{h}_{\text{reasoning}}\|_2, \tag{19}$$

where $\beta \in (0,1)$ is a small constant satisfying $\beta \ll 1$, and $\langle \cdot, \cdot \rangle$ denotes the inner product. This implies the cosine similarity between the two components is bounded by $\beta$, ensuring weak correlation.

*Justification.* We take this approximate orthogonality as a modeling prior to separate base semantics from reasoning dynamics. This aligns with cognitive neuroscience evidence that human reasoning functions largely independently of core language processing (Amalric & Dehaene, 2016), extending to LLMs where semantic and reasoning representations can be effectively disentangled (Li et al., 2025b). It also conforms to representation learning principles that independent feature factors tend to be encoded as weakly correlated components (Bengio et al., 2013). Moreover, modern reasoning-oriented LLMs are optimized via additional reasoning-specific signals on top of pretrained language models (Guo et al., 2025), naturally fostering relatively separable semantic and reasoning representations.

**Assumption B.5** (Feature Disentanglement via SAE)**.** When sufficiently trained, the SAE learns disentangled monosemantic features that capture the orthogonal decomposition in Assumption B.4. These features naturally partition into two essentially disjoint index sets:

- $\mathcal{I}_{\text{base}}$: Indices of features capturing base semantic properties, with average activation shared by both thinking and non-thinking responses;
- $\mathcal{I}_{\text{reasoning}}$: Indices of features capturing reasoning-specific properties, whose average activation is predominantly induced by thinking responses.

It holds that $|\mathcal{I}_{\text{base}} \cap \mathcal{I}_{\text{reasoning}}| \ll \min(|\mathcal{I}_{\text{base}}|, |\mathcal{I}_{\text{reasoning}}|)$. Let $z_i^{\text{base}}$ and $z_i^{\text{reasoning}}$ denote the average activation coefficients of the $i$-th SAE feature for base semantics and reasoning, respectively. Then the linear decomposition of the average embeddings via SAE decoder basis $\{\mathbf{d}_i\}$ satisfies:

$$\mathbf{h}_{\text{base}} \approx \sum_{i \in \mathcal{I}_{\text{base}}} z_i^{\text{base}} \mathbf{d}_i, \tag{20}$$

$$\mathbf{h}_{\text{reasoning}} \approx \sum_{i \in \mathcal{I}_{\text{reasoning}}} z_i^{\text{reasoning}} \mathbf{d}_i, \tag{21}$$

with $\|\mathbf{h}_{\text{base}} - \sum_{i \in \mathcal{I}_{\text{base}}} z_i^{\text{base}} \mathbf{d}_i\|_2 \leq \epsilon_{\text{dis}}$ and $\|\mathbf{h}_{\text{reasoning}} - \sum_{i \in \mathcal{I}_{\text{reasoning}}} z_i^{\text{reasoning}} \mathbf{d}_i\|_2 \leq \epsilon_{\text{dis}}$, where $\epsilon_{\text{dis}} > 0$ is a small constant.

*Justification.* The $\ell_1$ penalty in Equation (13) encourages the SAE to learn monosemantic features that capture specific interpretable patterns (Cunningham et al., 2023). Given the approximate orthogonality of $\mathbf{h}_{\text{base}}$ and $\mathbf{h}_{\text{reasoning}}$, the SAE optimizes to separate these two components into disjoint feature sets—consistent with findings that SAEs can disentangle target representations from redundant signals (Chen et al., 2025b). This separation minimizes reconstruction error while maintaining sparsity, leveraging the SAE's inherent capacity to disentangle independent semantic factors (Bengio et al., 2013) and thus capturing the orthogonal decomposition of base semantics and reasoning.

**Assumption B.6** (Base Feature Consistency). For base semantic features, the activations at thinking and non-thinking tokens are approximately equal:

$$z_i^{\text{think}} \approx z_i^{\text{non}}, \quad \forall i \in \mathcal{I}_{\text{base}}, \tag{22}$$

or more precisely, $|z_i^{\text{think}} - z_i^{\text{non}}| \leq \gamma z_i^{\text{non}}$ for small $\gamma \ll 1$.

*Justification:* Since both thinking and non-thinking tokens process the same problem, they should activate similar base semantic features. By Assumption B.3, $\bar{\mathbf{h}}_{\text{think}}$ includes $\mathbf{h}_{\text{base}}$ which is approximately $\bar{\mathbf{h}}_{\text{non}}$ (up to noise $\epsilon_{\text{noise}}$). Through the SAE encoder, this translates to similar activations on $\mathcal{I}_{\text{base}}$ features.

### B.1.2. PRELIMINARY LEMMAS

**Lemma B.7** ($\ell_2$ Energy Decomposition). *Under Assumption B.1, the $\ell_2$ norm of the SAE reconstruction satisfies:*

$$\|\hat{\mathbf{h}}\|_2^2 = \sum_{i=1}^{m} z_i^2 + \mathcal{O}(\epsilon_{orth}\|\mathbf{z}\|_2^2). \tag{23}$$

*Proof.* By the definition of the decoder:

$$\|\hat{\mathbf{h}}\|_2^2 = \left\|\sum_{i=1}^{m} z_i \mathbf{d}_i\right\|_2^2 = \sum_{i=1}^{m}\sum_{j=1}^{m} z_i z_j \langle \mathbf{d}_i, \mathbf{d}_j \rangle. \tag{24}$$

By Assumption B.1, $\langle \mathbf{d}_i, \mathbf{d}_j \rangle = \delta_{ij} + \mathcal{O}(\epsilon_{\text{orth}})$. Thus:

$$\begin{aligned}
\|\hat{\mathbf{h}}\|_2^2 &= \sum_{i=1}^{m} z_i^2 + \sum_{i \neq j} z_i z_j \mathcal{O}(\epsilon_{\text{orth}}) \\
&= \sum_{i=1}^{m} z_i^2 + \mathcal{O}\left(\epsilon_{\text{orth}} \sum_{i,j} |z_i z_j|\right) \\
&= \sum_{i=1}^{m} z_i^2 + \mathcal{O}(\epsilon_{\text{orth}}\|\mathbf{z}\|_1^2) \\
&= \sum_{i=1}^{m} z_i^2 + \mathcal{O}(\epsilon_{\text{orth}}\|\mathbf{z}\|_2^2),
\end{aligned} \tag{25}$$

where the last step uses $\|\mathbf{z}\|_1^2 \leq m\|\mathbf{z}\|_2^2$ and absorbs the constant into $\mathcal{O}(\cdot)$. $\square$

**Lemma B.8** (Reconstruction Error Propagation). *Under Assumptions B.1 and B.2:*

$$\left|\|\mathbf{h}\|_2^2 - \sum_{i=1}^{m} z_i^2\right| \leq 2\epsilon_{recon}\|\mathbf{h}\|_2 + 3\epsilon_{recon}^2 + \mathcal{O}(\epsilon_{orth}\|\mathbf{z}\|_2^2). \tag{26}$$

*Proof.* Let $\mathbf{r} = \mathbf{h} - \hat{\mathbf{h}}$ be the reconstruction residual. By Assumption B.2, $\|\mathbf{r}\|_2 \leq \epsilon_{\text{recon}}$. Then:

$$\|\mathbf{h}\|_2^2 = \|\hat{\mathbf{h}} + \mathbf{r}\|_2^2 = \|\hat{\mathbf{h}}\|_2^2 + \|\mathbf{r}\|_2^2 + 2\langle \hat{\mathbf{h}}, \mathbf{r} \rangle. \tag{27}$$

By Cauchy-Schwarz inequality:

$$|\langle \hat{\mathbf{h}}, \mathbf{r} \rangle| \leq \|\hat{\mathbf{h}}\|_2 \|\mathbf{r}\|_2 \leq \epsilon_{\text{recon}}(\|\mathbf{h}\|_2 + \|\mathbf{r}\|_2) \leq \epsilon_{\text{recon}}\|\mathbf{h}\|_2 + \epsilon_{\text{recon}}^2. \tag{28}$$

Combining with Lemma B.7:

$$\|\mathbf{h}\|_2^2 = \sum_{i=1}^{m} z_i^2 + \mathcal{O}(\epsilon_{\text{orth}}\|\mathbf{z}\|_2^2) + \|\mathbf{r}\|_2^2 + 2\langle \hat{\mathbf{h}}, \mathbf{r} \rangle. \tag{29}$$

Taking the absolute value of the difference:

$$\left| \|\mathbf{h}\|_2^2 - \sum_{i=1}^{m} z_i^2 \right| \leq 2\epsilon_{\text{recon}}\|\mathbf{h}\|_2 + 3\epsilon_{\text{recon}}^2 + \mathcal{O}(\epsilon_{\text{orth}}\|\mathbf{z}\|_2^2). \tag{30}$$

$\square$

## B.2. Main Theorem

**Theorem B.9** (SAE Differential Features and $\ell_2 NormCorrelation$). *Under Assumptions B.1–B.6, let $\Delta_{obs} = \|\bar{\mathbf{h}}_{think}\|_2^2 - \|\bar{\mathbf{h}}_{non}\|_2^2$ denote the observable*

$$\ell_2$$

*norm difference, and let $\epsilon_{total} = E_{recon}^{think} + E_{recon}^{non}$ with $E_{recon} = 2\epsilon_{recon}\|\mathbf{h}\|_2 + 3\epsilon_{recon}^2 + \mathcal{O}(\epsilon_{orth}\|\mathbf{z}\|_2^2)$ being the reconstruction error for a single representation. Let $k \ll |\mathcal{I}_{reasoning}|$ be the number of active reasoning features per token for the sparse SAE. Then the sum of differential feature activations on reasoning features satisfies the two-sided bound:*

$$\frac{1 - \mathcal{O}(\beta)}{\sqrt{1 + \mathcal{O}(\epsilon_{orth})}} \sqrt{\Delta_{obs} - \epsilon_{total}} \leq \sum_{i \in \mathcal{I}_{reasoning}} \Delta z_i \leq \sqrt{k}\sqrt{\Delta_{obs} + \epsilon_{total}}. \tag{31}$$

*In particular, when $\epsilon_{total} \ll \Delta_{obs}$ and $\beta \ll 1$ (small approximation and reconstruction errors), the aggregate differential activation is tightly characterized by the observable norm difference:*

$$\sum_{i \in \mathcal{I}_{reasoning}} \Delta z_i = \Theta\left( \sqrt{\|\bar{\mathbf{h}}_{think}\|_2^2 - \|\bar{\mathbf{h}}_{non}\|_2^2} \right), \tag{32}$$

*where the $\Theta(\cdot)$ constant factors depend only on $\beta$, $\epsilon_{orth}$, and the sparse activation count $k$ of the SAE.*

*Proof.* The proof proceeds in seven steps.

**Step 1: Decompose thinking token representation.** By Assumption B.3:

$$\bar{\mathbf{h}}_{\text{think}} = \mathbf{h}_{\text{base}} + \mathbf{h}_{\text{reasoning}}, \quad \bar{\mathbf{h}}_{\text{non}} = \mathbf{h}_{\text{base}} + \boldsymbol{\epsilon}_{\text{noise}}. \tag{33}$$

Computing the squared

$$\ell_2$$

norms:

$$\|\bar{\mathbf{h}}_{\text{think}}\|_2^2 = \|\mathbf{h}_{\text{base}}\|_2^2 + \|\mathbf{h}_{\text{reasoning}}\|_2^2 + 2\langle \mathbf{h}_{\text{base}}, \mathbf{h}_{\text{reasoning}}\rangle, \tag{34}$$

$$\|\bar{\mathbf{h}}_{\text{non}}\|_2^2 = \|\mathbf{h}_{\text{base}}\|_2^2 + \|\boldsymbol{\epsilon}_{\text{noise}}\|_2^2 + 2\langle \mathbf{h}_{\text{base}}, \boldsymbol{\epsilon}_{\text{noise}}\rangle. \tag{35}$$

**Step 2: Apply orthogonality and bound error terms.** By Assumption B.4 and Cauchy-Schwarz:

$$|\langle \mathbf{h}_{\text{base}}, \mathbf{h}_{\text{reasoning}}\rangle| \leq \beta\|\mathbf{h}_{\text{base}}\|_2\|\mathbf{h}_{\text{reasoning}}\|_2, \tag{36}$$

$$|\langle \mathbf{h}_{\text{base}}, \boldsymbol{\epsilon}_{\text{noise}}\rangle| \leq \epsilon_{\text{noise}}\|\mathbf{h}_{\text{base}}\|_2, \tag{37}$$

with $\|\boldsymbol{\epsilon}_{\text{noise}}\|_2 \leq \epsilon_{\text{noise}} \ll \|\mathbf{h}_{\text{reasoning}}\|_2$. Subtracting:

$$\|\bar{\mathbf{h}}_{\text{think}}\|_2^2 - \|\bar{\mathbf{h}}_{\text{non}}\|_2^2 = \|\mathbf{h}_{\text{reasoning}}\|_2^2 - \|\boldsymbol{\epsilon}_{\text{noise}}\|_2^2 + 2\langle \mathbf{h}_{\text{base}}, \mathbf{h}_{\text{reasoning}}\rangle - 2\langle \mathbf{h}_{\text{base}}, \boldsymbol{\epsilon}_{\text{noise}}\rangle$$

$$\geq \|\mathbf{h}_{\text{reasoning}}\|_2^2 - 2\beta\|\mathbf{h}_{\text{base}}\|_2\|\mathbf{h}_{\text{reasoning}}\|_2 - 2\epsilon_{\text{noise}}\|\mathbf{h}_{\text{base}}\|_2 - \epsilon_{\text{noise}}^2. \tag{38}$$

**Step 3: Lower and upper bounds on the**

$$\ell_2$$

**norm difference.** From Step 2, we obtain a **lower bound**:

$$\|\bar{\mathbf{h}}_{\text{think}}\|_2^2 - \|\bar{\mathbf{h}}_{\text{non}}\|_2^2 \geq (1-\delta)\|\mathbf{h}_{\text{reasoning}}\|_2^2, \tag{39}$$

where $\delta = 2\beta \frac{\|\mathbf{h}_{\text{base}}\|_2}{\|\mathbf{h}_{\text{reasoning}}\|_2} + o(1)$, assuming $\|\mathbf{h}_{\text{reasoning}}\|_2 \sim \|\mathbf{h}_{\text{base}}\|_2$.

Similarly, using the upper bound on the inner product:

$$\|\bar{\mathbf{h}}_{\text{think}}\|_2^2 - \|\bar{\mathbf{h}}_{\text{non}}\|_2^2 \leq (1+\delta)\|\mathbf{h}_{\text{reasoning}}\|_2^2 + \epsilon_{\text{noise}}^2. \tag{40}$$

Thus, the

$$\ell_2$$

norm difference sandwiches the reasoning energy:

$$(1-\delta)\|\mathbf{h}_{\text{reasoning}}\|_2^2 \lesssim \|\bar{\mathbf{h}}_{\text{think}}\|_2^2 - \|\bar{\mathbf{h}}_{\text{non}}\|_2^2 \lesssim (1+\delta)\|\mathbf{h}_{\text{reasoning}}\|_2^2.$$

**Step 4: Connect to SAE feature decomposition.** By Assumptions B.5 and Lemma B.7:

$$\|\mathbf{h}_{\text{reasoning}}\|_2^2 = \sum_{i \in \mathcal{I}_{\text{reasoning}}} (z_i^{\text{reasoning}})^2 + \mathcal{O}(\epsilon_{\text{orth}}\|\mathbf{z}^{\text{reasoning}}\|_2^2). \tag{41}$$

By Assumption B.6, base features change little: $\Delta z_i = \mathcal{O}(\gamma z_i^{\text{non}})$ for $i \in \mathcal{I}_{\text{base}}$, so

$$\Delta z_i \approx \begin{cases} 0, & i \in \mathcal{I}_{\text{base}}, \\ z_i^{\text{reasoning}}, & i \in \mathcal{I}_{\text{reasoning}}. \end{cases}$$

Hence:

$$\sum_{i \in \mathcal{I}_{\text{reasoning}}} \Delta z_i \approx \sum_{i \in \mathcal{I}_{\text{reasoning}}} z_i^{\text{reasoning}}.$$

**Step 5: Bounds via sparsity and non-negativity.** Since SAE uses ReLU activation, all feature activations are non-negative: $z_i \geq 0$. For any non-negative sequence, the sum is at least the $\ell_2$ norm:

$$\sum_{i \in \mathcal{I}_{\text{reasoning}}} z_i^{\text{reasoning}} \geq \sqrt{\sum_{i \in \mathcal{I}_{\text{reasoning}}} (z_i^{\text{reasoning}})^2} = \|\mathbf{h}_{\text{reasoning}}\|_2 \sqrt{1 + \mathcal{O}(\epsilon_{\text{orth}})}. \tag{42}$$

Moreover, by the sparsity of SAEs, only a small number $k$ of reasoning features are active per token (i.e., $|\{i : z_i^{\text{reasoning}} > 0\}| \leq k$). Applying Cauchy-Schwarz over the active set yields:

$$\sum_{i \in \mathcal{I}_{\text{reasoning}}} z_i^{\text{reasoning}} \leq \sqrt{k}\, \|\mathbf{h}_{\text{reasoning}}\|_2. \tag{43}$$

Thus, the total change in reasoning features satisfies:

$$\|\mathbf{h}_{\text{reasoning}}\|_2 \lesssim \sum_{i \in \mathcal{I}_{\text{reasoning}}} \Delta z_i \lesssim \sqrt{k}\, \|\mathbf{h}_{\text{reasoning}}\|_2.$$

**Step 6: Relate to observable**

$$\ell_2$$

**norm difference.** From the sandwich bounds (39)–(40), we have:

$$\|\mathbf{h}_{\text{reasoning}}\|_2 \asymp \sqrt{\|\bar{\mathbf{h}}_{\text{think}}\|_2^2 - \|\bar{\mathbf{h}}_{\text{non}}\|_2^2},$$

where $\asymp$ means "bounded above and below up to constants depending on $\beta$".

Now incorporate reconstruction error via Lemma B.8. The observed norms satisfy:

$$\left| \|\mathbf{h}\|_2^2 - \sum_i (z_i)^2 \right| \le E_{\text{recon}},$$

with $E_{\text{recon}} = \mathcal{O}(\epsilon_{\text{recon}} \|\mathbf{h}\|_2 + \epsilon_{\text{recon}}^2 + \epsilon_{\text{orth}} \|\mathbf{z}\|_2^2)$. Define total error $\epsilon_{\text{total}} = E_{\text{recon}}^{\text{think}} + E_{\text{recon}}^{\text{non}}$. Then:

$$\left| \left( \|\bar{\mathbf{h}}_{\text{think}}\|_2^2 - \|\bar{\mathbf{h}}_{\text{non}}\|_2^2 \right) - \left( \sum_i (z_i^{\text{think}})^2 - (z_i^{\text{non}})^2 \right) \right| \le \epsilon_{\text{total}}.$$

Since $\sum_i (z_i^{\text{think}})^2 - (z_i^{\text{non}})^2 \approx \|\mathbf{h}_{\text{reasoning}}\|_2^2$, we conclude:

$$\|\mathbf{h}_{\text{reasoning}}\|_2^2 \in [\Delta_{\text{obs}} - \epsilon_{\text{total}}, \, \Delta_{\text{obs}} + \epsilon_{\text{total}}], \quad \text{where } \Delta_{\text{obs}} := \|\bar{\mathbf{h}}_{\text{think}}\|_2^2 - \|\bar{\mathbf{h}}_{\text{non}}\|_2^2.$$

**Step 7: Final two-sided bound on $\sum \Delta z_i$.** Combining (42), (43), and Step 6:

$$\frac{1 - \mathcal{O}(\beta)}{\sqrt{|\mathcal{I}_{\text{reasoning}}|}} \sqrt{\Delta_{\text{obs}} - \epsilon_{\text{total}}} \lesssim \sum_{i \in \mathcal{I}_{\text{reasoning}}} \Delta z_i$$

$$\lesssim \sqrt{|\mathcal{I}_{\text{reasoning}}|} \sqrt{\Delta_{\text{obs}} + \epsilon_{\text{total}}}. \tag{44}$$

In particular, when $\epsilon_{\text{total}} \ll \Delta_{\text{obs}}$ and $\beta \ll 1$, we obtain the clean two-sided control:

$$\sum_{i \in \mathcal{I}_{\text{reasoning}}} \Delta z_i = \Theta\left( \sqrt{\|\bar{\mathbf{h}}_{\text{think}}\|_2^2 - \|\bar{\mathbf{h}}_{\text{non}}\|_2^2} \right),$$

up to factors depending on $|\mathcal{I}_{\text{reasoning}}|$ and small constants.

This shows that the aggregate change in reasoning features is **both lower- and upper-bounded** by the observable

$$\ell_2$$

norm difference, completing the proof. $\qquad\square$

### B.3. Further Discussion

Our finding that high $\ell_2$-norm tokens tend to mark reasoning-intensive steps admits an interesting connection to a recent theoretical analysis from the computer vision community. Xu et al. (Park et al., 2023) established that the feature norm in deep classifiers is intrinsically linked to the cross-entropy loss: minimizing the loss drives the model to produce both a large norm and a confident alignment with the correct class. The norm, in effect, reflects how strongly the network is pushed to commit to a prediction.

This insight extends naturally to the sequential cross-entropy objective used in language model training. At each generation step, the loss for predicting the ground-truth next token $k$ is $\mathcal{L}_t = -\log p_k$. The gradient with respect to the hidden state $\mathbf{h}_t$ takes the form:

$$\frac{\partial \mathcal{L}_t}{\partial \mathbf{h}_t} = \sum_j \left( p_j - \mathbf{1}[j = k] \right) \mathbf{e}_j,$$

where $p_j$ is the predicted probability of token $j$ and $\mathbf{e}_j$ is its output embedding. Crucially, the magnitude of this gradient is large precisely when the model is uncertain ($p_k \ll 1$). There are two notable scenarios where this condition holds. First, tokens that are inherently difficult to learn—due to rarity, ambiguity, or long-range dependencies—naturally exhibit low initial confidence and thus receive disproportionately large updates. Second, contemporary language models are increasingly trained with reasoning-intensive data, both during large-scale pretraining (e.g., code, mathematics, and logical

reasoning corpora) and during post-training stages (e.g., instruction tuning and reinforcement learning with reasoning traces). In these data, tokens that form reasoning patterns, such as deductive connectors or intermediate conclusions, are precisely those that demand sustained optimization effort. Following the logic of Xu et al. (Park et al., 2023), these stronger updates repeatedly drive the hidden state to align with the correct token embedding and simultaneously increase its $\ell_2$-norm.

Hence, high-norm tokens in a trained language model are not exclusively those that the model finds trivially easy; they also include tokens that were historically hard to learn and reasoning-critical tokens that received sustained corrective pressure during both pretraining and post-training. This provides a complementary theoretical perspective for why $\ell_2$-norm can serve as a proxy for reasoning intensity: the norm implicitly records the cumulative optimization force expended on each prediction, making it a particularly suitable signal in models exposed to reasoning-rich training regimes.

## C. Empirical Validation

Following the theoretical analysis, we empirically validate our hypothesis using the training data introduced in Section 2. Our experiments reveal a strong positive correlation between the

$$\ell_2$$

norm of hidden states and the activation strength of reasoning-associated features identified by the SAE. Specifically, we observe that both metrics exhibit highly aligned growth trends across layers, co-localize on similar reasoning-related tokens, and demonstrate significant linear correlation. Below, we present visualizations and quantitative results supporting this finding. What's more, we validate our fundamental assumptions in Appendix B here to make our theoretical analysis more solid.

### C.1. Validity of Theoretical Assumptions

We provide empirical evidence for the main assumptions used in our geometric analysis. Unless otherwise stated, the validation is conducted on Qwen3-4B and Llama3-8B, with statistics averaged over the last quarter of layers. Overall, the results indicate that these assumptions hold approximately but reliably enough to support the interpretations in the main text.

**Decoder orthonormality.** We first examine whether SAE decoder bases are approximately orthonormal. Empirically, the diagonal entries remain very close to 1.0 (mean $\approx 1.0000$, std $7.6 \times 10^{-7}$), while the average off-diagonal correlation is small ($\epsilon_{\text{orth}} = 0.0173 \ll 1$). These results indicate that the decoder basis is near-orthonormal, consistent with prior observations.

**Reconstruction quality.** We evaluate reconstruction fidelity in both magnitude and direction. The average reconstruction error is $8.7\%$, while the cosine similarity between the original and reconstructed representations remains high:

$$\cos(h_{\text{think}}, \hat{h}_{\text{think}}) = 0.9932, \qquad \cos(h_{\text{nothink}}, \hat{h}_{\text{nothink}}) = 0.9831.$$

This suggests that SAE reconstruction preserves the geometry of the representation space sufficiently well for our angle-based analysis.

**Semantic orthogonality and reasoning separation.** We next evaluate the relation between base semantics $h_{\text{base}}$ and reasoning signals $h_{\text{reasoning}}$. Their cosine similarity is small,

$$\cos(h_{\text{base}}, h_{\text{reasoning}}) = 0.0570,$$

indicating near-orthogonality and substantial disentanglement between semantic content and reasoning-related directions. Moreover, the norms of the two components are comparable ($\|h_{\text{base}}\| \approx 186.1$, $\|h_{\text{reasoning}}\| \approx 116.9$), showing that the reasoning component is not negligible noise but a substantial independent factor. This separation also becomes stronger in deeper layers, with cosine similarity decreasing from 0.3144 (Layer 5) to approximately 0.057 in later layers.

**Feature disentanglement.** Finally, we test whether the identified reasoning feature subset can faithfully reconstruct the reasoning direction. Using only the selected $k = 668$ reasoning features, we obtain a high directional recovery:

$$\cos\left(\sum_{i \in I_{\text{reasoning}}} \Delta z_i \cdot d_i, \; h_{\text{reasoning}}\right) = 0.9499.$$

Even under extreme sparsity (top-5 features only), the recovery remains high (0.8747). In contrast, base features exhibit near-zero correlation with the reasoning direction,

$$\cos\left(\sum_{i \in I_{\text{base}}} \Delta z_i \cdot d_i, \; h_{\text{reasoning}}\right) = 0.1443,$$

confirming that the reasoning signal is highly concentrated in the identified reasoning subset and well separated from base semantic features.

**Conclusion.** Taken together, these results support the validity of our theoretical assumptions: the relevant directions are approximately separable, reconstruction preserves geometry, and the identified reasoning features capture a substantial and distinct reasoning component. While the assumptions hold only approximately, they do so reliably enough for the geometric analysis in the main text.

### C.2. Growth Trends across Layers of LLMs

We observe a pronounced two-stage pattern in Qwen3 models (1.7B–32B) in Figure 10a, 11a, 12a, 13a, 14a, when inspecting SAE activations of reasoning-related features across layers. For each layer, we compute the mean activation over the top-5 reasoning features and normalize the resulting score by the corresponding mean activation under non-thinking responses to 0 to the maximum value. Across the early and middle layers, this normalized mean top-5 score remains relatively low and increases only gradually. However, in roughly the last quarter of the network, the score rises sharply, suggesting a late-layer regime in which the model more clearly separates and amplifies reasoning features.

For convenience, and to enable a consistent layer-wise alignment with our subsequent $\ell_2$-norm analysis (which uses the same prefix–suffix partition), we set a fixed boundary at $0.75L$ (i.e., three-quarters of the total depth $L$) as the stage transition in the figure. To verify that this choice reflects the data, we additionally ran a simple change-point detection on the layer-wise score curve (selecting the split that maximizes the pre/post difference in mean score, equivalently minimizing within-segment variance), which consistently places the transition in approximately the last quarter of layers. This "final $\sim25\%$" estimate is aggregated across multiple Qwen3 model sizes; we therefore use the unified $0.75L$ boundary for a consistent presentation across models.

This two-stage pattern is mirrored in the layer-wise $\ell_2$-norm of hidden states of thinking response. As shown in Figure 10b, 11b, 12b, 13b, 14b the $\ell_2$-norm remains relatively stable across the first three-quarters of the network and then exhibits a sharp increase in approximately the final 25% of layers—closely aligning with the surge observed in the normalized SAE reasoning-feature activations. This striking similarity suggests that the late-layer amplification of reasoning-related representations coincides with an overall increase in activation magnitude, reinforcing the view of a distinct late-stage reasoning regime in Qwen3 models.

### C.3. Token-Level Commonality of Reasoning Signals Across Depth

To complement our layer-wise analysis of reasoning feature separation, we further examine how the reasoning signal manifests at the *token level*. Across model families, we observe substantial overlap in the lexical items that co-occur with high activations of reasoning-related SAE features, suggesting that token-level indicators of reasoning are relatively stable and comparable across models. For clarity and consistency, we focus on Qwen3-8B and construct token word clouds for two representative depths (Layer 20 and Layer 30) in Figure 15, where each token is sized by its aggregated activation under top-5 reasoning SAE features.

**Reasoning-feature-associated tokens.** At Layer 20 (Figure 15a), the tokens most associated with high reasoning-feature activation are dominated by *domain and object words* that anchor the problem in a mathematical semantic space, such as

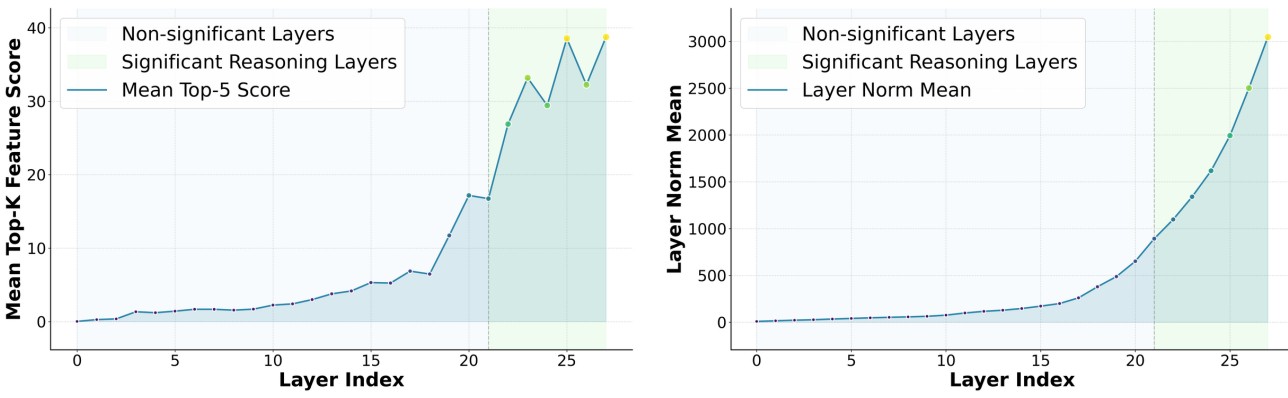

*(a)* Layer-wise normalized mean activation of the top-5 reasoning-related SAE features.

*(b)* Average $\ell_2$ norm of thinking responses.

*Figure 10.* Layer-wise trends of reasoning-related SAE feature activation and hidden-state $\ell_2$ norm for Qwen3-1.7B.

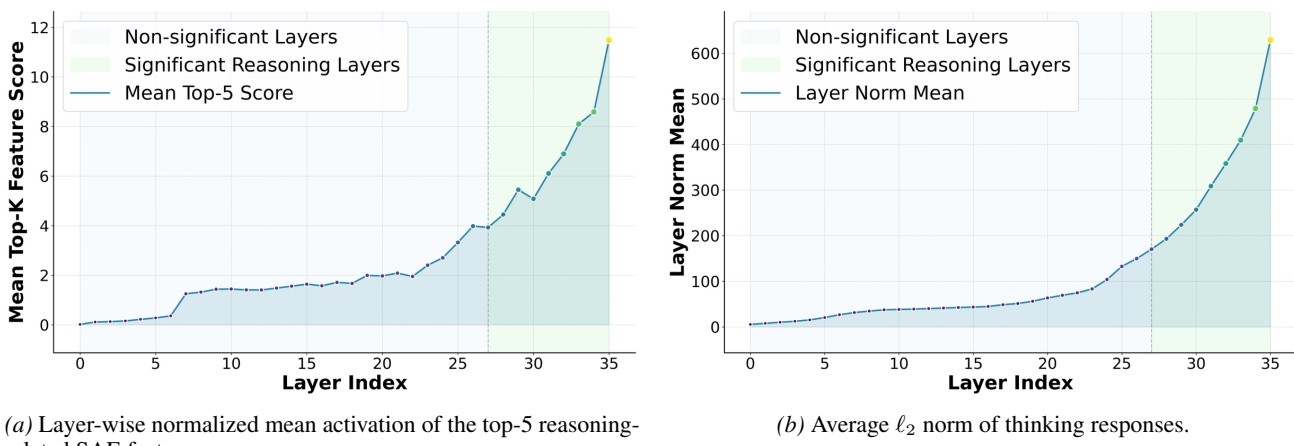

*(a)* Layer-wise normalized mean activation of the top-5 reasoning-related SAE features.

*(b)* Average $\ell_2$ norm of thinking responses.

*Figure 11.* Layer-wise trends of reasoning-related SAE feature activation and hidden-state $\ell_2$ norm for Qwen3-4B.

`math`, `polynomial`, `quadratic`, `law`, `extension`, `dimension`, `vector`, `product`, and `basis`. Alongside these content tokens, we also observe action-like tokens that indicate local progress in a solution attempt (e.g., `find`, `compute`, `check`, `verify`, `step`, `part`, `final`, `answer`). Overall, the Layer 20 cloud reflects a regime where the model is primarily organizing *what the problem is about*—identifying relevant mathematical objects and operators—while only partially expressing explicit reasoning control.

By Layer 30 (Figure 15b), the high-activation tokens shift markedly toward *discourse and control-flow markers* that structure multi-step reasoning trajectories. The most salient tokens include contrast and progression markers such as `but`, `so`, `thus`, `hence`, `then`, `also`, and `again`, as well as logical scaffolding tokens such as `if`, `any`, `all`, `not`, `or`, `given`, and `possible`. While some mathematical terms remain present (e.g., `rational`, `irrational`, `equation`), they appear more as *arguments embedded in an explicit proof-like structure* rather than as the dominant signal. This layer therefore emphasizes *how the model reasons*—introducing assumptions, branching over cases, drawing implications, and advancing toward conclusions—rather than merely grounding the problem in mathematical content.

**Depth-wise similarity and its link to late-layer separation.** Importantly, the qualitative transition from content-heavy tokens (Layer 20) to structure-heavy tokens (Layer 30) mirrors the two-stage pattern observed in our layer-wise reasoning feature score: earlier deep layers primarily support semantic grounding and local operations, whereas later layers increasingly implement explicit reasoning control that is more separable from non-thinking behavior. This provides an interpretable token-level view of why reasoning/non-reasoning separation becomes clearer in late layers: non-thinking responses can still contain domain content words, but they typically exhibit fewer discourse-level control markers that

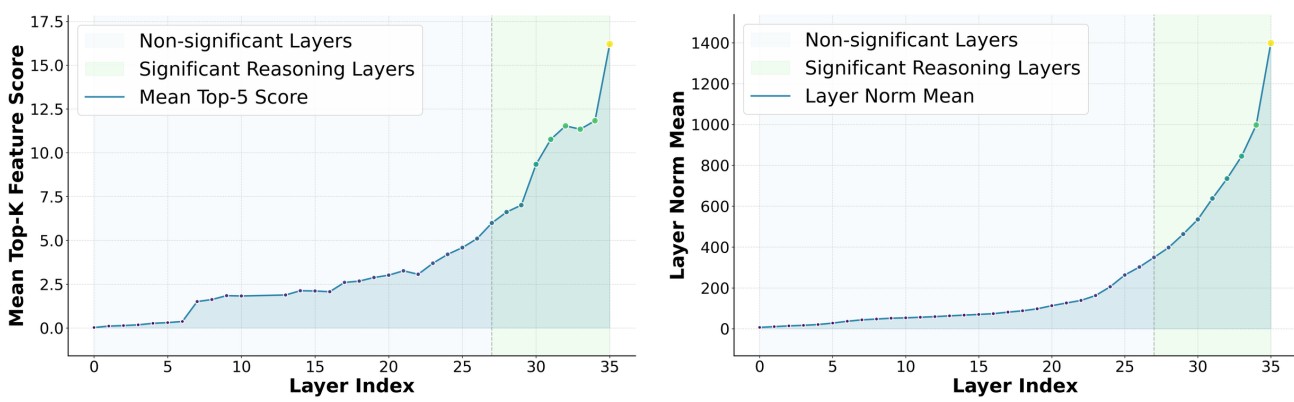

*(a)* Layer-wise normalized mean activation of the top-5 reasoning-related SAE features.

*(b)* Average $\ell_2$ norm of thinking responses.

*Figure 12.* Layer-wise trends of reasoning-related SAE feature activation and hidden-state $\ell_2$ norm for Qwen3-8B.

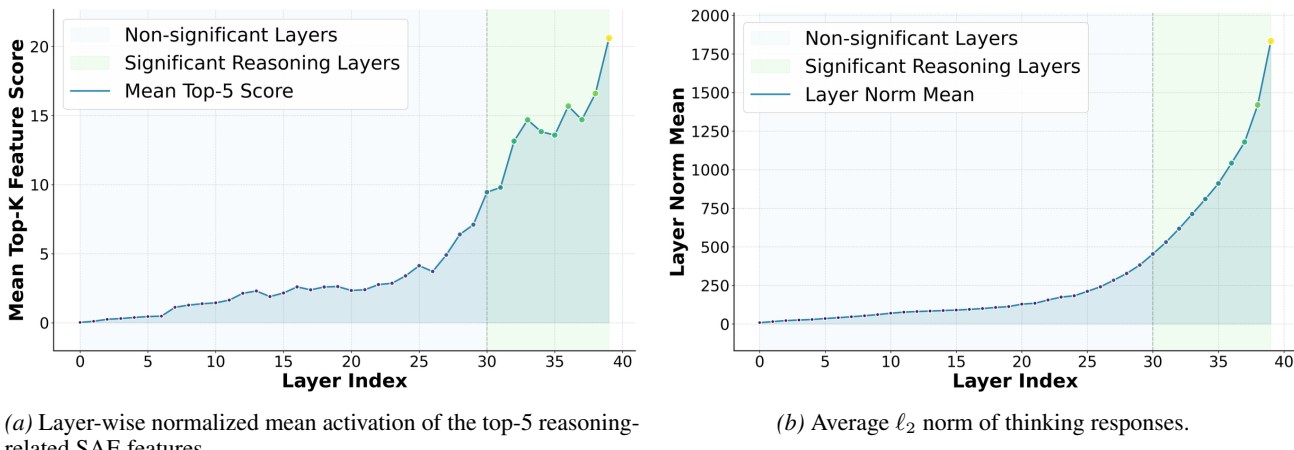

*(a)* Layer-wise normalized mean activation of the top-5 reasoning-related SAE features.

*(b)* Average $\ell_2$ norm of thinking responses.

*Figure 13.* Layer-wise trends of reasoning-related SAE feature activation and hidden-state $\ell_2$ norm for Qwen3-14B.

are characteristic of structured, step-by-step reasoning trajectories. Consequently, once late-layer representations begin to preferentially route through reasoning-related SAE features, the associated token distribution becomes more dominated by these structural markers, reinforcing the observed late-layer specialization.

$\ell_2$**-norm word clouds across depth.** We perform an analogous token-level inspection using the $\ell_2$ magnitude signal. For Qwen3-8B, we construct word clouds at Layer 20 and Layer 30 by collecting tokens associated with high $\ell_2$-norm activations (token size proportional to the aggregated $\ell_2$ score). Despite being derived from a different signal than SAE features, the $\ell_2$-based word clouds exhibit a depth-dependent shift that mirrors the SAE-based analysis: as depth increases, the salient tokens move away from content-heavy lexical items and toward discourse-level and control-flow markers that structure multi-step reasoning (e.g., function words and connectors such as `if`, `then`, `so`, `but`, `and`, `or`, `not`).

At Layer 20 (Figure 15c), high-$\ell_2$ tokens already include many generic connective and modality words (e.g., `can`, `may`, `might`, `should`, `still`), reflecting a regime in which the model is actively maintaining and updating intermediate state while remaining relatively broad and non-specialized. By Layer 30 (Figure 15d), the high-$\ell_2$ tokens are even more dominated by explicit structural scaffolding (e.g., `if`, `then`, `so`, `but`, `and`, `or`, `not`), consistent with the emergence of a late-layer "reasoning-control" phase where the model routes computation through long-horizon dependency management and step-to-step transitions. This qualitative progression aligns with our SAE finding that reasoning-related features become substantially more separable in the last portion of the network: the same late layers that show elevated reasoning-feature scores are also the layers where $\ell_2$ saliency concentrates on tokens that govern reasoning flow rather than topical content.

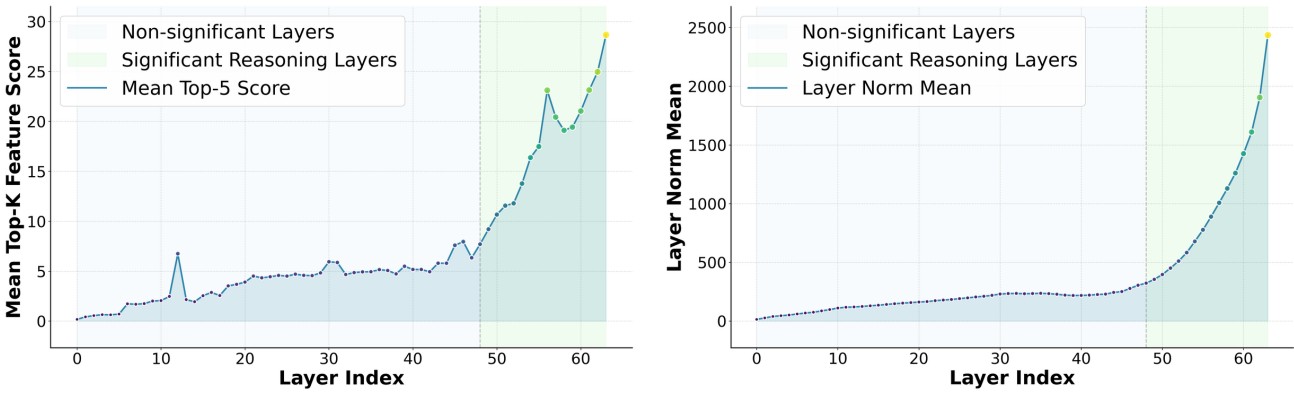

*(a)* Layer-wise normalized mean activation of the top-5 reasoning-related SAE features.

*(b)* Average $\ell_2$ norm of thinking responses.

*Figure 14.* Layer-wise trends of reasoning-related SAE feature activation and hidden-state $\ell_2$ norm for Qwen3-32B.

**Why $\ell_2$-norm and SAE word clouds are similar but not identical.** Importantly, we do *not* expect the $\ell_2$-based and SAE-based word clouds to match token-by-token, even though we previously showed that SAE activations are constrained by (and correlate with) the residual-stream magnitude measured by the $\ell_2$ norm. The reason is that the $\ell_2$ norm is a *scalar* summary of the overall representation magnitude at a token, whereas SAE reasoning features correspond to a *directional* decomposition that isolates specific subspaces aligned with reasoning. As a result, tokens can attain large $\ell_2$ magnitude for multiple reasons unrelated to our selected reasoning features (e.g., generic distribution shifts, attention-driven routing, or other high-variance subspaces), and conversely, a token can strongly activate a small set of reasoning directions without being among the largest-magnitude tokens under $\ell_2$. Therefore, $\ell_2$ saliency provides a coarse "energy-like" view of where the model expends representational capacity, while SAE reasoning features provide a more selective lens for *which* computational directions (in particular, reasoning-aligned ones) are being engaged. The partial overlap and systematic depth-wise co-variation between the two signals support our interpretation that late-layer reasoning separation is accompanied by a concentration of representation magnitude on tokens that implement reasoning control, even though the two token sets are not expected to coincide exactly.

## C.4. Quantitative Correlation Analysis

To validate the effectiveness of our proposed $\ell_2$ norm as a proxy for reasoning dynamics in large language models, we conduct a comprehensive correlation analysis across four representative models: Qwen3-1.7B, Qwen3-8B, Qwen3-14B, and LLaMA3-8B. Specifically, we compute the Spearman correlation between four layer-wise intrinsic metrics and two key metrics related to reasoning:

- Activations of Sparse Autoencoder (SAE) reasoning features (compute the correlation between all layers' metrics and the SAE activation of the corresponding layer, across all layers.)
- The entropy of the final model output (Wang et al., 2025) (computed correlation between output entropy and the last quarter of layers, which are closest to the prediction head and thus most reflective of the model's ultimate uncertainty).

The three baseline metrics are defined as follows:

- **Layer Hidden Entropy** ($H_h$):

$$H_h = -\sum_{i=1}^{d} p_i \log p_i, \quad \text{where } p_i = \frac{e^{h_i}}{\sum_j e^{h_j}}.$$

This measures the entropy of the softmax-normalized hidden state vector, capturing how uniformly information is distributed across semantic dimensions. A low $H_h$ indicates that the representation is concentrated in a few dominant features, which are often interpreted as strong, interpretable semantic signals, while high entropy suggests diffuse or ambiguous representations.

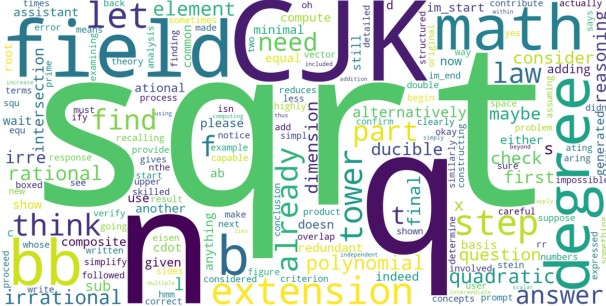

*(a)* High reasoning-feature activation tokens in Layer 20

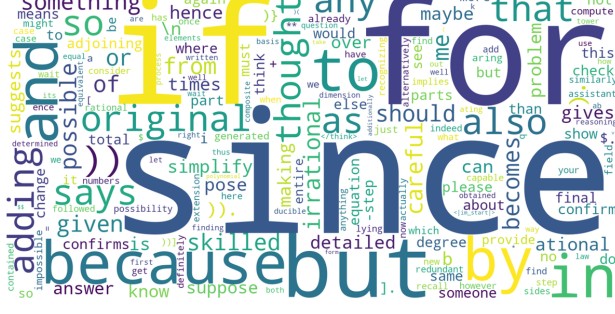

*(b)* High reasoning-feature activation tokens in Layer 30

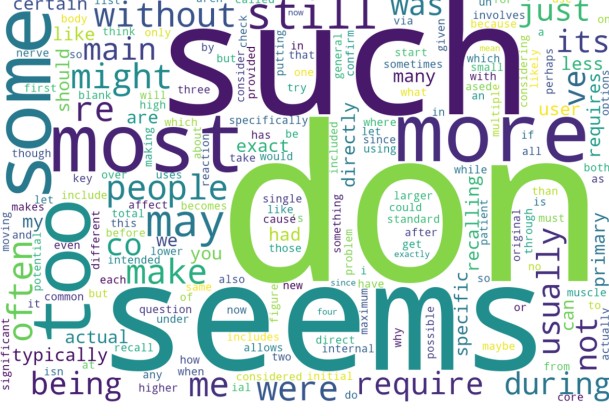

*(c)* High $\ell_2$ norm tokens in Layer 20

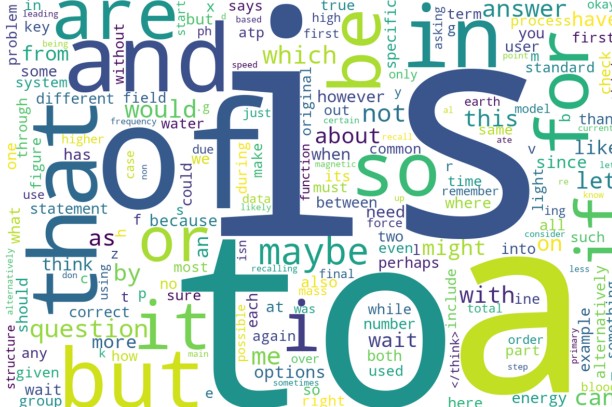

*(d)* High $\ell_2$ norm tokens in Layer 30

*Figure 15.* Word clouds of frequent tokens associated with (top) high reasoning-feature activation and (bottom) high $\ell_2$ norm at different layers of Qwen3-8B.

- **Layer Output Entropy** ($H_o$):

$$H_o = -\sum_{v=1}^{V} q_v \log q_v, \quad \text{where } q_v = \left[\texttt{softmax}(h \cdot W_{\text{lm\_head}})\right]_v.$$

This quantifies the uncertainty of the next-token prediction at an intermediate layer by projecting the hidden state through the language modeling head. It reflects the model's local confidence in its generative decision and is closely related to perplexity-based analyses in prior work (Wang et al., 2025).

- **Attention Entropy** ($H_a$):

$$H_a = -\sum_{j=1}^{n} a_j \log a_j, \quad \text{where } a_j = \left[\texttt{softmax}(\frac{QK^\top}{\sqrt{d_k}})\right]_j,$$

averaged over all attention heads. This metric characterizes the dispersion of attention weights: low entropy implies focused attention on a few tokens (suggesting selective reasoning), while high entropy indicates broad contextual integration. It has been widely used in Transformer interpretability studies (Vaswani et al., 2017).

As shown in Table 1, the $\ell_2$ norm exhibits consistently strong positive correlations with both SAE reasoning feature activations (ranging from 84.11 to 87.62) and final output entropy (52.10 to 66.65) across all models. In stark contrast, the alternative entropy-based metrics show relatively weak and unstable correlations with these targets suggesting they capture different aspects of model behavior.

This empirical evidence strongly supports two claims: (1) the theoretical connection between signal magnitude and reasoning saliency established in Appendix B is not only mathematically sound but also statistically robust across model scales and architectures; and (2) the $\ell_2$ norm serves as a reliable, scalable, and architecture-agnostic indicator of both

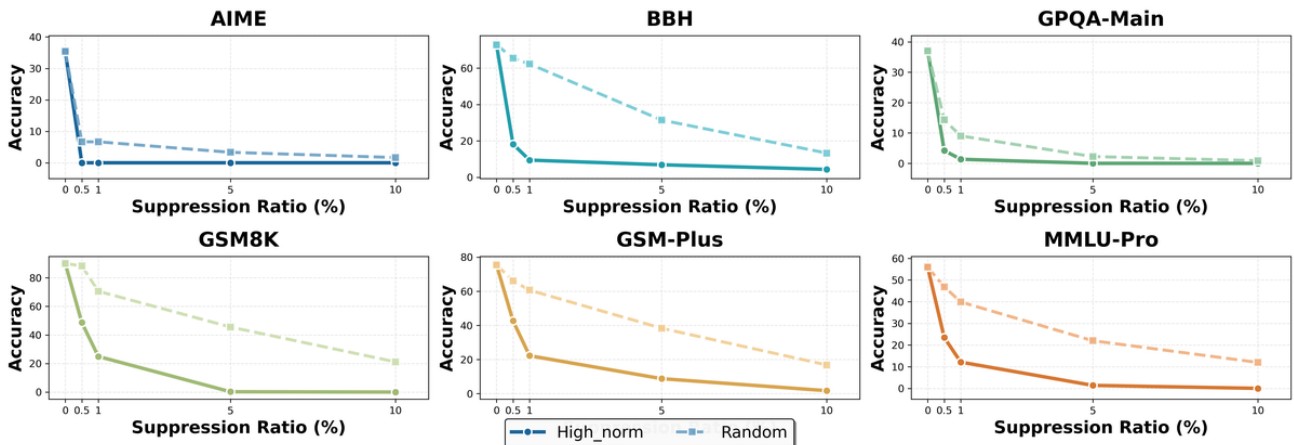

*Figure 16.* Causal intervention results on Qwen3-1.7B across reasoning benchmarks under different suppression ratios. We compare suppressing high-$\ell_2$-norm hidden states against Random Suppression.

internal reasoning activity (via SAEs) and output uncertainty. Consequently, it provides a principled foundation for layer-wise probing, dynamic routing, or intervention strategies aimed at enhancing controllability and interpretability in large language models.

## D. Causal Intervention via $\ell_2$-Norm-Based Suppression

To further validate the functional role of high-$\ell_2$-norm hidden states in reasoning, we perform causal intervention experiments across multiple models. Specifically, we suppress a fraction of hidden state dimensions with the largest $\ell_2$ norms at each layer and measure the resulting performance drop on a suite of reasoning benchmarks. As a control, we compare against random suppression of the same number of dimensions.

### D.1. Complete Suppresion Result

We show complete experiment results in Table 2 and Table 3.

Our suppression is applied *layer-wise* and *token-wise*. Specifically, for each layer $l$ in the designated suppression set (e.g., the last quarter of layers), we independently decide whether to suppress each token's hidden state at that layer.

For example, in `random` mode with suppression ratio $p = 0.1$, every token in each targeted layer is suppressed with probability $p = 10\%$, independently across tokens and layers. Consequently, each of the selected layers (e.g., all layers in the final 25% of the model) experiences an expected suppression rate of 10% of its active tokens—not a global 10% across the entire model. The reported suppression ratios always refer to the *per-layer expected or empirical token suppression rate*, not a cumulative or global fraction.

### D.2. Analysis

Figures 16–22 show the results for all the models, respectively. Across all model scales, suppressing high-$\ell_2$-norm components leads to a significantly sharper decline in performance compared to random suppression—especially under higher suppression ratios. This demonstrates that dimensions with large $\ell_2$ norms are not merely energetic but causally contribute to correct reasoning behavior. The consistency of this effect across architectures and benchmark suites reinforces our claim that the $\ell_2$ norm serves as a reliable indicator of reasoning-critical features in transformer hidden states.

What's more, we find that as model size increases, the model becomes increasingly robust to random token suppression. However, suppressing tokens with high-norm hidden states remains consistently detrimental across all model scales.

*Table 2.* Causal suppression results on Qwen3 model family under different intervention ratios. We compare random suppression with targeted suppression of high-$\ell_2$-norm hidden states (High-Norm). All values are accuracy scores (%).

| Ratio | Method | Mathematical | | | Complex | Scientific | Knowledge |
|---|---|---|---|---|---|---|---|
| | | AIME | GSM8K | GSM-Plus | BBH | GPQA-main | MMLU-Pro |
| *Qwen3-1.7B* | | | | | | | |
| 0% (Baseline) | – | 35.42 | 90.03 | 75.45 | 72.83 | 37.05 | 55.92 |
| 0.5% | High-Norm | 0.00 | 48.67 | 42.69 | 18.22 | 4.20 | 23.48 |
| | Random | 6.67 | 88.32 | 66.12 | 65.43 | 14.32 | 46.87 |
| 1% | High-Norm | 0.00 | 24.94 | 22.22 | 9.43 | 1.34 | 12.13 |
| | Random | 6.67 | 70.58 | 60.63 | 62.27 | 9.03 | 39.88 |
| 5% | High-Norm | 0.00 | 0.23 | 8.76 | 6.87 | 0.00 | 1.38 |
| | Random | 3.33 | 45.56 | 38.32 | 31.45 | 2.23 | 22.00 |
| 10% | High-Norm | 0.00 | 0.00 | 1.72 | 4.32 | 0.00 | 0.00 |
| | Random | 1.67 | 21.15 | 16.81 | 13.30 | 0.89 | 11.97 |
| *Qwen3-4B* | | | | | | | |
| 0% (Baseline) | – | 72.83 | 95.41 | 79.90 | 84.78 | 48.88 | 67.92 |
| 0.5% | High-Norm | 46.67 | 85.29 | 68.66 | 66.87 | 18.30 | 53.75 |
| | Random | 63.33 | 90.76 | 75.13 | 79.38 | 34.73 | 60.32 |
| 1% | High-Norm | 15.00 | 70.43 | 54.21 | 57.23 | 8.26 | 35.98 |
| | Random | 48.33 | 83.32 | 69.35 | 68.30 | 19.63 | 47.11 |
| 5% | High-Norm | 0.67 | 44.96 | 50.41 | 37.60 | 3.79 | 21.34 |
| | Random | 20.00 | 70.36 | 58.07 | 46.67 | 5.13 | 33.94 |
| 10% | High-Norm | 0.00 | 10.28 | 4.30 | 8.56 | 0.67 | 2.04 |
| | Random | 0.00 | 18.80 | 14.93 | 15.87 | 1.79 | 14.62 |
| *Qwen3-8B* | | | | | | | |
| 0% (Baseline) | – | 61.83 | 95.91 | 81.29 | 84.67 | 54.69 | 71.11 |
| 0.5% | High-Norm | 45.00 | 91.74 | 78.30 | 70.63 | 35.49 | 63.07 |
| | Random | 53.33 | 94.17 | 80.65 | 80.24 | 50.62 | 67.54 |
| 1% | High-Norm | 25.00 | 83.78 | 71.62 | 51.84 | 11.83 | 50.87 |
| | Random | 33.33 | 89.52 | 79.93 | 75.17 | 41.83 | 58.02 |
| 5% | High-Norm | 1.67 | 45.03 | 35.76 | 34.12 | 1.12 | 24.76 |
| | Random | 5.00 | 75.59 | 66.03 | 56.54 | 13.62 | 42.32 |
| 10% | High-Norm | 0.00 | 5.00 | 7.55 | 4.43 | 0.00 | 9.22 |
| | Random | 0.00 | 12.51 | 11.39 | 7.14 | 1.12 | 18.67 |
| *Qwen3-14B* | | | | | | | |
| 0% (Baseline) | – | 66.25 | 96.13 | 83.71 | 85.33 | 58.79 | 73.56 |
| 0.5% | High-Norm | 53.33 | 92.34 | 80.43 | 69.84 | 28.79 | 64.02 |
| | Random | 60.00 | 95.07 | 82.17 | 81.64 | 53.30 | 69.45 |
| 1% | High-Norm | 30.13 | 86.05 | 70.26 | 52.76 | 11.38 | 50.87 |
| | Random | 43.33 | 90.32 | 79.97 | 75.40 | 46.83 | 60.20 |
| 5% | High-Norm | 0.67 | 33.60 | 32.11 | 31.21 | 0.00 | 24.76 |
| | Random | 23.33 | 61.33 | 66.03 | 47.83 | 5.58 | 41.65 |
| 10% | High-Norm | 0.00 | 0.27 | 4.87 | 3.52 | 0.00 | 9.22 |
| | Random | 0.00 | 2.20 | 10.39 | 10.87 | 0.67 | 20.17 |
| *Qwen3-32B* | | | | | | | |
| 0% (Baseline) | – | 70.00 | 96.44 | 82.62 | 85.79 | 64.40 | 77.83 |
| 0.5% | High-Norm | 63.33 | 95.06 | 80.98 | 83.24 | 61.73 | 74.30 |
| | Random | 68.33 | 85.39 | 64.4 | 96.44 | 83.87 | 78.91 |
| 1% | High-Norm | 50.33 | 92.77 | 72.62 | 78.65 | 32.12 | 60.74 |
| | Random | 66.67 | 85.16 | 53.84 | 96.06 | 80.79 | 67.17 |
| 5% | High-Norm | 16.67 | 63.87 | 37.41 | 53.38 | 10.74 | 44.98 |
| | Random | 48.33 | 72.77 | 42.72 | 83.45 | 61.85 | 52.1 |
| 10% | High-Norm | 0.00 | 31.72 | 13.87 | 13.78 | 0.00 | 15.53 |
| | Random | 28.33 | 41.07 | 21.61 | 70.1 | 20.93 | 30.29 |

*Table 3.* Causal suppression results on LLaMA models under different intervention ratios. We compare random suppression with targeted suppression of high-$\ell_2$-norm hidden states (High-Norm). All values are accuracy scores (%).

| Ratio | Method | Mathematical | | | Complex | Scientific | Knowledge |
|---|---|---|---|---|---|---|---|
| | | AIME | GSM8K | GSM-Plus | BBH | GPQA-main | MMLU-Pro |
| | | | | *LLaMA-8B* | | | |
| 0% (Baseline) | – | 36.67 | 88.34 | 67.95 | 70.48 | 50.45 | 59.27 |
| 0.5% | High-Norm | 26.67 | 87.72 | 65.29 | 69.13 | 50.34 | 58.04 |
| | Random | 30.00 | 88.08 | 67.91 | 70.49 | 52.24 | 58.62 |
| 1% | High-Norm | 18.33 | 84.88 | 64.36 | 66.63 | 47.55 | 56.75 |
| | Random | 23.33 | 86.11 | 65.88 | 69.14 | 48.64 | 58.04 |
| 5% | High-Norm | 5.00 | 80.14 | 54.98 | 58.34 | 37.56 | 52.65 |
| | Random | 8.33 | 85.90 | 58.02 | 65.14 | 43.97 | 57.45 |
| 10% | High-Norm | 0.00 | 72.83 | 47.34 | 40.08 | 15.22 | 44.28 |
| | Random | 0.00 | 82.71 | 52.61 | 57.40 | 25.67 | 49.69 |
| | | | | *LLaMA-70B* | | | |
| 0% (Baseline) | – | 60.83 | 94.31 | 79.84 | 84.81 | 66.91 | 76.64 |
| 0.5% | High-Norm | 38.33 | 89.27 | 75.76 | 75.12 | 63.19 | 74.04 |
| | Random | 58.33 | 94.47 | 80.54 | 79.31 | 66.96 | 76.19 |
| 1% | High-Norm | 26.67 | 86.32 | 69.22 | 58.64 | 56.93 | 60.38 |
| | Random | 55.00 | 92.31 | 77.88 | 76.36 | 62.46 | 68.40 |
| 5% | High-Norm | 13.33 | 82.65 | 57.56 | 49.07 | 37.56 | 57.56 |
| | Random | 41.67 | 88.87 | 68.20 | 62.53 | 46.64 | 63.54 |
| 10% | High-Norm | 0.67 | 74.36 | 49.23 | 28.76 | 25.98 | 49.13 |
| | Random | 20.00 | 85.45 | 55.43 | 47.80 | 34.45 | 54.18 |

## D.3. Example Of Suppression

To illustrate the effect of our suppression experiments, we select a simple arithmetic problem as a case study to highlight the differences between the two suppression strategies (Example 1, 2).

Our main observation is that the $\ell_2$-norm criterion highlights tokens that function as *critical pivot points* in the model's reasoning: suppressing only a small fraction of such tokens can catastrophically break the chain of thought, while a larger amount of random suppression often leaves the reasoning trajectory intact (despite surface-level corruption such as code-switching or garbled characters).

> **Prompt 1**
>
> Solve this problem step by step:
> Three friends Alice, Bob, and Carol share some money. Alice gets 1/3 of the total, Bob gets 1/4 of the remaining, and Carol gets the rest which is $60. How much money did they have in total?

**Random-token suppression.** Even when we suppress more than 10% of tokens (271 / 2048), the model output may exhibit severe corruption (e.g., mixed languages, broken words, and noisy symbols). However, the underlying reasoning structure remains recoverable: the model still defines the total amount, computes the remaining share after Alice, applies Bob's fraction on the remainder, and solves for the total consistently. In other words, the *reasoning chain does not collapse*, and the model still reaches the correct total ($120).

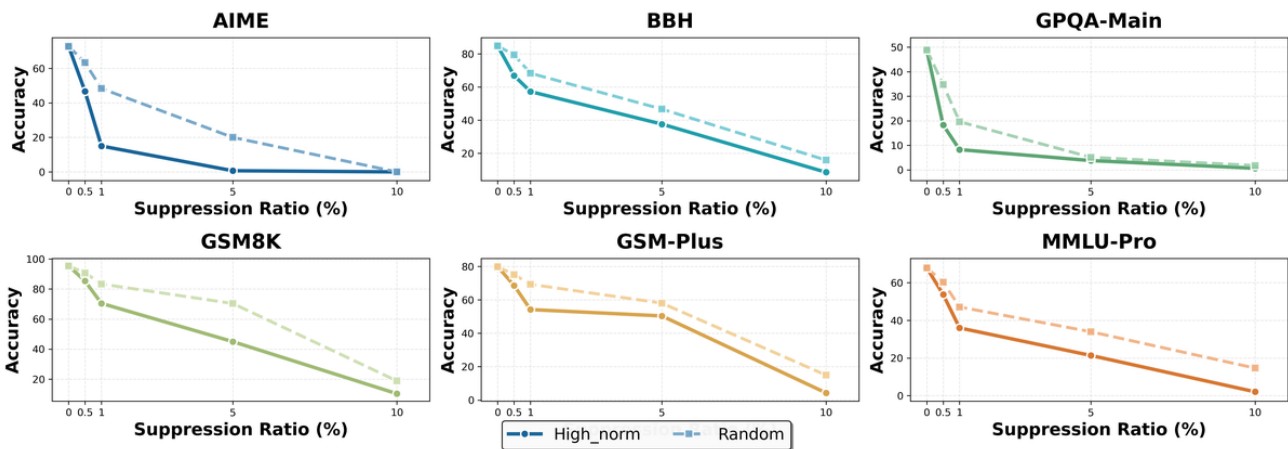

*Figure 17.* Causal intervention results on Qwen3-4B across reasoning benchmarks under different suppression ratios. We compare suppressing high-$\ell_2$-norm hidden states against Random Suppression.

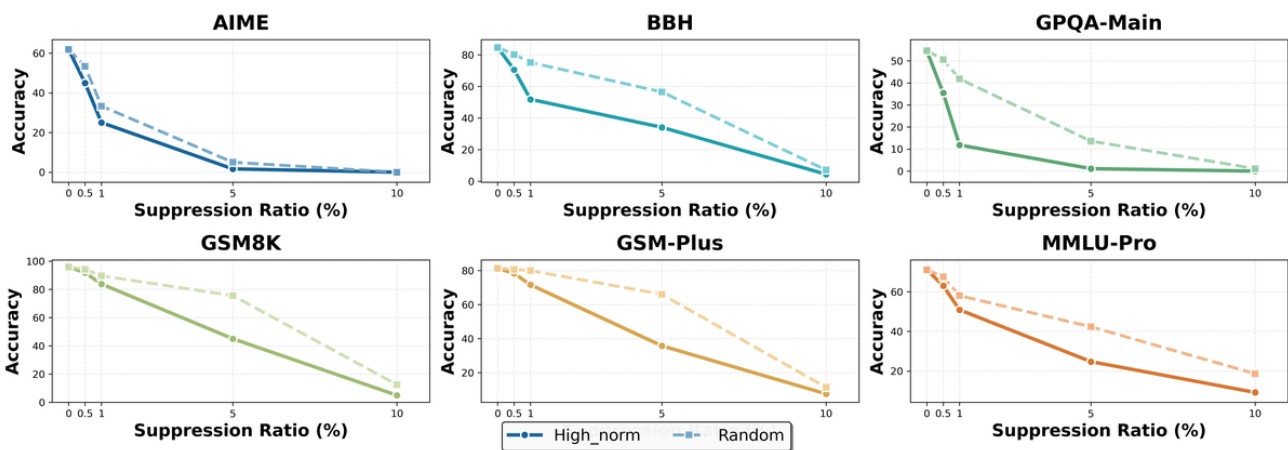

*Figure 18.* Causal intervention results on Qwen3-8B across reasoning benchmarks under different suppression ratios. We compare suppressing high-$\ell_2$-norm hidden states against Random Suppression.

---

**Example 1**

```
<think> O.K., let's try to solve this problem step by step.  So很  满  意, we need to
find out how much total mName they has.  Let me think.
First, the problem says that three friends|Alice, Bob, and congrat, Rita|share some
subsidi.  Alice gets 1/ of the total, then Bob gets 1/4 of the叫    我 remaining, and
Carol gets the rest, which is $60.
Hmm,叫我 is <non-EN>.  Let me rewrite it in my own words.
Let me.  Let's denoteagra aValue for the total amount of money they(withId.  Letś
<non-EN>.  Alice gets 1/3 ofROID. So, Alice's share is (1/3 of total).  Then, after
Alice takes要 不 然, there isERVED left.尼 斯, Bob 三 大 阶 段 1/4 of the remaining amount.
And whatever, the rest isgons,什么事情 $ SCRI..  60.夏
Soagra, let's set up the equations.  Let me denote the total amount as T可以更好.
所 以  他. Alice gets .cx(1/3)T. So, the remaining amount after Alice takes her share
is T - (1/3)T%/3.  Wait, Jinping, 1/也  是  很. So, the津 is T - ("<1/3)犁 =.  2/3 T?
Wait, <non-EN>.  Let me confirmPassword..  If you is 1/3 of芾, then theandal is 2/3.
So,agra,.  The remaining after Alice is 2/3 T.
Thenagra,,...
now, Bob getsrogen of the remaining.  The remaining is 2/3 T. Soagra,.  Bob gets
1/4 of that.  So, Bob痿 1/4 * (2/3 T"3.  Let..  Let me.  1/4 of 2/3.  Let.  (1/4 *
```

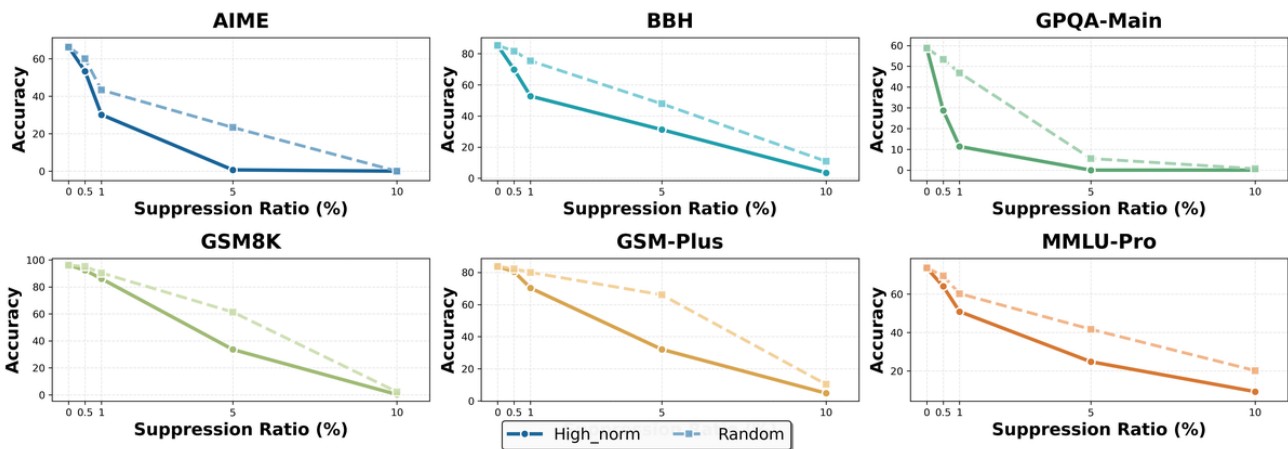

*Figure 19.* Causal intervention results on Qwen3-14B across reasoning benchmarks under different suppression ratios. We compare suppressing high-$\ell_2$-norm hidden states against Random Suppression.

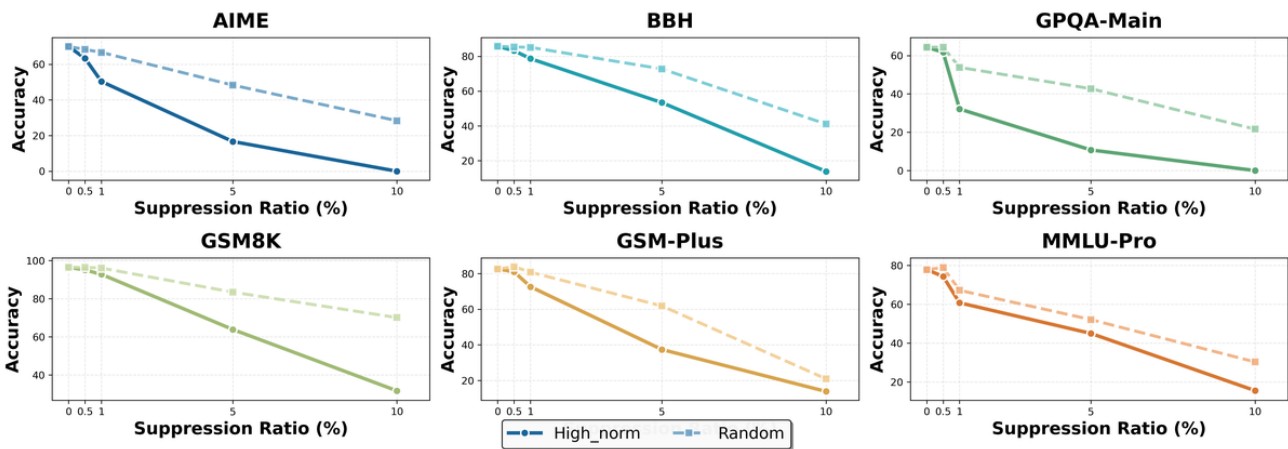

*Figure 20.* Causal intervention results on Qwen3-32B across reasoning benchmarks under different suppression ratios. We compare suppressing high-$\ell_2$-norm hidden states against Random Suppression.

```
2/3) = 2/12.  So, Bob's share is 2/12 of ...?  No, wait, apologies, let叫我 <non-EN>.
*---------------------
Let me do it againagra.
Total amount:  T.
Alice gets 1/3 * T. So, Alice is 1/3 T.
Remaining after Alice:  (T - 1/3 T) = 2/3 T. Now, Bob ...  of the remaining.  So,
Bob gets 1/4 of 2/3 T. So, 1/4 * 2/3 T = 2/12 T = 1/6 T.
So?, Bob getCode 犁 1/6 T. <ctrl>
Then, what is left after 1/3 T and 1/6 T? Let me calculate.
Total可以更好.  T.
三 大 阶 段.  Д Жaffe. 1/ Garrison T + 1/6 T = (1/3 + 1/6) T = (2-thirds津 1/6 T = 1/2
T? Wait, 1/+=(1/.  nameLabel.  1/3 1/6 is 2/6, which is 1/三 大 阶 段.  So, 1.  1/3/6
T = 1/ 3 + 1 <clock> 6 = 2/6agra.  1/3 + 1/6 = (2/6).  .,g. 2/6juana.  So, the
remaining is 1/2 T.
But the problem says that Carol gets piss which is $60.  Og, so 1/2 T = 60:[[.  So,
T = 6en..6 2.  So, ; Wait, 1党的十 2 is 60.  So,可以更好奥但不限 2T = 6, soofi, T = 60 *
2 = 120?  Wait,elleicht., Let me check5.  If T 120, then:
Alice gets也是很 1/3 of 120 = 40.
Wait, but let me check if there's another way to approach it.  Maybeartner.
urved.  Let's say et.  Total is三大阶段.
```

```
Alice, 1/3.  So GDP tslib..  1 third.  Bale, 1/3.
Bale.  urved.  1/3 of total.  Sourved 2/3 remains.
Then, Bob gets 1/4 of the 2/3.  So, 1/4 * 2/tractorackson 1/_plural.
So, 1/3 + 1/6三大阶段.='  1/?  + 1/6 = 2/6 + 1/1/6 = 3/6 = 1/2.  So, 1/2 of the total
is left人大常委.
.7, 1 2 T = 60.  So, T / 2 = 60,)?;
..  T = 120.
</think>
```

**High-$\ell_2$-norm suppression.** In contrast, when we suppress only 4.5% of tokens (91 / 2048) selected by high $\ell_2$-norm, the model's reasoning fails abruptly. The output shows a structural breakdown: after partially restating the problem and attempting to introduce a variable, the generation degenerates into fragmented text and repetitive placeholders, and the step-by-step derivation is never completed. This indicates that the suppressed set is not arbitrary; rather, it disproportionately contains tokens that the model relies on to maintain coherent intermediate state updates.

**Example 2**

```
<think> Okay, let's see.  So there are 3 friends:  Alice, Bob, and Cinderella?
Wait|was 3 friends ACLU?";
RD,.wait Algebra,.wait,...
Wait, 3叫我,.wait,...
NoSuch, März,agra...
Wait,', the problem is about threesome,| Trio,
!";
Alice, Bob, cdecl.
Alright, so the problem says:''  Three的女孩  >,...
叫我 aValue,...
,
Alice gets 1/3 of the in overall, $
%");
$
Bob gets ''(1/4:[[remaining],|
and|
%</60.
How much什麼;
total?
Okay,�budget let's try to parse this step by step.
First, let me denote theennifer total amount of money as T. So, T is what we|-------
need to sof.  Now,,...
%");
Alice gets 1/3 of ?");
So, Alice's part is (1/3)*/
T. So, Alice has (1/3)*/ T.
Then,avier要不然也是很三大阶段isorasmine 3 Ы,edi!";
the remaining after3×</ is what
$
$
$
$ ...  (degeneration) ...
```

# E. Evaluation Setting

### E.1. Evaluation Framework

We conduct all evaluations using the `lm-evaluation-harness` (lm-evaluation) framework (Gao et al., 2024), which provides standardized task implementations, prompt templates, answer extraction, and metric aggregation. All experiments are executed via the `vLLM` backend to ensure efficient batched generation and consistent decoding behavior across models.

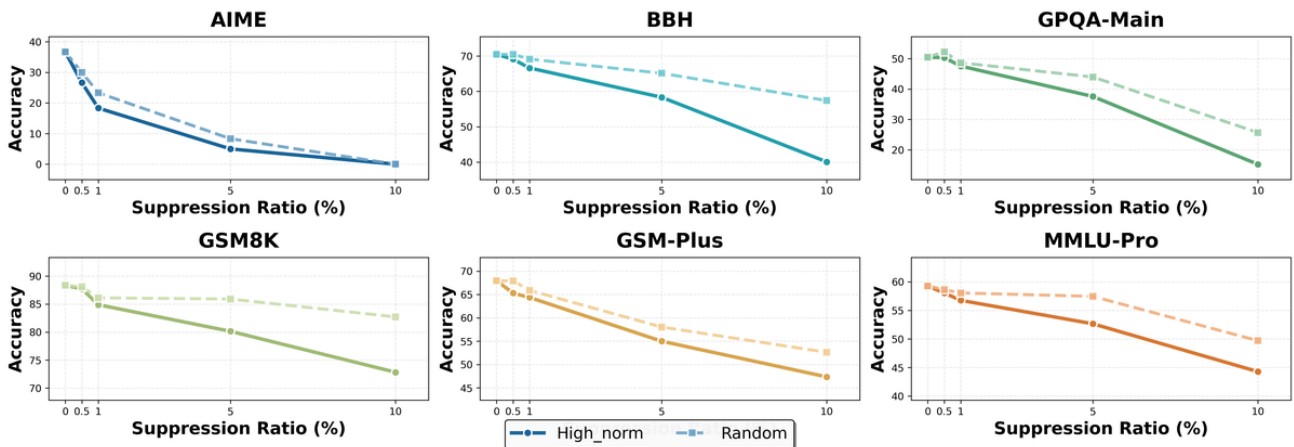

*Figure 21.* Causal intervention results on R1-Distill-Llama-8B across reasoning benchmarks under different suppression ratios. We compare suppressing high-$\ell_2$-norm hidden states against Random Suppression.

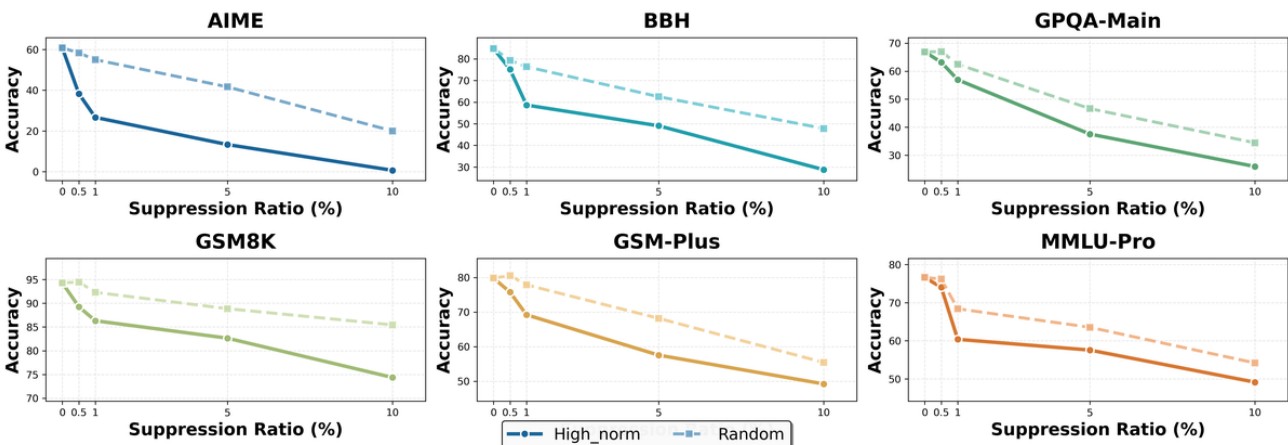

*Figure 22.* Causal intervention results on R1-Distill-Llama-70B across reasoning benchmarks under different suppression ratios. We compare suppressing high-$\ell_2$-norm hidden states against Random Suppression.

### E.2. Benchmarks

To cover both mathematical reasoning and general-purpose reasoning/knowledge, we evaluate on a diverse set of public benchmarks:

- Competition-style and grade-school math reasoning (GSM8K (Cobbe et al., 2021), GSM-Plus (Li et al., 2024), AIME 2024, AIME 2025 (Zhang et al., 2023)), and
- Broader reasoning and knowledge-intensive benchmarks (BBH (Suzgun et al., 2023), MMLU-Pro (Wang et al., 2024)). For each benchmark, we use the task configuration specified in lm-evaluation-harness (notably the CoT zero-shot variants where applicable).

**Benchmark descriptions and targeted abilities.** Below, we summarize each benchmark, provide the corresponding Hugging Face dataset link, and describe the primary capability being measured.

### E.3. Metric Definition and Subset Aggregation

**Per-task accuracy.** For each task (or subtask) with $N$ evaluation instances, we compute accuracy as:

$$\text{Acc} = \frac{1}{N} \sum_{i=1}^{N} \mathbb{I}\left[\hat{y}_i = y_i\right], \tag{45}$$

| Benchmark | HuggingFace link | Dataset summary | Primary abilities tested |
|---|---|---|---|
| BBH | lukaemon/bbh | Big-Bench Hard (BBH) is a curated subset of difficult BIG-bench tasks spanning logical, symbolic, linguistic, and algorithmic problem types. | Multi-step reasoning; compositional generalization; symbolic manipulation; instruction following. |
| GSM-Plus | qintongli/GSM-Plus | A GSM-style arithmetic word-problem benchmark designed to increase difficulty and reduce shortcut biases via more diverse (often adversarial) problem constructions. | Robust math word-problem solving; numerical reasoning; reduced reliance on superficial heuristics. |
| GSM8K | openai/gsm8k | A widely used grade-school math word-problem dataset with short (often integer) answers; commonly evaluated under chain-of-thought prompting. | Multi-step arithmetic reasoning; mapping text to equations; consistent computation. |
| AIME 2024 | Maxwell-Jia/AIME_2024 | Competition mathematics problems from AIME 2024, typically involving algebra, counting, geometry, and creative reasoning; answers are usually integers (often 0–999). | Advanced mathematical reasoning; decomposition; symbolic manipulation; combinatorics/geometry. |
| AIME 2025 | math-ai/aime25 | Competition mathematics problems from AIME 2025 with similar style and difficulty characteristics as AIME 2024. | Advanced mathematical reasoning; generalization to unseen contest problems; multi-step symbolic reasoning. |
| MMLU-Pro | TIGER-Lab/MMLU-Pro | An enhanced variant of MMLU with increased difficulty and stronger contamination resistance, spanning many academic and professional subjects. | Broad knowledge + reasoning; domain generalization; reading comprehension; selecting among close options. |

*Table 4.* Benchmarks and their Hugging Face dataset pages. The table shows relative dataset paths, each linking to the full Hugging Face URL.

where $y_i$ is the ground-truth answer, $\hat{y}_i$ is the extracted model answer, and $\mathbb{I}[\cdot]$ is the indicator function.

**weight_by_size aggregation for benchmarks with subsets.** Some benchmarks (e.g., BBH, MMLU-Pro) consist of multiple subsets $\{s\}_{s=1}^{S}$ with different sizes $N_s$. Following lm-evaluation-harness's weight_by_size strategy, we aggregate subset accuracies by weighting each subset by its number of examples:

$$\text{Acc}_{\text{weighted}} = \sum_{s=1}^{S} w_s \cdot \text{Acc}_s, \quad w_s = \frac{N_s}{\sum_{j=1}^{S} N_j}, \tag{46}$$

where $\text{Acc}_s$ is computed using Eq. (45) on subset $s$.

### E.4. Inference Configuration

We use temperature=0.6, top_p=0.95, top_k=20, repetition_penalty=1.05, min_p=0, and max_gen_toks=16384. These values follow the officially recommended decoding configuration for the Qwen3 series *think* models. Since the DeepSeek-R1-Distill-LLaMA family does not provide an officially recommended inference setting for reasoning-focused decoding, we apply the same decoding configuration to ensure consistent evaluation across model families.

**Output format constraint and answer extraction.** For all benchmarks, we require the model to output the final answer in the normalized format:

```
The answer is <answer>
```

This constraint simplifies deterministic rule-based extraction of $\hat{y}$ from model generations and reduces ambiguity caused by free-form outputs. For methods like suppression that may impair instruction-following fidelity, we adopt a more lenient extraction strategy that relies on identifying numerical values or inherent output patterns (e.g., explicit answer delimiters or canonical formats) to determine $\hat{y}$.

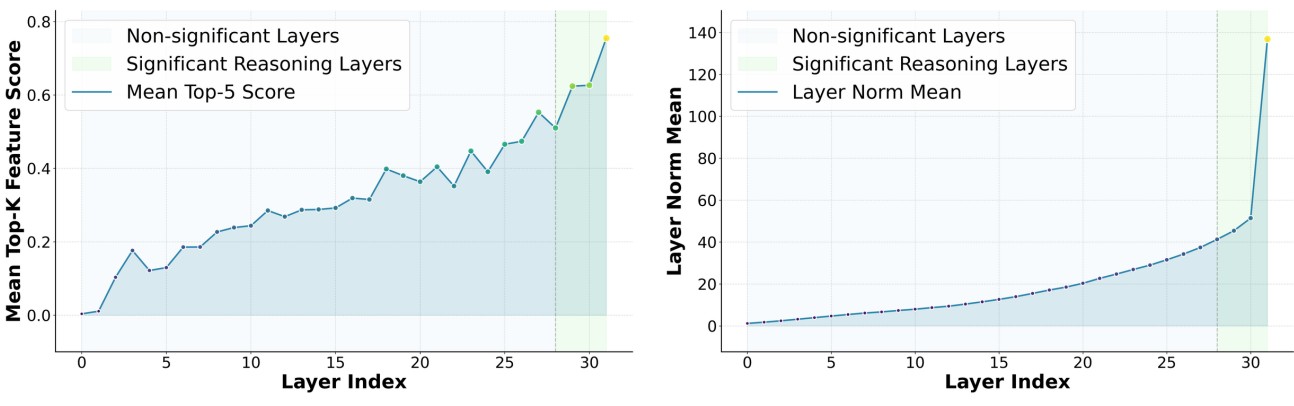

*(a)* Layer-wise normalized mean activation of the top-5 reasoning-related SAE features.

*(b)* Average $\ell_2$ norm of thinking responses.

*Figure 23.* Layer-wise trends of reasoning-related SAE feature activation and hidden-state $\ell_2$ norm for Deepseek-R1-Distill-LLama-8B.

### E.5. Repeated Runs and Randomness Control

To mitigate evaluation variance induced by stochastic decoding, each benchmark evaluation is repeated 8 times, and we report the mean accuracy across runs:

$$\overline{\text{Acc}} = \frac{1}{8} \sum_{r=1}^{8} \text{Acc}^{(r)}. \tag{47}$$

This protocol is applied consistently to all tasks. Due to space constraints, we were unable to report standard deviations for all benchmarks in the tables. Nevertheless, the standard deviations are consistently small across evaluations: for the AIME suite—which contains relatively few questions—the standard deviation is below 5%; for GPQA, it is under 2%; and for all other benchmarks, it is less than 0.5%. Crucially, the magnitude of the performance gains we report (or the degradation caused by suppression) is substantially larger than the corresponding standard deviations. This indicates that our experimental results are statistically reliable and our conclusions are well-supported.

### E.6. Rationale for Benchmark Coverage

Beyond standard mathematical reasoning benchmarks used to validate step-by-step problem solving (GSM8K, GSM-Plus, AIME 2024/2025), we also include BBH and MMLU-Pro to probe broader reasoning competence and knowledge-intensive generalization. This combined suite provides a more comprehensive assessment of our method and experimental outcomes across both math-centric and general reasoning settings.

## F. Model Differences between Qwen3 Series and R1-Distilled-Llama3 Series

We strictly follow the setup in Appendix C to plot the *Layer-wise Reasoning Feature Score* (Figure 23a, 24a) and the layer-wise mean $\ell_2$ norm under thinking responses (Figure 23b, 24b). For each layer, we select the top-5 highest-scoring reasoning features, average their SAE activations, and normalize by the corresponding mean activation of these features on *non-thinking* responses at the same layer, yielding a layer-comparable measure of relative reasoning-representation strength.

With this metric (identical to Appendix C), we observe a systematic difference in where *reasoning vs. non-reasoning representation separation* emerges across model families: Qwen models enter a regime of strongly increased reasoning-feature activation earlier in depth, whereas DeepSeek-R1-Distilled-Llama models often do not exhibit a clear separation until only the final few layers.

Importantly, the R1-Distill-Llama-family results still exhibit the same qualitative *SAE–norm correspondence* observed for Qwen: layers with increased reasoning-feature scores are also the layers where the mean $\ell_2$ norm under thinking responses rises, indicating that stronger reasoning-feature engagement coincides with increased representational "energy" (Figure 23a–24b). Moreover, token-level $\ell_2$-norm word clouds in late layers (Layer 79 in Figure 25b) show that

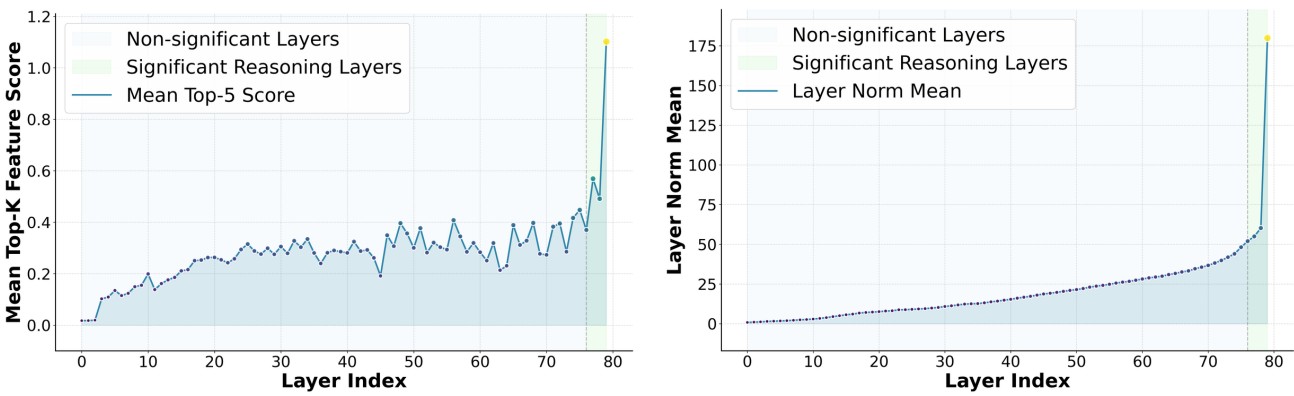

*(a)* Layer-wise normalized mean activation of the top-5 reasoning-related SAE features.

*(b)* Average $\ell_2$ norm of thinking responses.

*Figure 24.* Layer-wise trends of reasoning-related SAE feature activation and hidden-state $\ell_2$ norm for Deepseek-R1-Distill-LLama-70B.

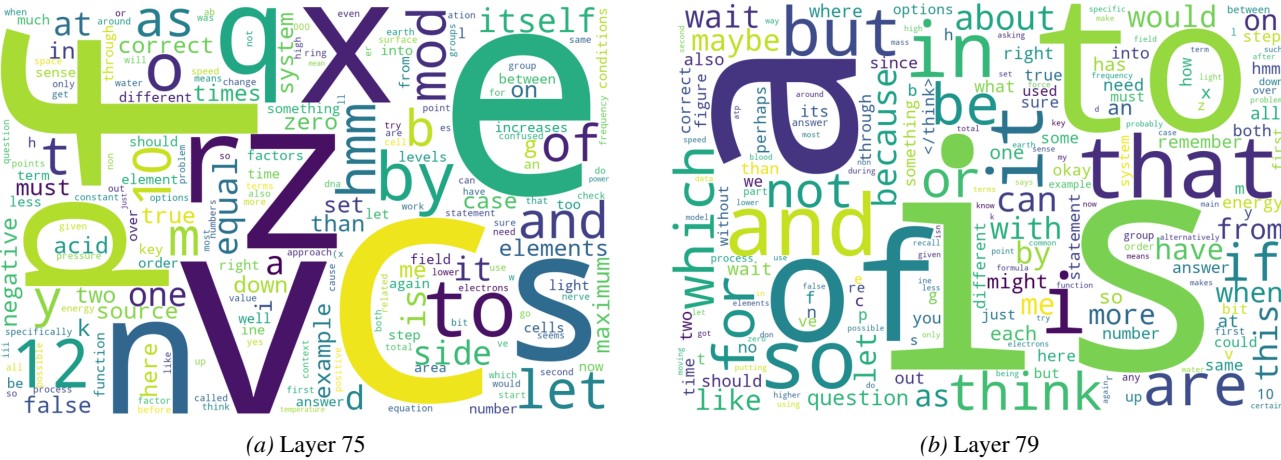

*(a)* Layer 75

*(b)* Layer 79

*Figure 25.* Frequent tokens most associated with high $\ell_2$ Norm in R1-Distill-Llama-70B. Both layers show reasoning-related tokens.

the model does begin to exhibit sensitivity to reasoning-style control tokens, with salient discourse and logical scaffolding markers such as if, then, so, but, and, not becoming increasingly prominent, consistent with the emergence of more structured multi-step reasoning behavior near the end of the LLM.

At the same time, this emergence is clearly *back-loaded*. Even for Llama-70B, the high-norm token distribution around Layer 75 (Figure 25a) remains dominated by surface-form artifacts and domain symbols (e.g., math variables and notation-like tokens), rather than a robust set of reasoning-control markers. Only in the final layers do discourse-level connectors and logical operators become strongly emphasized, aligning with the late-stage increase in our reasoning feature score. This delayed transition suggests that, compared to Qwen, Llama relies more heavily on terminal-layer transformations to express reasoning-like behavior, and that its internal representations allocate less depth (and thus less representational bandwidth) to forming a stable, separable reasoning subspace which aligns with the conclusions of existing works (Qian et al., 2025; Gandhi et al., 2025).

We attribute this discrepancy to the prior *imprint of reasoning processes during pretraining* in the base policy, rather than a superficial behavior that post-training can arbitrarily rewrite. Gandhi et al. propose and empirically demonstrate that whether RL/SFT can elicit and amplify structured cognitive behaviors such as verification, backtracking, subgoal setting, and backward chaining depends critically on whether these behaviors already appear with non-trivial frequency in the base model policy. (Gandhi et al., 2025) Models lacking such *behavioral primitives* are more prone to plateau under matched RL settings, whereas shifting the pretraining distribution (e.g., filtering OpenWebMath to amplify these behaviors) can substantially change the subsequent self-improvement trajectory (Gandhi et al., 2025).

Beyond emphasizing the collection and filtering of high-quality mathematical and problem-solving data, prior work also suggests that explicitly training on *process-oriented* reasoning signals and structured reasoning formats (e.g., chain-of-thought, programmatic reasoning, and tool-integrated reasoning) tightly couples reasoning capability to the underlying training distribution—in particular, to how often and in what form reasoning processes appear in the data (Shao et al., 2024).

Accordingly, we interpret the observed difference in the depth at which separation appears as a stable inductive bias shaped by pretraining in parameter space: when the base policy has internalized richer reasoning-process primitives, reasoning-related features are more likely to form a separable and persistently activatable subspace in deep (but not only terminal) layers; conversely, when such primitives are scarce during pretraining, post-training (including distillation-based alignment) is less likely to fundamentally reorganize internal representations, and may instead yield more localized, output-adjacent manifestations of reasoning behavior.

Moreover, the structure of reasoning demonstrations (not merely answer correctness) can significantly affect what the model learns, providing a mechanistic rationale for why earlier injection of reasoning processes is more likely to reshape global representational organization. In particular, evidence indicates that models can learn reasoning primarily from the structural scaffolding of demonstrations rather than their specific content, implying that the organization of reasoning trajectories can directly drive parameter updates and influence how reasoning is implemented (Li et al., 2025a).

Taken together, we interpret the different separation depths between Qwen and Deepseek-R1-Distill-Llama (distilled primarily during post-training and retaining substantial traces of Llama pretraining) as reflecting different levels of exposure to structured reasoning processes during pretraining, leading to fundamentally different speeds and extents with which a dedicated reasoning subspace is formed in deep layers.

This account also helps explain the difference in intervention results of other experiments between Qwen and Llama Series Models. Because the R1-Distilled-Llama-8B family only develops a clearly separable reasoning subspace very late in the network, interventions have less representational "room" to act on and are therefore less effective than for Qwen3, whose reasoning-related subspace is established earlier and more broadly across deep layers.

# G. Adaptive Layer-wise Reasoning Recursion

We evaluate our Adaptive Layer-wise Reasoning Recursion (ALRR, also referred to as RR for brevity) method across seven state-of-the-art thinking models spanning two major model families: Qwen (1.7B to 32B parameters) and R1-Distill-LLama (8B and 70B parameters). Our comprehensive evaluation set is detailed described in E and our detailed experiment results are in Table 5. Figure 26 provides a holistic view of the performance improvements across all models and benchmarks, while Figure 27 illustrates the joint relationship between model size, baseline benchmark performance, and the performance gains achieved by the proposed method across benchmarks. The detailed performance of each model is shown in the Figure 28.

## G.1. Universal Effectiveness Across Model Families

Our results demonstrate the universal applicability of the RR method across diverse model architectures and scales. As shown in Figure 26, **every model achieves higher average performance after applying RR**, with consistent gains across both model families and sizes. This validates our core hypothesis: by identifying high-$\ell_2$-norm layers—where activation spikes signal critical reasoning moments—and recursively refining representations at these points, we can systematically enhance problem-solving capabilities regardless of base architecture.

The Qwen series exhibits particularly great improvements. For instance, Qwen3-4B improves from 60.0% to 70.0% (+10.0%) on AIME24, while Qwen3-14B advances from 65.8% to 73.3% (+7.5%). Smaller models like Qwen3-1.7B also benefit substantially (e.g., +8.3% on AIME2024), suggesting that when reasoning capacity is limited, targeted recursion at activation peaks provides disproportionate value. In contrast, the LLaMA family shows more modest but notably **uniform** gains: LLaMA-8B and LLaMA-70B both improve by +2.8%. This difference aligns with our earlier analysis in Appendix F: LLaMA models develop a separable reasoning subspace only very late in the network, leaving less representational "room" for interventions to act upon, whereas Qwen establishes reasoning-relevant structure earlier and more broadly across deep layers—making it more responsive to layer-wise refinement.

*Table 5.* Performance comparison between Base models and Loop-enhanced variants across reasoning benchmarks. Accuracy (%) is reported, with relative improvement over Base shown in Performance Gain (%).

| Model | Setting | Mathematical | | | | Complex | Scientific | Knowledge |
|---|---|---|---|---|---|---|---|---|
| | | AIME24 | AIME25 | GSM8K | GSM-Plus | BBH | GPQA-main | MMLU-Pro |
| Qwen3-1.7B | Base | 40.00 | 30.83 | 90.03 | 75.45 | 72.83 | 37.05 | 55.92 |
| | Recur | **48.33** | **40.00** | 90.03 | **76.28** | **74.57** | **40.40** | **58.63** |
| *Performance Gain (%)* | | *+20.83* | *+29.76* | *+0.00* | *+1.09* | *+2.41* | *+9.10* | *+4.82* |
| Qwen3-4B | Base | 60.00 | 56.66 | 95.41 | 78.90 | 84.78 | 50.22 | 67.92 |
| | Recur | **70.00** | **60.00** | **95.53** | **79.88** | **87.39** | **51.55** | **71.66** |
| *Performance Gain (%)* | | *+16.67* | *+5.97* | *+0.13* | *+1.01* | *+3.10* | *+2.66* | *+5.50* |
| Qwen3-8B | Base | 63.67 | 60.00 | 95.91 | 81.29 | 84.67 | 54.69 | 71.11 |
| | Recur | **71.66** | **61.66** | **96.36** | **82.11** | **87.41** | **57.37** | **75.24** |
| *Performance Gain (%)* | | *+12.57* | *+2.77* | *+0.47* | *+1.02* | *+3.26* | *+4.86* | *+5.94* |
| Qwen3-14B | Base | 65.83 | 67.66 | 96.13 | 83.71 | 85.33 | 58.71 | 73.56 |
| | Recur | **73.33** | **71.66** | **96.51** | **84.31** | **88.01** | **62.95** | **77.79** |
| *Performance Gain (%)* | | *+11.43* | *+5.91* | *+0.39* | *+0.71* | *+3.17* | *+7.17* | *+5.77* |
| Qwen3-32B | Base | 70.83 | 69.17 | **96.44** | 82.62 | 85.79 | 64.40 | 77.83 |
| | Recur | **75.00** | **73.33** | 96.36 | **84.98** | **88.11** | **65.96** | **79.72** |
| *Performance Gain (%)* | | *+5.81* | *+6.00* | *-0.08* | *+2.80* | *+2.75* | *+2.46* | *+2.42* |
| LLaMA-8B | Base | 40.00 | 33.33 | 89.23 | 67.95 | 70.48 | 50.45 | 59.27 |
| | Recur | **47.50** | **36.67** | **90.78** | **68.25** | **72.53** | **52.23** | **60.47** |
| *Performance Gain (%)* | | *+18.75* | *+10.04* | *+1.73* | *+0.44* | *+2.90* | *+3.63* | *+2.03* |
| LLaMA-70B | Base | 63.33 | 55.83 | 94.31 | 79.84 | 84.81 | 67.19 | 76.64 |
| | Recur | **66.67** | **58.33** | **95.73** | **80.72** | **86.41** | **68.30** | **79.15** |
| *Performance Gain (%)* | | *+5.25* | *+4.44* | *+1.50* | *+1.10* | *+1.88* | *+1.66* | *+3.27* |

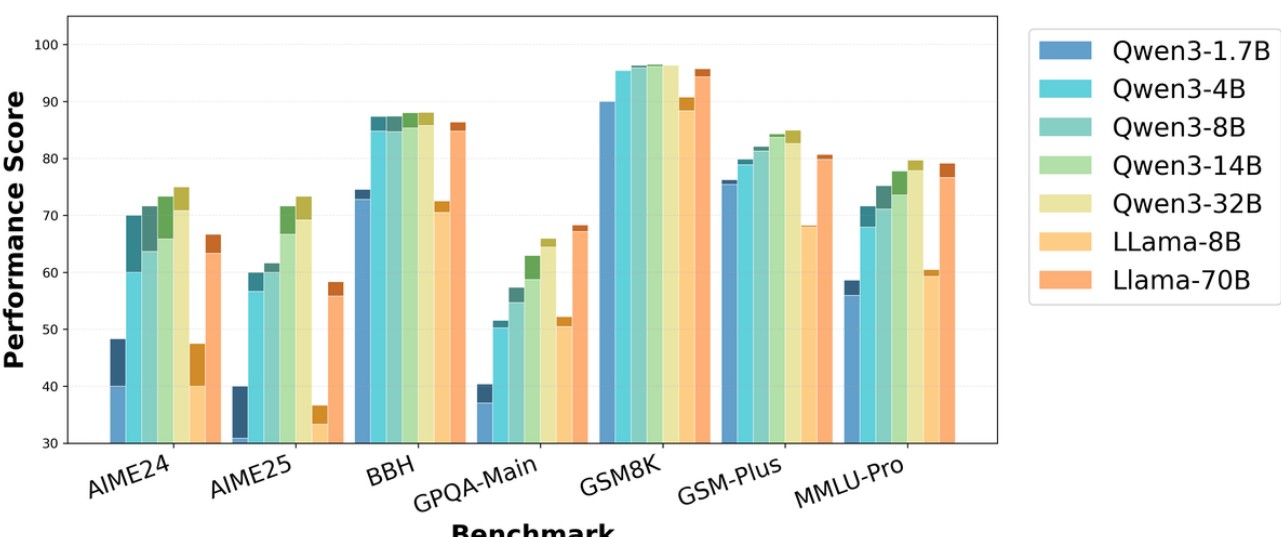

*Figure 26.* Overall performance comparison across all models and benchmarks. Light-colored bars represent baseline performance, while dark-colored bars show results after applying our RR method. Each model family uses a distinct color scheme for easy identification.

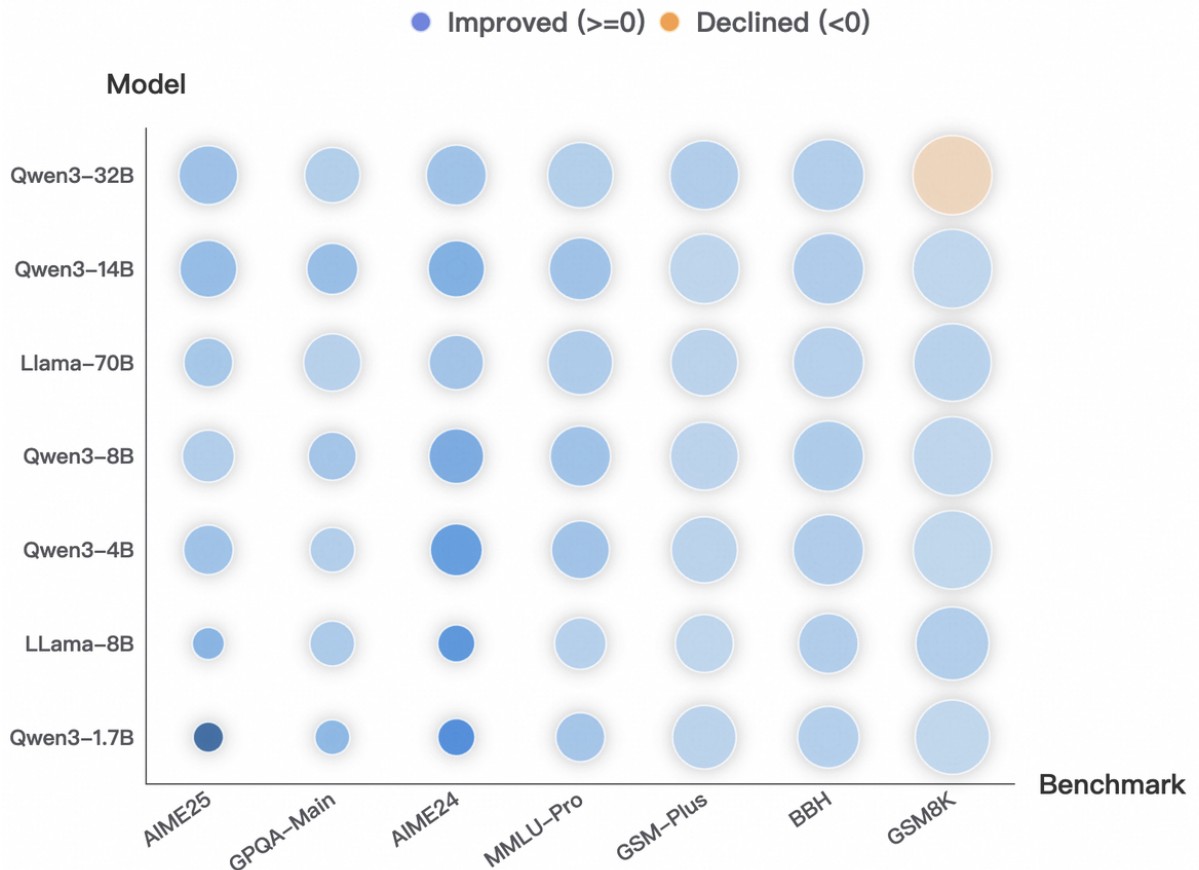

*Figure 27.* Bubble chart showing the combined relationship among baseline performance, RR improvement, and model size across benchmarks. The x-axis lists benchmarks (sorted by average baseline score) and the y-axis lists models (sorted by overall baseline performance). Bubble size encodes the baseline score on each benchmark, while bubble color encodes the RR improvement (blue for positive gains, orange for negative drops; darker shades indicate larger magnitude).

### G.2. Benchmark-Specific Performance Patterns

The magnitude of improvement varies meaningfully across benchmarks, closely tied to how intensively models engage in multi-step reasoning at the layers where RR intervenes (see Section K). The most dramatic gains occur on competition-level mathematical tasks: on AIME2024, Qwen3-4B improves by +10.0% (60.0% → 70.0%), and Qwen3-1.7B gains +9.2% on AIME2025 (30.8% → 40.0%). These tasks often induce bottlenecks—such as strategy switching or constraint integration—that trigger high-norm activation spikes; RR's iterative refinement at these moments enables effective "double-checking" and representation strengthening.

In contrast, benchmarks like BBH show smaller but consistent gains (+1–3%). BBH's diverse task types yield mixed sensitivity to norm-based recursion, averaging to moderate improvement. On well-optimized benchmarks like GSM8K, performance remains stable, confirming that RR selectively enhances reasoning where it is most needed.

In summary, RR delivers consistent and meaningful improvements across diverse architectures and tasks. These results demonstrate that RR is not only a practical and universally applicable method for enhancing reasoning, but also provide strong empirical validation that peaks in the $\ell_2$ norm of hidden states reliably signal critical reasoning moments worthy of targeted refinement.

### G.3. How Many Loops Are Enough?

It is worth noting that increasing the number of loop iterations is *not* always beneficial. We conduct a simple diagnostic experiment on the 25-th layer of Qwen3-1.7B by repeatedly applying the same transformation and tracking the activation

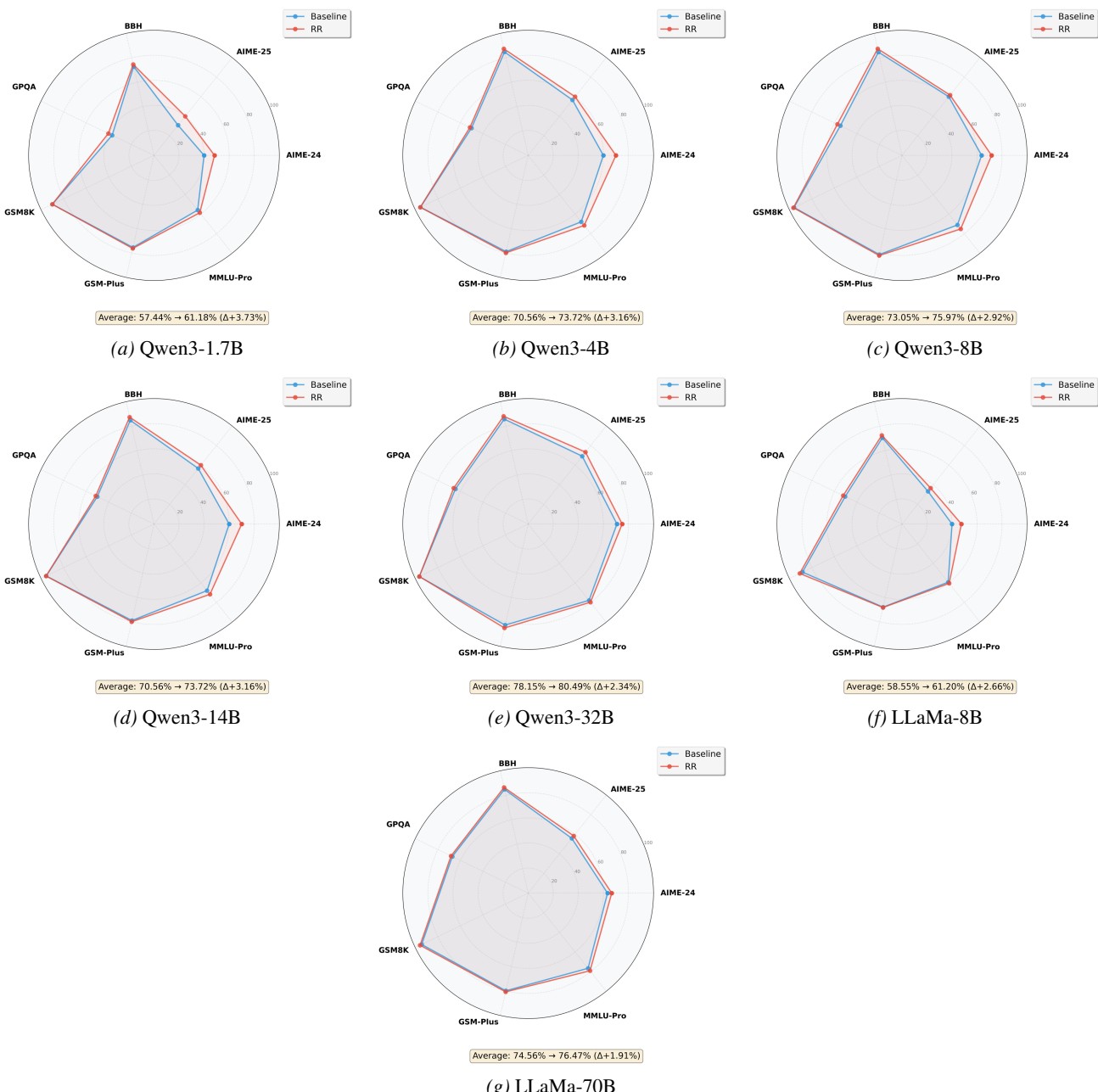

*Figure 28.* Individual performance analysis for all seven models. Each radar chart compares baseline performance (dashed lines) with RR-enhanced results (solid lines) across seven benchmarks. The consistent outward expansion of the RR curves demonstrates universal effectiveness across different model sizes and architectures.

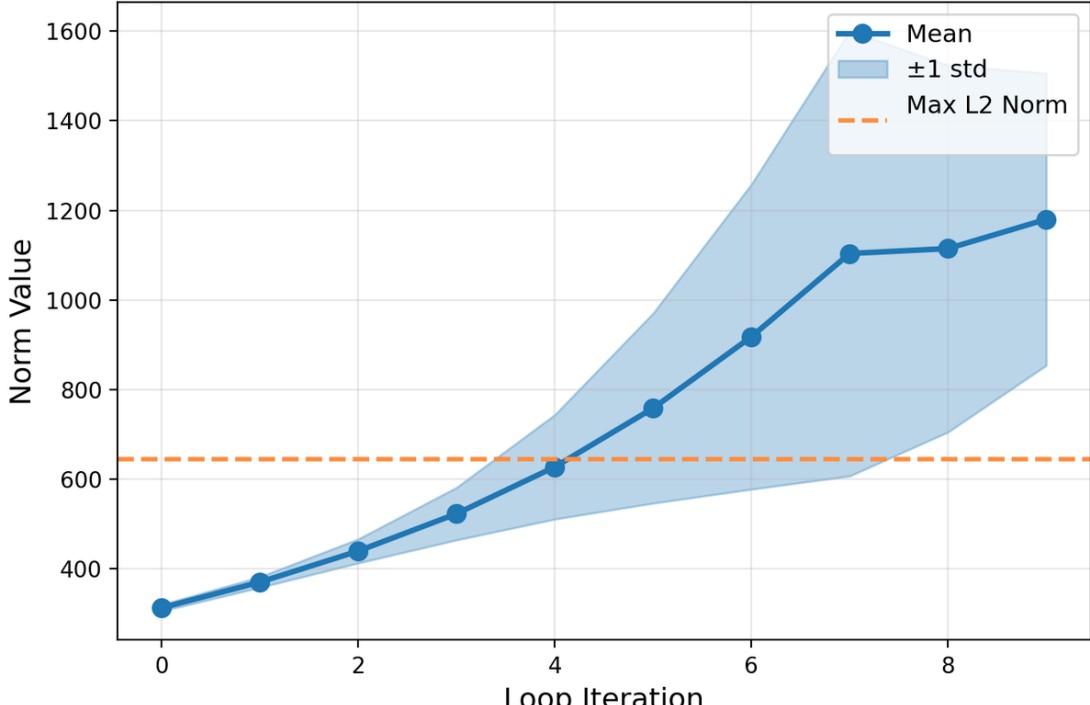

*Figure 29.* Activation $\ell_2$-norm statistics versus loop iterations on the 25-th layer of Qwen3-1.7B. The blue curve denotes the mean with $\pm 1$ standard deviation shading, and the dashed line indicates the current maximum $\ell_2$ norm.

$\ell_2$-norm statistics across loop iterations. As shown in Figur e 29, the norm grows rapidly as the loop count increases; in our estimate, looping up to 3 times is relatively safe, while 4 loops is likely to exceed the current maximum norm observed in the model, which may lead to instability. Moreover, due to a fixed-point-like effect, the marginal gain of additional loops diminishes quickly, making it unnecessary to increase the loop iterations indefinitely.

## H. Endogenous Reasoning State Steering

We apply the same evaluation protocol to our Endogenous Reasoning State Steering (ERSS) method across the identical set of seven state-of-the-art thinking models under the same benchmark suite described in Appendix E. And our detailed results are in Table 8. Figure 30 provides a holistic overview of performance improvements, while Figure 31 visualizes the interplay between model size, baseline performance, and ERSS-induced gains across benchmarks. The detailed performance of each model is shown in the Figure 32.

The observed trends closely mirror those reported in Appendix G for Adaptive Layer-wise Reasoning Recursion: ERSS consistently enhances reasoning performance across all models and tasks, with particularly strong gains on complex mathematical benchmarks and more modest but stable improvements on general reasoning suites. This reinforces the robustness of norm-guided intervention strategies and further validates the $\ell_2$-norm as a reliable signal for identifying critical reasoning states.

The strong similarity in empirical outcomes between RR and ERSS stems from their shared mechanistic foundation: both methods target the same underlying phenomenon that high-$\ell_2$-norm hidden states are indicators of critical reasoning moments and seek to deepen the model's internal processing at these points. Specifically, RR achieves this through *recursive refinement*, re-injecting and iteratively enhancing representations at norm peaks to simulate "thinking longer." In contrast, ERSS performs *endogenous state steering* by nudging the high-norm activations toward semantically similar, high-quality reasoning trajectories drawn from external memory or past successful executions.

Despite their operational differences, both strategies amount to a form of *intensive exploitation of high-signal reasoning states*. This common objective explains why they yield quantitatively and qualitatively consistent improvements across models and benchmarks, reinforcing the central role of $\ell_2$-norm peaks as actionable anchors for reasoning enhancement.

*Table 6.* Performance comparison with steering methods on Qwen3-4B across reasoning benchmarks.

| Method | aime24 | aime25 | bbh | gpqa_main | gsm8k | gsm-plus | mmlu-pro |
|---|---|---|---|---|---|---|---|
| Base | 60.00 | 56.66 | 84.78 | 50.22 | 95.41 | 78.90 | 67.92 |
| **ERSS (Ours)** | **66.67** | **60.83** | 87.18 | **51.33** | **96.68** | 79.76 | **71.15** |
| Entropy | 56.67 | 53.33 | 85.87 | 48.06 | 96.38 | 79.48 | 68.55 |
| LogProb | 56.67 | 55.00 | 85.61 | 47.29 | 95.49 | 79.46 | 69.41 |
| SAE | 65.00 | 58.33 | **87.51** | 50.73 | 96.45 | **80.15** | 70.64 |

*Table 7.* Performance comparison with steering methods on R1-Distilled-LLaMA-8B across reasoning benchmarks.

| Method | aime24 | aime25 | bbh | gpqa_main | gsm8k | gsm-plus | mmlu-pro |
|---|---|---|---|---|---|---|---|
| Base | 40.00 | 33.33 | 70.48 | 50.45 | 88.34 | **67.95** | 59.27 |
| **ERSS (Ours)** | 51.67 | **40.00** | 70.90 | **55.13** | 90.40 | 66.43 | **60.67** |
| Entropy | 40.00 | 36.67 | 70.77 | 53.75 | 88.47 | 66.33 | 58.84 |
| LogProb | 43.33 | 33.33 | **71.27** | 53.53 | 88.36 | 65.93 | 59.10 |
| SAE | **53.33** | 38.33 | 70.94 | 54.20 | **90.87** | 65.88 | 59.57 |

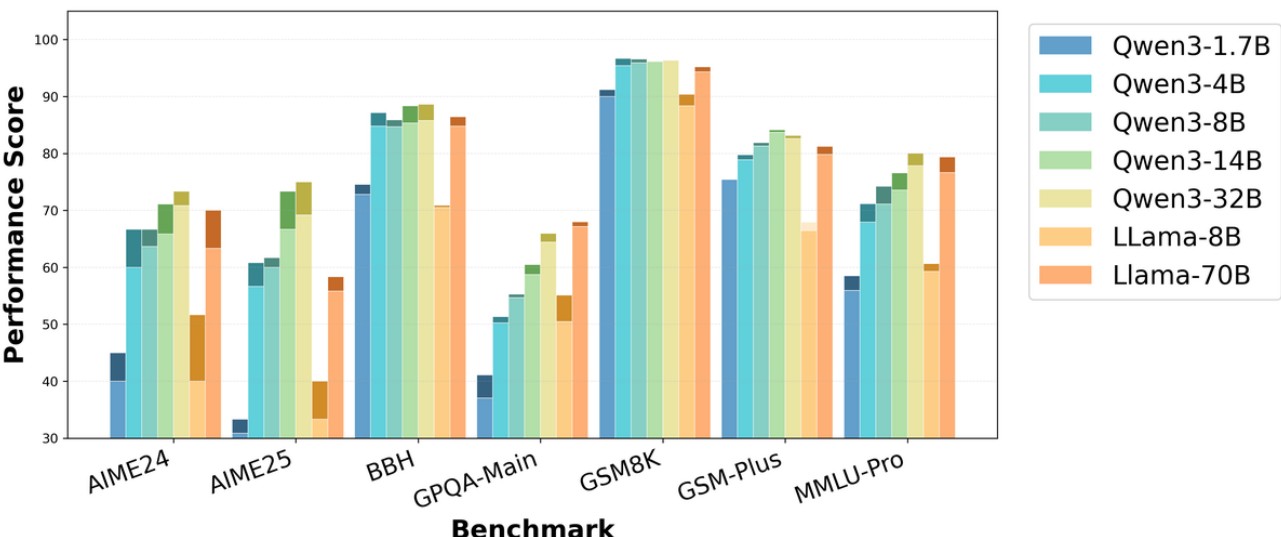

*Figure 30.* Overall performance comparison across all models and benchmarks. Light-colored bars represent baseline performance, while dark-colored bars show results after applying our ERSS method. Each model family uses a distinct color scheme for easy identification.

**Comparison with steering baselines.** Tables 6 and 7 compare **ERSS (Ours)** with representative steering-based baselines. Across both model families, **ERSS** delivers the most consistent gains over the base model, achieving the strongest overall performance on AIME, GPQA, and MMLU-Pro, while remaining competitive on GSM8K and BBH. Although SAE is occasionally the best method on individual benchmarks, its gains are less uniform across tasks and models. Entropy- and LogProb-based steering show even smaller or less stable improvements. Overall, these results indicate that **ERSS** offers a more robust steering signal for enhancing reasoning behavior across heterogeneous benchmarks.

# I. $\ell_2$-guided Response Selection

Beyond internal intervention, we explore the use of the $\ell_2$ norm as a *discriminative signal* for post-hoc response selection among multiple model generations. Specifically, we propose a simple yet effective ranking criterion: for each candidate response, we compute the Reasoning Score—**defined as the sum of $\ell_2$-norm peak heights over the final one-third of generated tokens in the deepest 25% layers.** This design reflects our observation that intense reasoning activity near the end of generation (i.e., during answer formulation) is more indicative of deliberative, high-quality reasoning than activity

*Table 8.* Performance comparison between Base models and Steer-enhanced variants across reasoning benchmarks. Accuracy (%) is reported, with relative improvement over Base shown in Impro (%).

| Model | Setting | Mathematical | | | | Complex | Scientific | Knowledge |
|---|---|---|---|---|---|---|---|---|
| | | AIME24 | AIME25 | GSM8K | GSM-Plus | BBH | GPQA-main | MMLU-Pro |
| Qwen3-1.7B | Base | 40.00 | 30.83 | 90.03 | **75.45** | 72.83 | 37.05 | 55.92 |
| | Steer | **45.00** | **33.33** | **91.21** | 75.43 | **74.55** | **41.07** | **58.49** |
| *Performance Gain (%)* | | *+12.50* | *+8.11* | *+1.31* | *-0.03* | *+2.36* | *+10.85* | *+4.59* |
| Qwen3-4B | Base | 60.00 | 56.66 | 95.41 | 78.90 | 84.78 | 50.22 | 67.92 |
| | Steer | **66.67** | **60.82** | **96.68** | **79.76** | **87.18** | **51.33** | **71.15** |
| *Performance Gain (%)* | | *+11.12* | *+7.34* | *+1.33* | *+1.09* | *+2.83* | *+2.21* | *+4.76* |
| Qwen3-8B | Base | 63.67 | 60.00 | 95.91 | 81.29 | 84.67 | 54.69 | 71.11 |
| | Steer | **66.67** | **61.67** | **96.60** | **81.88** | **85.87** | **55.28** | **74.21** |
| *Performance Gain (%)* | | *+4.71* | *+2.78* | *+0.72* | *+0.73* | *+1.42* | *+1.08* | *+4.36* |
| Qwen3-14B | Base | 65.83 | 67.66 | 96.13 | 83.71 | 85.33 | 58.71 | 73.56 |
| | Steer | **71.11** | **73.33** | **96.21** | **84.15** | **88.38** | **60.49** | **76.57** |
| *Performance Gain (%)* | | *+8.02* | *+8.38* | *+0.08* | *+0.53* | *+3.58* | *+3.03* | *+4.09* |
| Qwen3-32B | Base | 70.83 | 69.17 | **96.44** | 82.62 | 85.79 | 64.40 | 77.83 |
| | Steer | **73.33** | **75.00** | 96.36 | **83.15** | **88.61** | **65.96** | **80.03** |
| *Performance Gain (%)* | | *+3.53* | *+8.43* | *-0.08* | *+0.64* | *+3.29* | *+2.42* | *+2.83* |
| LLaMA-8B | Base | 40.00 | 33.33 | 89.23 | **67.95** | 70.48 | 50.45 | 59.27 |
| | Steer | **51.67** | **40.00** | **90.40** | 66.43 | **70.90** | **55.13** | **60.67** |
| *Performance Gain (%)* | | *+29.18* | *+20.01* | *+1.31* | *-2.24* | *+0.60* | *+9.27* | *+2.36* |
| LLaMA-70B | Base | 63.33 | 55.83 | 94.31 | 79.84 | 84.81 | 67.19 | 76.64 |
| | Steer | **70.00** | **58.33** | **95.19** | **81.25** | **86.42** | **67.97** | **79.40** |
| *Performance Gain (%)* | | *+10.53* | *+4.48* | *+0.93* | *+1.77* | *+1.90* | *+1.16* | *+3.60* |

distributed uniformly across the entire sequence.

As shown in Figure 33 and Table 11, selecting the candidate with the highest Reasoning Score consistently outperforms the naive baseline of choosing the response with the highest average log-probability (or equivalently, lowest perplexity, often approximated by mean token likelihood). Across all models and benchmarks, our $\ell_2$-guided selection yields higher accuracy, demonstrating that norm-based signals capture aspects of reasoning quality and further corroborates the functional significance of high-norm states: not only are they causally involved in reasoning (as shown in RR and ERSS), but their temporal profile also serves as a reliable proxy for solution correctness.

**Comparison with selection baselines.** Tables 9 and 10 compare **LRS (Ours)** with representative selection-based baselines on Qwen3-4B and R1-Distilled-LLaMA-8B. Overall, **LRS** achieves the most balanced improvements across benchmarks, consistently outperforming the base model and attaining the best or tied-best results on most tasks, especially on AIME, GPQA, GSM-Plus, and MMLU-Pro. In contrast, DeepConf-based methods occasionally obtain stronger results on individual benchmarks such as BBH or GSM8K, but their gains are less stable and often come at the cost of regressions on other tasks. This suggests that **LRS** provides a more reliable selection criterion for improving reasoning performance across diverse evaluation settings.

## J. Sensitivity Analysis

### J.1. Hyperparameters Sensitivity Analysis

We conduct extensive sensitivity analyses to assess the robustness of our methods. Table 12 reports the results across three key design dimensions.

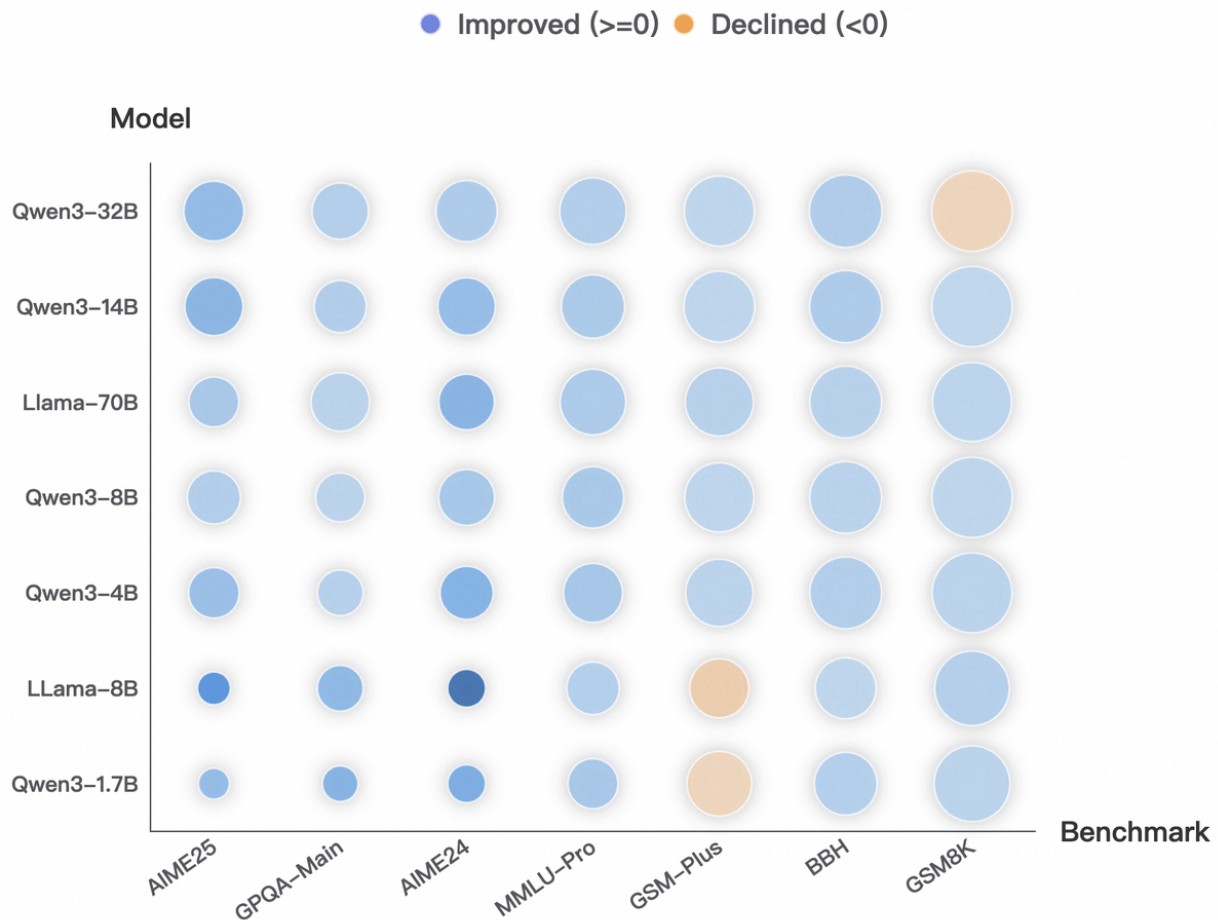

*Figure 31.* Bubble chart showing the combined relationship among baseline performance, ERSS improvement, and model size across benchmarks. The x-axis lists benchmarks (sorted by average baseline score) and the y-axis lists models (sorted by overall baseline performance). Bubble size encodes the baseline score on each benchmark, while bubble color encodes the RR improvement (blue for positive gains, orange for negative drops; darker shades indicate larger magnitude).

**IQR-based thresholding (ALRR and ERSS).** We compare our default dynamic IQR-based threshold against a fixed threshold (computed as the mean $\ell_2$-norm of MMLU-Pro samples) and several values of the multiplier $\tau$. Fixed thresholds dramatically degrade performance on hard reasoning tasks such as AIME (e.g., ALRR drops from 70.00 to 23.33 on AIME24), confirming that adaptive, per-sample thresholding is essential for distinguishing reasoning-intensive tokens across tasks of varying difficulty. The dynamic threshold remains stable under reasonable choices of $\tau$ (1.5, 3, 6), with $\tau = 3$ even yielding marginal improvements on several benchmarks. However, an overly aggressive threshold ($\tau = 0.5$) triggers excessive and unnecessary interventions, severely degrading reasoning coherence and causing substantial drops across all tasks (e.g., GSM-Plus drops by over 26 points for both methods).

**ALRR loop iterations.** A single refinement loop (Loop-1) already achieves significant gains over the base model (e.g., AIME24: $60.00 \rightarrow 66.67$), indicating that lightweight, one-step refinement suffices to capture the benefit. In contrast, excessive loops (Loop-8) substantially degrade performance, falling well below the base model on several tasks. This confirms that the effectiveness of ALRR stems from targeted feature refinement rather than from repeated computation, and that over-refinement disrupts the model's original reasoning dynamics.

**LRS layer aggregation strategy.** We compare three aggregation rules: the default (peak count normalized by total tokens), pure peak count, and sum of norms. All three yield comparable performance, indicating that the exact aggregation formula has only minor impact. In contrast, the choice of layers is critical: restricting aggregation to middle layers (middle layers) consistently underperforms the default, which draws signals from middle-to-late layers. This aligns with the intuition that deeper layers carry stronger and more task-specific reasoning signals, corroborating prior findings on the localization of

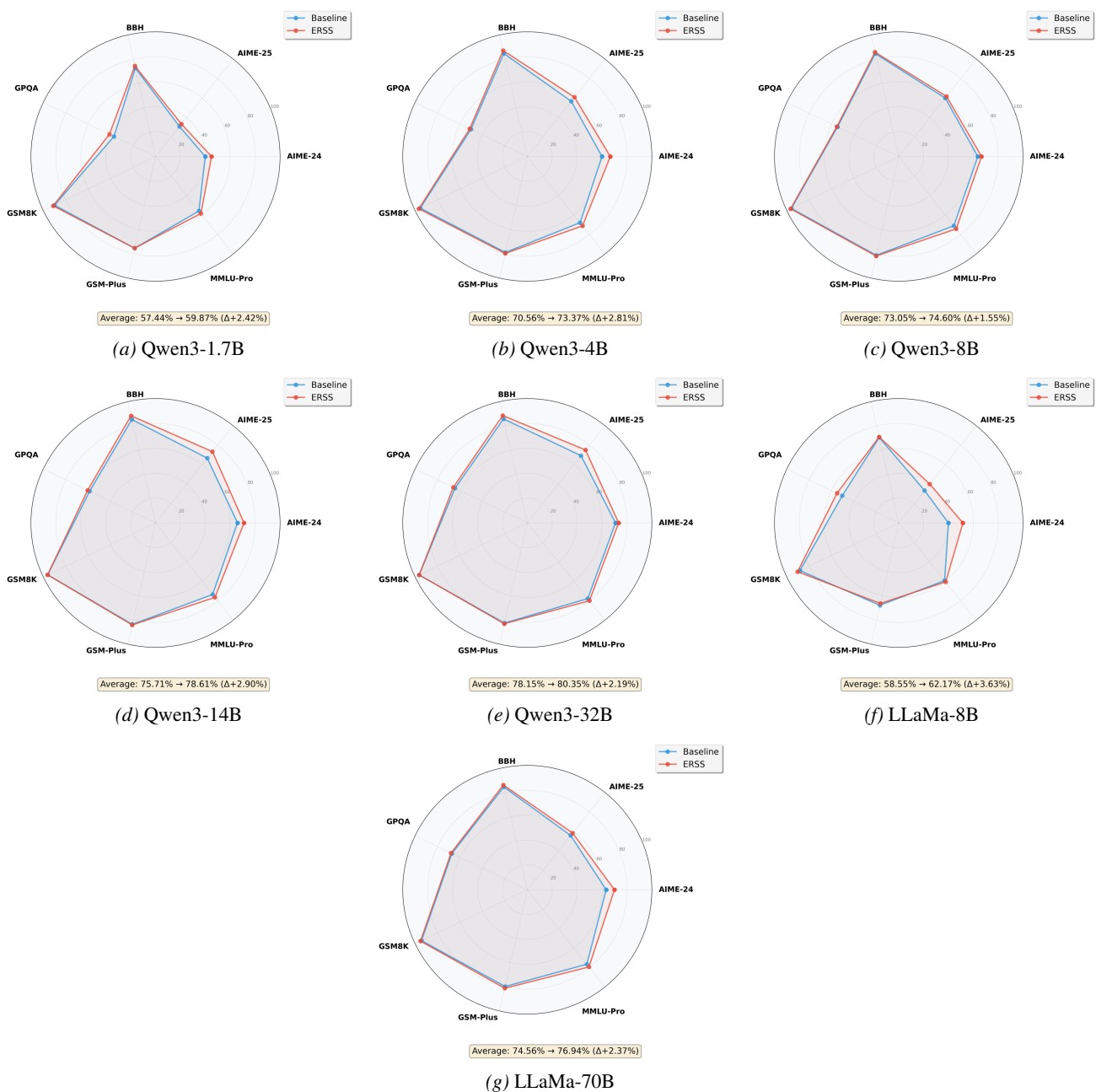

*Figure 32.* Individual performance analysis for all seven models. Each radar chart compares baseline performance (dashed lines) with ERSS-enhanced results (solid lines) across seven benchmarks. The consistent outward expansion of the ERSS curves demonstrates universal effectiveness across different model sizes and architectures.

reasoning in late transformer layers.

### J.2. Model Family Sensitivity Analysis

**Diversity of model families and additional evaluation.** Although both LLaMA3 and Qwen3 are transformer-based, they differ substantially in how reasoning ability is acquired: Qwen3 uses RL post-training to enhance reasoning, while the R1-distilled LLaMA3 variants rely primarily on SFT with distilled reasoning traces. Our original setup therefore already spans two qualitatively different reasoning regimes. To further test robustness, we additionally evaluate on Phi-4 and Gemma3. Phi-4 introduces a different data source and reasoning enhancement strategy, as its reasoning variant is trained primarily on

*Table 9.* Performance comparison with selection methods on Qwen3-4B across reasoning benchmarks.

| Method | aime24 | aime25 | bbh | gpqa_main | gsm8k | gsm-plus | mmlu-pro |
|---|---|---|---|---|---|---|---|
| Base | 60.00 | 56.66 | 84.78 | 48.88 | 95.41 | 79.90 | 67.92 |
| **LRS (Ours)** | **66.67** | **60.00** | 87.20 | **50.22** | 96.20 | **81.04** | **71.41** |
| DeepConf_filter | 56.67 | 53.33 | 88.56 | 47.67 | **96.45** | 79.90 | 65.20 |
| DeepConf_weighted | 63.33 | **60.00** | **89.81** | 48.36 | 95.83 | 80.08 | 61.12 |

*Table 10.* Table 2: Performance comparison with selection methods on R1-Distilled-LLaMA-8B across reasoning benchmarks.

| Method | aime24 | aime25 | bbh | gpqa_main | gsm8k | gsm-plus | mmlu-pro |
|---|---|---|---|---|---|---|---|
| Base | 40.00 | 33.33 | 70.48 | 50.45 | 88.34 | 67.95 | 39.75 |
| **LRS (Ours)** | **43.33** | **36.67** | 74.14 | **54.68** | 90.82 | **70.57** | **42.31** |
| DeepConf_filter | **43.33** | 33.33 | 76.83 | 53.13 | 88.40 | 68.38 | 39.02 |
| DeepConf_weighted | **43.33** | **36.67** | **80.65** | 50.45 | **90.96** | 68.47 | 39.16 |

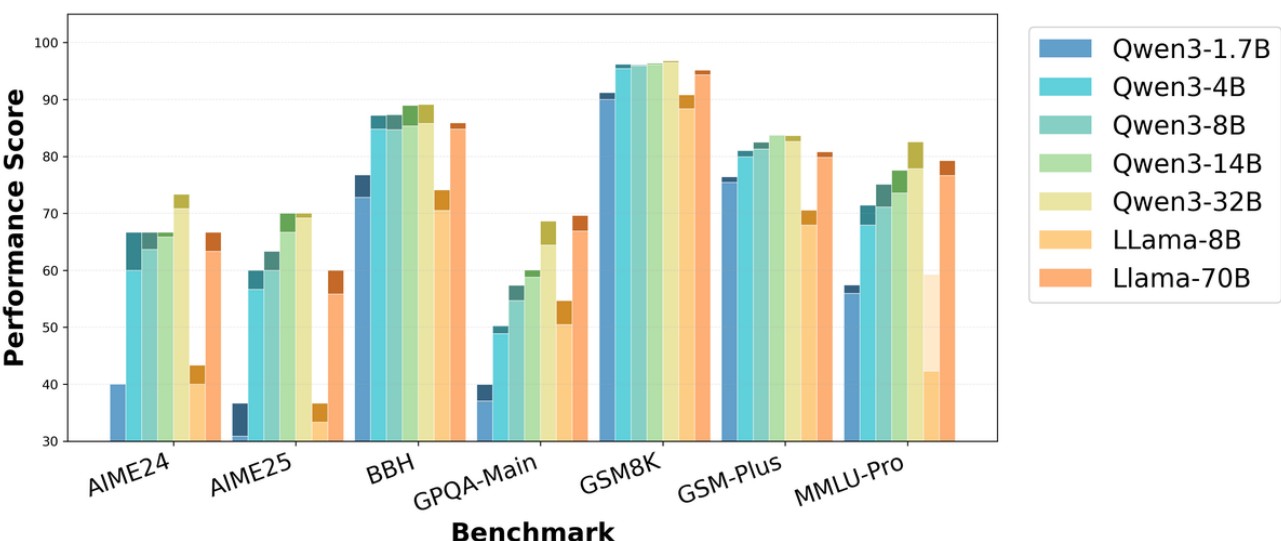

*Figure 33.* Overall performance comparison across all models and benchmarks. Light-colored bars represent mean performance, while dark-colored bars show results after applying our Norm Selection method. Each model family uses a distinct color scheme for easy identification.

high-quality synthetic data with dedicated reasoning optimization. Gemma3 further broadens coverage through a different training mixture, including multimodal data, and allows us to test scale sensitivity using both 4B and 12B instruction-tuned variants. Together, these models cover diverse axes including training data composition, model scale, and reasoning optimization strategy.

**Results and analysis.** As shown in Table 13, ALRR, ERSS, and LRS consistently improve performance across all model families and tasks. The gains are especially pronounced on reasoning-optimized models: for example, on Phi-4-Reasoning, ERSS improves AIME24 and GSM8K by 10.0 and 8.6 points, respectively, suggesting that our norm-based token selection complements rather than duplicates built-in reasoning optimization. At the same time, the methods remain effective on weaker or non-reasoning-specialized models. For instance, on Gemma3-4B-it, both ALRR and ERSS improve AIME24 by 3.33 points, and LRS improves GSM8K by 2.6 points despite the model's relatively modest baseline performance. Overall, the consistent gains across models trained with RL, SFT-distillation, synthetic data, and multimodal data indicate that our approach is not tied to a specific architecture or training paradigm, but instead captures a more general $\ell_2$-norm-based signal of token importance.

*Table 11.* Performance comparison between Base (Mean@8) and $\ell_2$-guided Selection across reasoning benchmarks. Accuracy (%) is reported, with relative improvement over Base shown in Impro (%).

| Model | Setting | Mathematical | | | | Complex | Scientific | Knowledge |
|---|---|---|---|---|---|---|---|---|
| | | AIME24 | AIME25 | GSM8K | GSM-Plus | BBH | GPQA-main | MMLU-Pro |
| Qwen3-1.7B | Mean | 40.00 | 30.83 | 90.03 | 75.45 | 72.83 | 37.05 | 55.92 |
| | Select | **40.00** | **36.67** | **91.21** | **76.43** | **76.75** | **39.96** | **57.42** |
| *Performance Gain (%)* | | *+0.00* | *+18.94* | *+1.31* | *+1.30* | *+5.38* | *+7.85* | *+2.68* |
| Qwen3-4B | Mean | 60.00 | 56.66 | 95.41 | 79.90 | 84.78 | 48.88 | 67.92 |
| | Select | **66.67** | **60.00** | **96.20** | **81.04** | **87.20** | **50.22** | **71.41** |
| *Performance Gain (%)* | | *+11.12* | *+5.89* | *+0.83* | *+1.43* | *+2.85* | *+2.74* | *+5.14* |
| Qwen3-8B | Mean | 63.67 | 60.00 | 95.91 | 81.29 | 84.67 | 54.69 | 71.11 |
| | Select | **66.67** | **63.33** | **96.06** | **82.52** | **87.30** | **57.36** | **75.12** |
| *Performance Gain (%)* | | *+4.71* | *+5.55* | *+0.16* | *+1.51* | *+3.11* | *+4.88* | *+5.64* |
| Qwen3-14B | Mean | 65.83 | 66.67 | 96.13 | 83.71 | 85.33 | 58.79 | 73.56 |
| | Select | **66.67** | **70.00** | **96.36** | **83.82** | **88.95** | **60.04** | **77.56** |
| *Performance Gain (%)* | | *+1.28* | *+5.00* | *+0.24* | *+0.13* | *+4.24* | *+2.13* | *+5.44* |
| Qwen3-32B | Mean | 70.83 | 69.17 | 96.44 | 82.62 | 85.79 | 64.40 | 77.83 |
| | Select | **73.33** | **70.00** | **96.81** | **83.67** | **89.14** | **68.64** | **82.54** |
| *Performance Gain (%)* | | *+3.53* | *+1.20* | *+0.38* | *+1.27* | *+3.90* | *+6.58* | *+6.05* |
| LLaMA-8B | Mean | 40.00 | 33.33 | 88.34 | 67.95 | 70.48 | 50.45 | 39.75 |
| | Select | **43.33** | **36.67** | **90.82** | **70.57** | **74.14** | **54.68** | **42.31** |
| *Performance Gain (%)* | | *+8.33* | *+10.02* | *+2.81* | *+3.86* | *+5.19* | *+8.38* | *+6.44* |
| LLaMA-70B | Mean | 63.33 | 55.83 | 94.31 | 79.84 | 84.81 | 66.91 | 74.64 |
| | Select | **66.67** | **60.00** | **95.14** | **80.82** | **85.92** | **69.64** | **79.26** |
| *Performance Gain (%)* | | *+5.27* | *+7.47* | *+0.88* | *+1.23* | *+1.31* | *+4.08* | *+6.19* |

### J.3. Robustness to SAE Design and Alternative Probes

We provide additional evidence that our main findings are robust to both SAE design choices and alternative probing signals.

**Robustness to SAE design.** We vary several key SAE configurations on Qwen3-4B, including the feature dimension, training budget, sparsity level, selection criterion, and feature subset size. Specifically, we reduce the feature dimension from 4 to 2, reduce training epochs from 6 to 3, increase the $L_1$ penalty $\lambda_{L_1}$ from 1e−3 to 5e−3, replace raw-difference selection with a relative-ratio criterion, and retain only the top-40 features. As shown in Figure 34, the late-layer increase in $\ell_2$ norm remains stable across all settings, indicating that our observations do not depend on a particular SAE hyperparameter choice.

**Alternative probes.** We further compare the $\ell_2$ norm with two information-theoretic probes, mutual information (MI) and information gain (IG), following prior work on identifying key reasoning tokens. Since MI and IG are token-level quantities, they are not directly visualized layer-wise. As shown in Figures 35, major $\ell_2$ peaks align well with MI and IG spikes at important reasoning steps. Some norm peaks also coincide with negative IG, which may reflect exploratory or trial-and-error reasoning. Overall, these results suggest that the $\ell_2$ norm captures similar key transitions while remaining substantially simpler and cheaper to compute.

### J.4. Compatibility of $\ell_2$-Based Methods

We further study the compatibility of our proposed $\ell_2$-based methods by evaluating different combinations of ALRR, ERSS, and LRS. The results are summarized in Table 14.

Overall, the results show that these methods are not uniformly additive. **ALRR+ERSS** is the most complementary combi-

*Table 12.* Ablation on layer aggregation strategies for ERSS, ALRR, and LRS on Qwen3-4B.

| Method | AIME24 | AIME25 | BBH | GPQA | GSM8K | GSM-Plus | MMLU-Pro |
|---|---|---|---|---|---|---|---|
| Qwen3-4B (Base) | 60.00 | 56.67 | 84.78 | 50.22 | 95.41 | 78.90 | 67.92 |
| **ALRR (default)** | | | | | | | |
| ALRR (default) | 70.00 | 60.00 | 87.39 | 51.55 | 95.53 | 79.88 | 71.66 |
| Fixed threshold | 23.33 | 16.67 | 85.02 | 25.78 | 95.39 | 78.93 | 69.59 |
| $\tau = 0.5$ | 36.67 | 23.33 | 70.56 | 36.43 | 80.64 | 52.69 | 39.88 |
| $\tau = 3$ | 70.83 | 60.83 | 86.88 | 52.91 | 95.59 | 79.91 | 72.73 |
| $\tau = 6$ | 69.17 | 60.83 | 86.70 | 52.25 | 96.34 | 79.98 | 71.41 |
| Loop-1 | 66.67 | 59.17 | 88.81 | 52.45 | 96.32 | 79.47 | 70.42 |
| Loop-8 | 36.67 | 20.00 | 60.25 | 35.94 | 67.92 | 53.49 | 46.55 |
| **ERSS (default)** | | | | | | | |
| ERSS (default) | 66.67 | 60.83 | 87.18 | 51.33 | 96.68 | 79.76 | 71.15 |
| Fixed threshold | 16.67 | 10.00 | 84.76 | 26.45 | 95.73 | 78.85 | 70.73 |
| $\tau = 0.5$ | 33.33 | 31.67 | 74.92 | 40.33 | 78.41 | 44.32 | 37.45 |
| $\tau = 3$ | 66.67 | 61.67 | 86.00 | 51.16 | 96.09 | 79.00 | 70.67 |
| $\tau = 6$ | 63.33 | 60.00 | 86.19 | 50.28 | 96.46 | 76.62 | 70.59 |
| **LRS (default)** | | | | | | | |
| LRS (default) | 66.67 | 60.00 | 87.20 | 50.22 | 96.20 | 81.04 | 71.41 |
| LRS (peak count) | 70.00 | 57.50 | 85.98 | 52.34 | 95.17 | 78.51 | 70.24 |
| LRS (sum) | 66.67 | 62.50 | 87.57 | 52.17 | 96.80 | 79.27 | 72.08 |
| LRS (middle layers) | 61.67 | 53.33 | 84.76 | 50.14 | 96.18 | 78.96 | 68.42 |

nation, yielding the strongest joint improvements on several benchmarks across both Qwen3-4B and R1-Distilled-LLaMA-8B. This suggests that refinement (high norm) and steering (consecutive low norm) act on different aspects of the reasoning process and can reinforce each other when combined. In contrast, **ERSS+LRS** generally performs worse than its individual components, indicating interference between steering and selection. A likely reason is that ERSS alters the norm distribution that LRS relies on for response ranking, thereby reducing its discriminability. **ALRR+LRS** remains more stable than ERSS+LRS, but still does not consistently outperform the best single-method results, suggesting partial interference between refinement and selection. Overall, these findings indicate that $\ell_2$-based interventions are most effective when they target distinct stages of the reasoning process, while combinations that directly perturb the selection signal require more careful calibration.

### J.5. Failure Modes and Boundary Conditions

We further evaluate our norm-guided methods on non-reasoning-centric benchmarks, including commonsense reasoning (HellaSwag), instruction following (IFEval), open-domain question answering (NQ-Open), contextual inference (PubMedQA), and truthfulness evaluation (TruthfulQA). The results are summarized in Table 15.

**Failure modes** While our methods are generally effective on reasoning-intensive tasks, their gains do not uniformly transfer to tasks centered on factual recall or direct answer retrieval. In particular, we observe consistent degradation on NQ-Open and PubMedQA across both model families. A plausible explanation is that increasing reasoning intensity can induce over-analysis: for open-domain QA, the model may attempt to "derive" facts that should instead be retrieved directly, while for contextual medical QA, excessive deliberation may lead to unnecessary caution or over-consideration of edge cases. These observations are consistent with prior findings that additional reasoning can be counterproductive on simple factual tasks. In contrast, IFEval often benefits from our interventions, suggesting that stronger reasoning can help the model better track and verify instruction constraints.

**Boundary conditions** These results highlight an important boundary condition of $\ell_2$-guided interventions. The $\ell_2$ norm appears to reflect computational intensity reliably, but whether increasing it is beneficial depends on the task. It serves as a useful signal for System-2-style tasks that require extended reasoning, verification, or constraint tracking, but may act as an overthinking signal for System-1-style tasks such as direct knowledge retrieval or intuitive commonsense judgments.

*Table 13.* Results across diverse model families (Phi-4 and Gemma3).

| Model | Method | AIME24 | AIME25 | BBH | GPQA | GSM8K | GSM-Plus | MMLU-Pro |
|---|---|---|---|---|---|---|---|---|
| Phi-4 | Baseline | 13.33 | 13.33 | 84.73 | 48.21 | 94.96 | 77.85 | 69.55 |
| | ALRR | 16.67 | 16.67 | 85.56 | 49.11 | **95.41** | 82.62 | 72.39 |
| | ERSS | **18.33** | **20.00** | 84.67 | **49.55** | 95.03 | 81.79 | 72.17 |
| | LRS | 15.00 | 15.83 | **85.61** | 48.55 | 95.38 | **82.86** | **72.55** |
| Phi-4-Reasoning | Baseline | 56.67 | 50.00 | 71.23 | 53.79 | 79.64 | 72.40 | 69.62 |
| | ALRR | 63.33 | **60.00** | 71.69 | 54.69 | 84.21 | 72.57 | 70.59 |
| | ERSS | **66.67** | 53.33 | **73.91** | **56.70** | **88.25** | **75.66** | **74.04** |
| | LRS | 65.00 | 56.67 | 71.27 | 55.47 | 82.68 | 72.45 | 71.83 |
| Gemma3-4B-it | Baseline | 10.00 | 13.33 | 69.03 | 26.67 | 87.60 | 70.79 | 42.73 |
| | ALRR | **13.33** | 16.67 | **69.92** | 26.45 | 87.64 | 71.16 | **43.41** |
| | ERSS | **13.33** | **20.00** | 69.67 | 27.12 | 88.78 | 70.80 | 42.98 |
| | LRS | 11.67 | 15.00 | 69.52 | **27.23** | **90.22** | **71.26** | 43.32 |
| Gemma3-12B-it | Baseline | 23.33 | 26.67 | 76.74 | 33.26 | 94.01 | 77.80 | 60.59 |
| | ALRR | **28.33** | **30.00** | 77.19 | 36.05 | **93.71** | 77.87 | 61.01 |
| | ERSS | 25.00 | 28.33 | **77.38** | **38.62** | 93.06 | **77.98** | **61.29** |
| | LRS | 26.67 | 28.33 | 76.79 | 36.05 | 93.75 | 77.82 | 60.90 |

*Table 14.* **Table 8: Combinations of $\ell_2$-based methods (ALRR, ERSS, LRS).**

| Model | Method | aime24 | aime25 | bbh | gpqa_main | gsm8k | gsm-plus | mmlu-pro |
|---|---|---|---|---|---|---|---|---|
| **Qwen3-4B** | Baseline | 60.00 | 56.67 | 84.78 | 50.22 | 95.41 | 78.90 | 67.92 |
| | ALRR | **70.00** | 60.00 | **87.39** | **51.55** | 95.53 | 79.88 | 71.66 |
| | ERSS | 66.67 | 60.83 | 87.18 | 51.33 | **96.68** | 79.76 | 71.15 |
| | LRS | 66.67 | 60.00 | 87.20 | 50.22 | 96.20 | **81.04** | 71.41 |
| | ALRR+ERSS | 68.33 | **63.33** | 87.29 | 50.54 | **96.68** | 80.46 | **71.90** |
| | ERSS+LRS | 60.00 | 50.00 | 84.81 | 46.88 | 95.00 | 79.02 | 67.64 |
| | ALRR+LRS | 63.33 | 58.33 | 87.17 | 49.41 | 96.13 | 79.46 | 71.78 |
| **R1-Distilled-LLaMA-8B** | Baseline | 40.00 | 33.33 | 70.48 | 50.45 | 88.34 | 67.95 | 59.27 |
| | ALRR | 47.50 | 36.67 | 72.53 | 52.23 | 90.78 | 68.25 | 60.47 |
| | ERSS | 51.67 | 40.00 | 70.90 | **55.13** | 90.40 | 66.43 | **60.67** |
| | LRS | 43.33 | 36.67 | **74.14** | 54.68 | **90.82** | 70.57 | 60.31 |
| | ALRR+ERSS | **53.33** | **43.33** | 72.03 | 53.12 | 90.61 | 66.86 | 60.23 |
| | ERSS+LRS | 36.67 | 23.33 | 69.44 | 48.21 | 87.64 | 65.77 | 58.04 |
| | ALRR+LRS | 46.67 | 33.33 | 72.03 | 52.12 | 89.86 | 67.38 | 59.32 |

# K. Layer Norm Variance in Different Benchmarks

The benchmark differences in $\ell_2$ norm statistics cannot be explained by a single factor. Instead, our results suggest that *task difficulty* and *task type* influence different aspects of the activation-norm dynamics. In particular, cumulative statistics such as total activation norm and layer amplification ratio are more strongly associated with task difficulty, whereas local spike statistics such as spike magnitude and spike frequency are more indicative of the underlying reasoning style, especially whether the task is mathematical or knowledge-intensive.

**Difficulty primarily affects cumulative norm statistics.** Figure 36 shows the per-layer total activation norm, averaged over sequences. All benchmarks exhibit a monotonic increase from layer 27 to layer 34, consistent with the norm growth phenomenon in transformer residual streams. More importantly, the hardest benchmarks, AIME and GPQA, accumulate substantially larger total norms than easier tasks, with a clear gap emerging in later layers. This pattern suggests that total norm primarily reflects *overall computational load*: harder tasks typically require longer reasoning trajectories and more decode steps, so norm contributions accumulate over a larger number of steps. In this sense, total activation norm behaves more like a proxy for the total amount of reasoning performed than for any specific task domain.

This interpretation is reinforced by the layer amplification ratio in Figure 38 (right), which measures the ratio between the last-layer and first-layer norms. Here again, the highest values are attained by AIME and GPQA, while easier tasks such

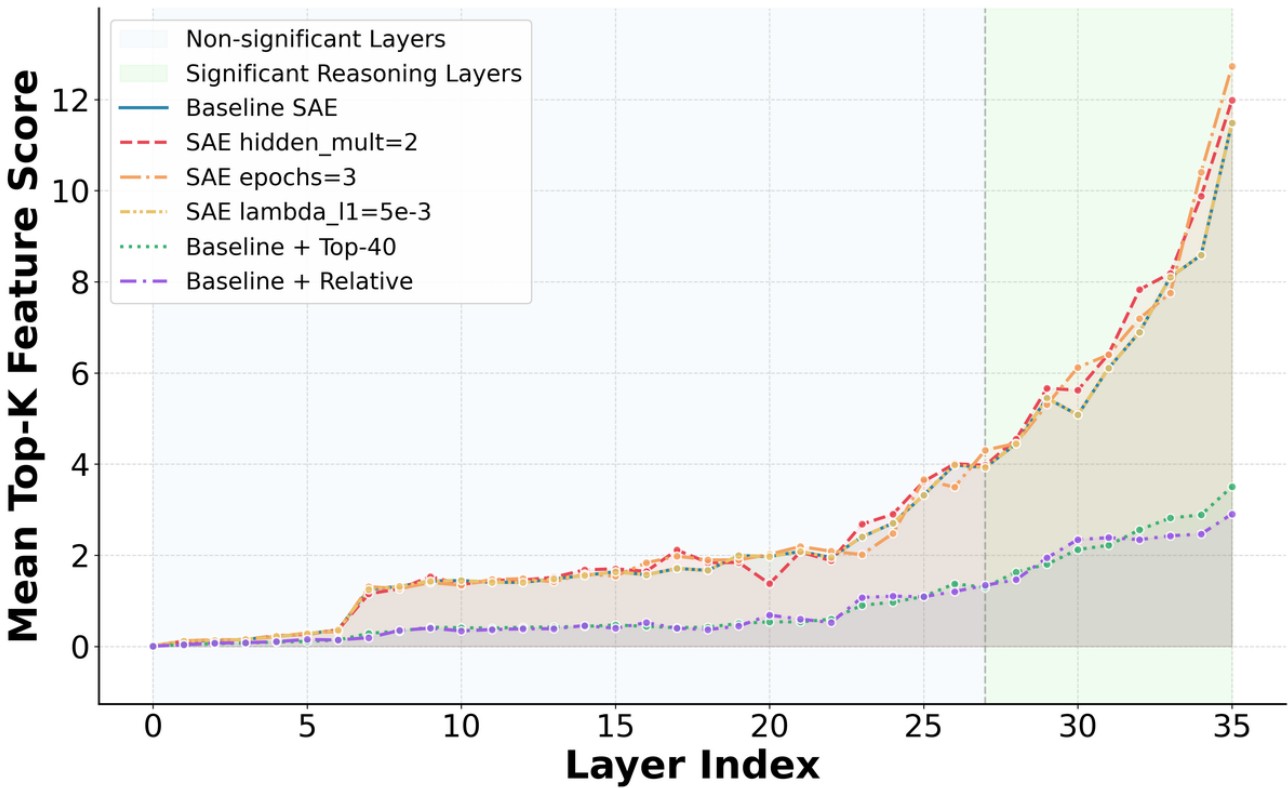

*Figure 34.* Robustness of the late-layer $\ell_2$ norm increase under different SAE configurations on Qwen3-4B. The overall pattern remains stable across changes in architecture, training budget, sparsity, selection criterion, and feature subset size.

as GSM_PLUS and GSM8K show smaller amplification. Unlike total norm, this ratio is less sensitive to absolute response length and more directly reflects the extent to which the model relies on deeper-layer processing. The fact that both difficult math reasoning (AIME) and difficult knowledge reasoning (GPQA) exhibit similarly strong amplification suggests that this metric is driven primarily by difficulty rather than by whether the task is mathematical.

**Task type primarily affects spike structure.** By contrast, the spike-based statistics in Figure 38 reveal a different axis of variation. The average over-threshold spike magnitude (left) is highest for math tasks, especially GSM8K and GSM_PLUS, whereas the spike frequency per decode step (middle) is lowest for these same tasks. In other words, mathematical reasoning tends to produce *rare but intense* activation spikes. This pattern is consistent with a chain-like computation process in which most steps proceed relatively smoothly, but a small number of key transitions require strong internal reorganization, for example when the model reaches a crucial intermediate deduction or a decisive computational step.

Knowledge-intensive and diverse reasoning tasks exhibit the opposite pattern: their spike magnitudes are more moderate,

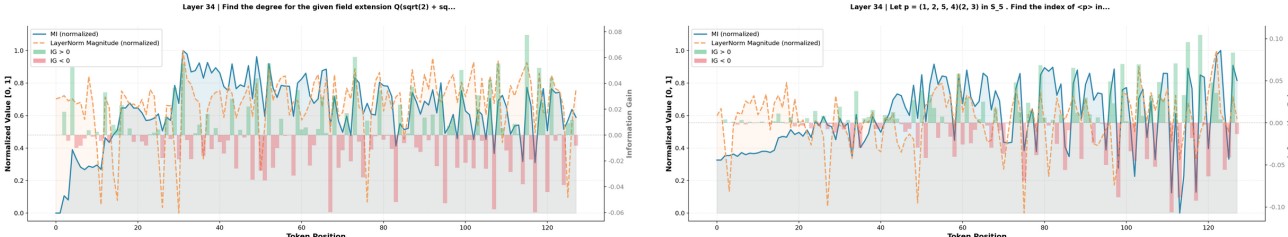

*Figure 35.* Comparison of $\ell_2$ norm dynamics with alternative information-theoretic probes. Major norm peaks align well with fitted MI/IG-related curves at important reasoning steps, while some peaks also coincide with negative IG, suggesting exploratory or trial-and-error reasoning behavior.

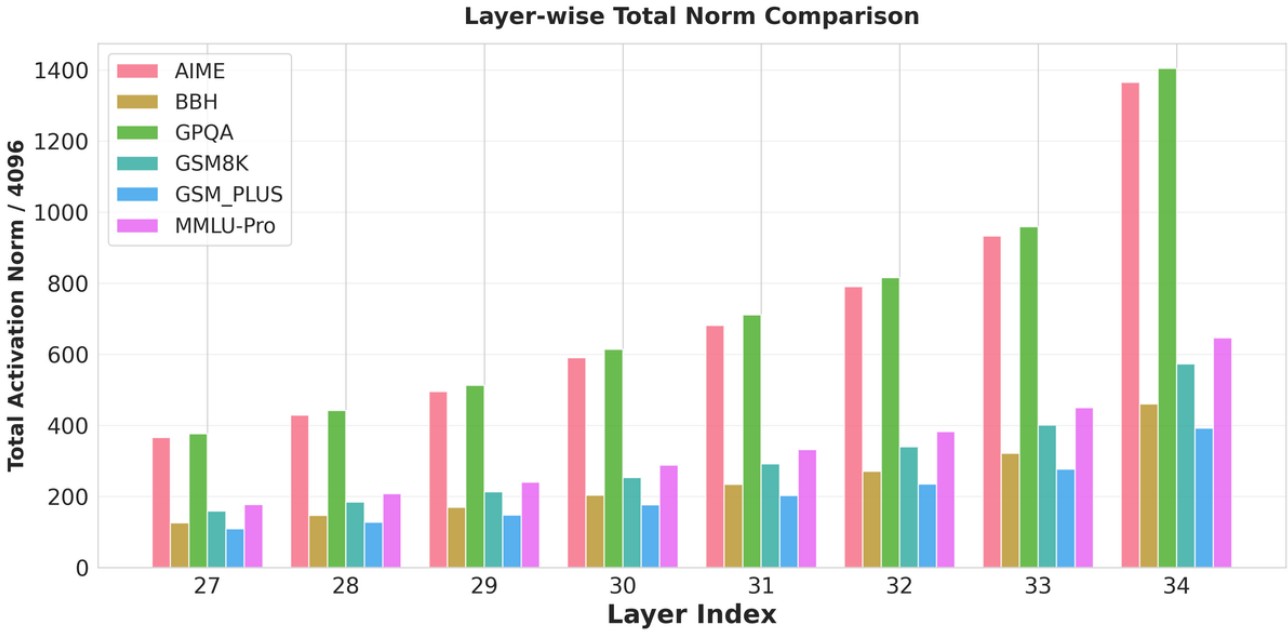

*Figure 36.* Per-layer total activation norm (divided by 4096), averaged over sequences. AIME and GPQA accumulate 2–3× more total norm than GSM tasks, reflecting their longer reasoning chains. The shaded region denotes ±1 standard deviation across sequences.

*Figure 37.* Grouped bar chart of the same per-layer total norm data as Figure 36. The consistent ordering across all layers confirms that the norm gap is a global property of the task, not an artifact of specific layers.

*Table 15.* **Table 9: Results on general benchmarks including commonsense reasoning, hallucination detection, knowledge probing, instruction following, and faithfulness-based extraction tasks.**

| Model | Method | hellaswag | ifeval | nq_open | pubmed_qa | truthfulqa |
|---|---|---|---|---|---|---|
| **Qwen3-4B** | Baseline | **80.61** | 49.04 | **28.80** | **68.20** | **46.14** |
| | ALRR | 80.58 | **49.76** | 25.18 | 66.80 | 43.76 |
| | ERSS | 80.60 | 49.28 | 25.24 | 67.00 | 44.67 |
| | LRS | 80.59 | 48.92 | 26.78 | 67.80 | 44.30 |
| **R1-Distilled-LLaMA-8B** | Baseline | 68.25 | 46.28 | **23.24** | **68.20** | 44.19 |
| | ALRR | 67.89 | 47.60 | 21.72 | 65.00 | 44.79 |
| | ERSS | 68.20 | **48.80** | 22.30 | 66.40 | 43.94 |
| | LRS | **68.61** | 47.48 | 22.05 | 67.40 | **44.80** |

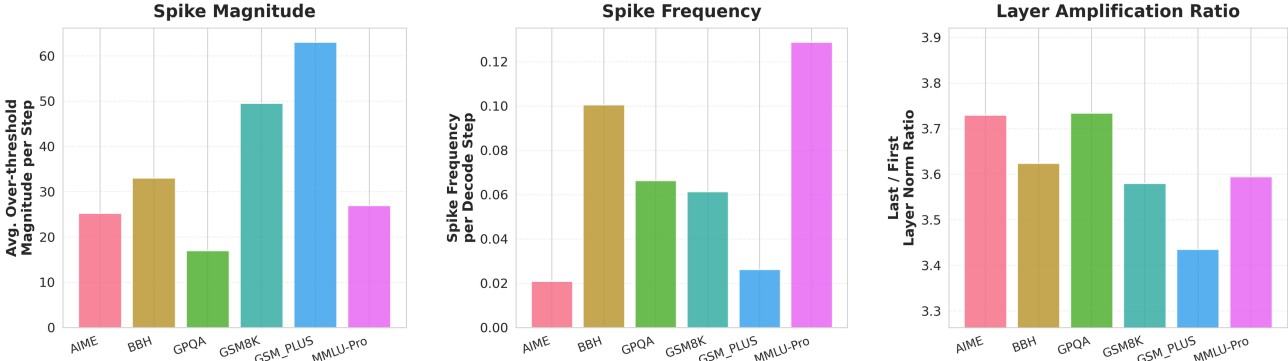

*Figure 38.* Three key metrics across benchmarks. **Left**: Average over-threshold magnitude per decode step. **Middle**: Spike frequency per decode step. **Right**: Last-to-first layer norm ratio.

but their spike frequencies are substantially higher. This suggests a more retrieval-like or switching-heavy computation pattern, in which the model repeatedly makes smaller local updates while moving among multiple knowledge subspaces or partial hypotheses. From this perspective, spike magnitude and spike frequency do not mainly track how hard a task is, but rather how the reasoning process is organized over time.

**Difficulty and task type are separable factors.** GPQA provides a particularly clear example of this separation. It behaves like AIME in cumulative metrics, with high total norm and high layer amplification, indicating strong computational demand. However, in spike behavior it is closer to knowledge-intensive tasks, with relatively frequent and less extreme activations. This suggests that GPQA is difficult in the sense of requiring substantial overall computation, but its reasoning dynamics are not math-like. Conversely, AIME reflects the combined effect of both factors: as a hard task, it has large cumulative norm and strong amplification, but as a math task, it retains the sparse, high-impact spike structure characteristic of mathematical reasoning.

**Summary.** Taken together, these results suggest that $\ell_2$ norm is not a one-dimensional signal. Different norm-based statistics capture different properties of the underlying computation. Total norm and layer amplification mainly reflect how much computation the model performs and how deeply that computation propagates through the network, making them more sensitive to task difficulty. Spike magnitude and frequency, in contrast, characterize the temporal structure of that computation, making them more sensitive to task type and reasoning style. This distinction helps explain why some benchmarks are similar along one norm axis but differ substantially along another, and provides a more structured interpretation of how activation norms relate to reasoning behavior across domains.

### K.1. Discussion

The analysis reveals that activation norm behavior is governed by two orthogonal factors:

**Task domain** determines the *pattern* of activation. Mathematical reasoning produces smooth, stable trajectories with rare but intense perturbations, while knowledge-intensive tasks produce noisy trajectories with frequent mild perturbations.

This likely reflects the fundamental difference between *compositional* computation (chaining logical steps) and *associative* computation (retrieving from parametric memory).

**Task difficulty** determines the *scale* of activation. Harder tasks accumulate more total norm (through longer responses) and require greater signal amplification through deeper layers (higher last/first ratio). The layer amplification ratio is the only metric where both easy-math benchmarks rank strictly in the bottom two, making it a promising difficulty proxy independent of domain.

The spike magnitude–frequency dissociation is particularly noteworthy: when mathematical reasoning *does* produce a spike, it is significantly larger than spikes in knowledge tasks. This may correspond to critical phase transitions in the reasoning chain — moments where the model must commit to a non-trivial logical step — and could provide useful signal for detecting reasoning bottlenecks during inference.

**Implications for Adaptive Inference.** These findings establish $\ell_2$ norm as a reliable runtime signal for computational load. The clear stratification of norms across difficulty levels—ranging from $\sim$200 for simple arithmetic to $>$2000 for olympiad mathematics, enables principled threshold selection for adaptive techniques. High-norm regions (e.g., AIME at Layers 34–35 with norms exceeding 1400) warrant enhanced computation such as iterative refinement loops, while low-norm regions (e.g., GSM8K in shallower layers with norms below 200) can safely skip redundant operations. Moreover, the strong deep-layer amplification suggests that adaptive interventions should concentrate on deeper layers where reasoning intensifies. For example, at Layer 35, the large gap between AIME ($\sim$2080) and BBH ($\sim$265) indicates where dynamic resource allocation is most impactful—applying expensive operations only to high-norm tasks while maintaining efficient processing for simpler ones.

## L. Pseudo Codes of Application Methods

In this section, we provide detailed pseudocode for the key algorithms used in our test-time reasoning techniques, which dynamically modulate LLM hidden states based on $\ell_2$ norm peaks. These methods can be flexibly applied for tasks such as causal suppression, reasoning recursion, or endogenous reasoning steering.

### L.1. Adaptive Threshold Estimation for Layer-wise $\ell_2$ Norm Peaks

To enable dynamic and layer-specific modulation, we employ an *adaptive threshold* mechanism based on online estimation of hidden state $\ell_2$ norms. For each layer, we maintain streaming estimates of the 25th and 75th percentiles (Q1 and Q3) of the observed norms. The threshold $\tau$ is computed as an upper fence of the interquartile range (IQR):

$$\tau = Q3 + k \cdot \text{IQR}, \quad \text{where IQR} = \max(\text{IQR}_{\min}, Q3 - Q1),$$

with hyperparameters $k > 0$ and $\text{IQR}_{\min} > 0$ ensuring numerical stability. This approach adapts automatically to the intrinsic scale of representations at each layer, obviating manual tuning. Algorithm 1 details the online computation procedure.

### L.2. Hidden State Suppression

In our causal intervention analysis, we apply *hidden state suppression* (Algorithm 2). Tokens with norms exceeding $\tau_\ell$ can be selectively suppressed, or optionally, tokens can be randomly suppressed for comparison. This allows controlled intervention to evaluate the causal effect of high-norm activations on final model outputs.

### L.3. Adaptive Layer-wise Reasoning Recursion

Norm peaks can trigger *adaptive recursion* (Algorithm 3), where a layer is recomputed iteratively until its hidden states fall below the threshold or a blowup limit is reached. This enables the model to internally refine high-activation states, effectively allocating more computation to layers exhibiting intense reasoning activity.

### L.4. Endogenous Reasoning State Steering

In *endogenous reasoning state steering* (Algorithm 4), historical high-norm hidden states are stored in a memory bank and selectively injected into current activations when low-activity streaks are detected. This mechanism promotes consistent

---

**Algorithm 1** Adaptive Threshold ($\tau$) Estimation via Online Quantiles

---

**Input:** Stream of scalar observations $x_t$ (e.g., $\|h_t\|_2$), target quantiles $q_1 = 0.25$, $q_3 = 0.75$, IQR multiplier $k > 0$, minimum IQR $\delta > 0$, floor $\tau_{\min} \geq 0$

**Output:** Time-varying threshold $\tau_t$

1: Initialize online quantile estimators $\mathcal{Q}_1 \leftarrow$ QuantileEstimator$(q_1)$
2: $\mathcal{Q}_3 \leftarrow$ QuantileEstimator$(q_3)$
3: **for** each new observation $x_t$ **do**
4:     $\mathcal{Q}_1$.update$(x_t)$
5:     $\mathcal{Q}_3$.update$(x_t)$
6:     $Q1 \leftarrow \mathcal{Q}_1$.value()
7:     $Q3 \leftarrow \mathcal{Q}_3$.value()
8:     IQR $\leftarrow \max(\delta, Q3 - Q1)$
9:     $\tau_t \leftarrow \max(\tau_{\min}, Q3 + k \cdot \text{IQR})$
10:    **use** $\tau_t$ for downstream decision (e.g., suppression)
11: **end for**

---

**Algorithm 2** Hidden State Suppression in Transformer Layers

---

**Input:** Hidden state $\mathbf{h} \in \mathbb{R}^d$, residual connection $\mathbf{r}$ (optional), layer index $\ell$, suppression mode $m \in$ {high_norm, random}, suppression layers $\mathcal{L}$, threshold $\tau_\ell$, suppression coefficient $\alpha$

**Output:** Modified hidden state $\mathbf{h}'$

1: **if** $\ell \notin \mathcal{L}$ **then**
2:     **return** $\mathbf{h}$
3: **end if**
4: $\mathbf{z} \leftarrow \mathbf{h} + \mathbf{r}$
5: $\mathbf{z}' \leftarrow \mathbf{z}$
6: **if** $m =$ high_norm **then**
7:     Compute norms: $n_i \leftarrow \|\mathbf{z}_i\|_2$ for all tokens $i$
8:     $\mathcal{M} \leftarrow \{i \mid n_i > \tau_\ell\}$
9: **else if** $m =$ random **then**
10:    Sample mask $\mathcal{M}$ with probability $p$
11: **end if**
12: **if** $\mathcal{M} \neq \emptyset$ **then**
13:    **for** each token $i \in \mathcal{M}$ **do**
14:       $\mathbf{z}'_i \leftarrow \alpha \cdot \mathbf{z}_i$
15:    **end for**
16: **end if**
17: $\mathbf{h}' \leftarrow \mathbf{z}' - \mathbf{r}$
18: **return** $\mathbf{h}'$

---

engagement of reasoning-relevant subspaces without additional training, providing a dynamic, data-free way to guide the model toward high-intensity reasoning trajectories.

## M. Traces Samples

Figures 39–43 visualize, for five representative samples, the tokens with unusually large hidden-state $\ell_2$ norms at the final layer of Qwen3-8B during <think> decoding (highlighted with red borders). Across these diverse trajectories, the highlighted tokens are consistently tied to reasoning-relevant content, such as logical operators and structural markers (e.g., *and/or*, *if*, *therefore*, *which*), recall/verification cues (e.g., *recall*, *check*), and domain-specific pivots that anchor intermediate hypotheses (e.g., *fixed*, *program*, *logic*, *array*). Importantly, the model does not merely amplify a fixed shortlist of words; instead, high norms emerge around *key decision points* that are specific to each sample, reflecting where the model concentrates computation to advance or revise the reasoning chain.

---

**Algorithm 3** Adaptive Layer-wise Reasoning Recursion

---

**Input:** Hidden state $\mathbf{h}$, layer index $\ell$, threshold $\tau_\ell$, max iterations $K$, blowup limit $B$
**Output:** Refined hidden state $\mathbf{h}'$
1: Compute norm $n \leftarrow \|\mathbf{h}\|_2$
2: **if** $n \leq \tau_\ell$ **then**
3:     **return** $\mathbf{h}$                                                      $\triangleright$ No recursion needed
4: **end if**
5: $\mathbf{h}^{(0)} \leftarrow \mathbf{h}$
6: **for** $k = 1$ to $K$ **do**
7:     $\mathbf{h}^{(k)} \leftarrow \texttt{Layer}_\ell(\mathbf{h}^{(k-1)})$                                          $\triangleright$ Recompute layer
8:     $n^{(k)} \leftarrow \|\mathbf{h}^{(k)}\|_2$
9:     **if** $n^{(k)} \leq \tau_\ell$ **or** $n^{(k)} \geq B$ **or** $|n^{(k)} - n^{(k-1)}|$ is negligible **then**
10:         **break**
11:     **end if**
12: **end for**
13: **return** $\mathbf{h}^{(k)}$

---

**Algorithm 4** Endogenous Reasoning State Steering

---

**Input:** Current hidden state $\mathbf{h}$, layer index $\ell$, adaptive threshold $\tau_\ell$, low-threshold factor $\gamma \in (0, 1)$, streak length $K$, injection probability $p$, cooldown period $C$
**Output:** Updated hidden state $\mathbf{h}'$
1: $n \leftarrow \|\mathbf{h}\|_2$
2: Retrieve per-token state $\mathcal{S} = (\texttt{bank}, \texttt{streak}, \texttt{cooldown})$
3: **if** $n < \gamma\tau_\ell$ **then**
4:     $\texttt{streak} \leftarrow \texttt{streak} + 1$
5: **else**
6:     $\texttt{streak} \leftarrow 0$
7:     **if** $n > \tau_\ell$ **then**
8:         Add $\mathbf{h}$ to $\texttt{bank}$ (keep top-$M$ by norm)
9:     **end if**
10: **end if**
11: **if** $\texttt{streak} \geq K$ **and** $\texttt{cooldown} = 0$ **and** $\text{rand}() < p$ **then**
12:     $\mathbf{v}^* \leftarrow \arg\max_{\mathbf{v} \in \texttt{bank}} \cos(\mathbf{v}, \mathbf{h})$
13:     **if** $\cos(\mathbf{v}^*, \mathbf{h}) \geq \theta_{\min}$ **then**
14:         $\mathbf{h}' \leftarrow \mathbf{h} + \beta \cdot \mathbf{v}^*$                                 $\triangleright$ Inject guidance
15:         $\texttt{streak} \leftarrow 0$
16:         $\texttt{cooldown} \leftarrow C$
17:     **end if**
18: **else**
19:     $\mathbf{h}' \leftarrow \mathbf{h}$
20:     **if** $\texttt{cooldown} > 0$ **then**
21:         $\texttt{cooldown} \leftarrow \texttt{cooldown} - 1$
22:     **end if**
23: **end if**
24: Update $\mathcal{S}$
25: **return** $\mathbf{h}'$

---

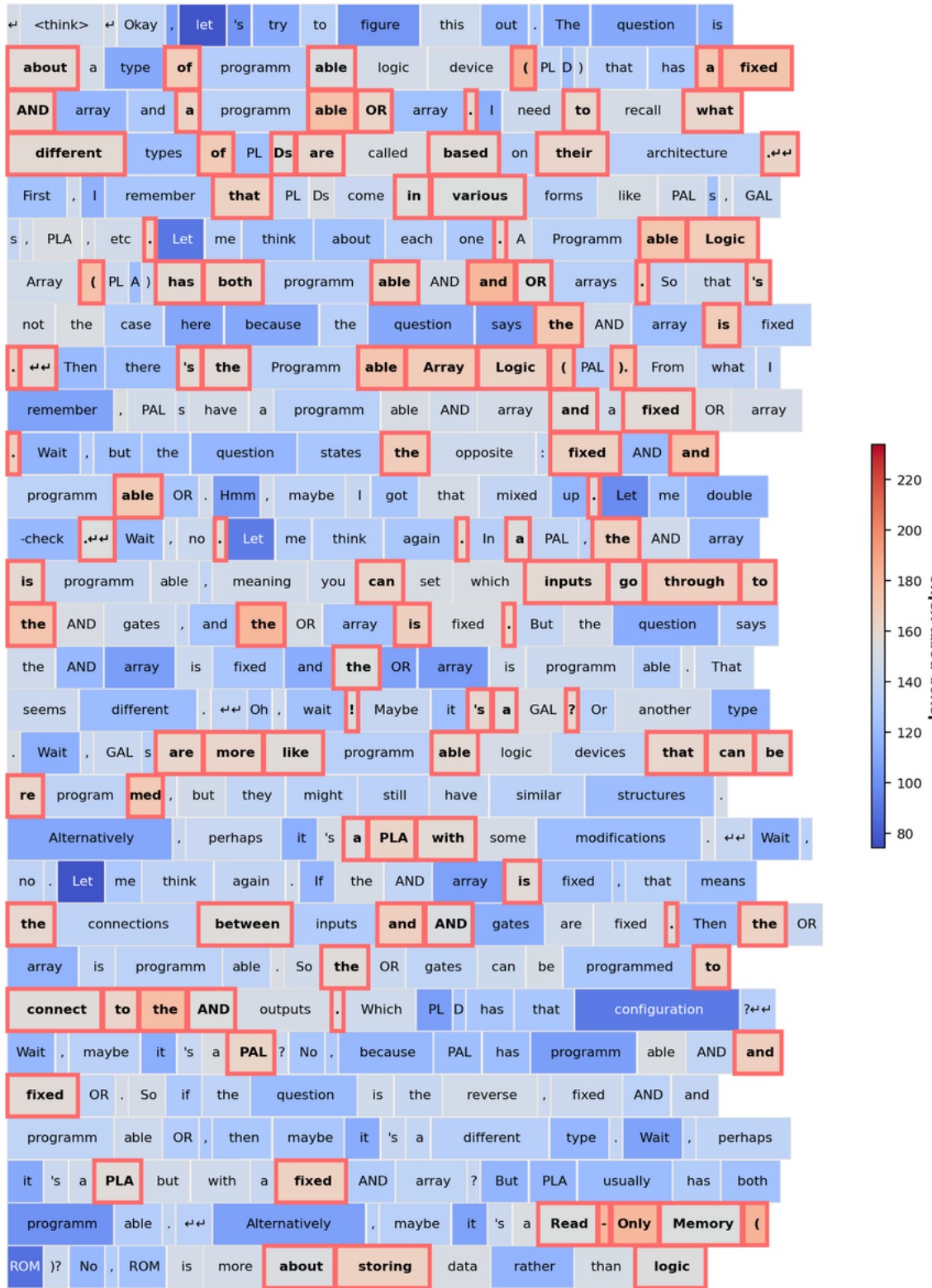

*Figure 39.* A reasoning example on Qwen3-8B

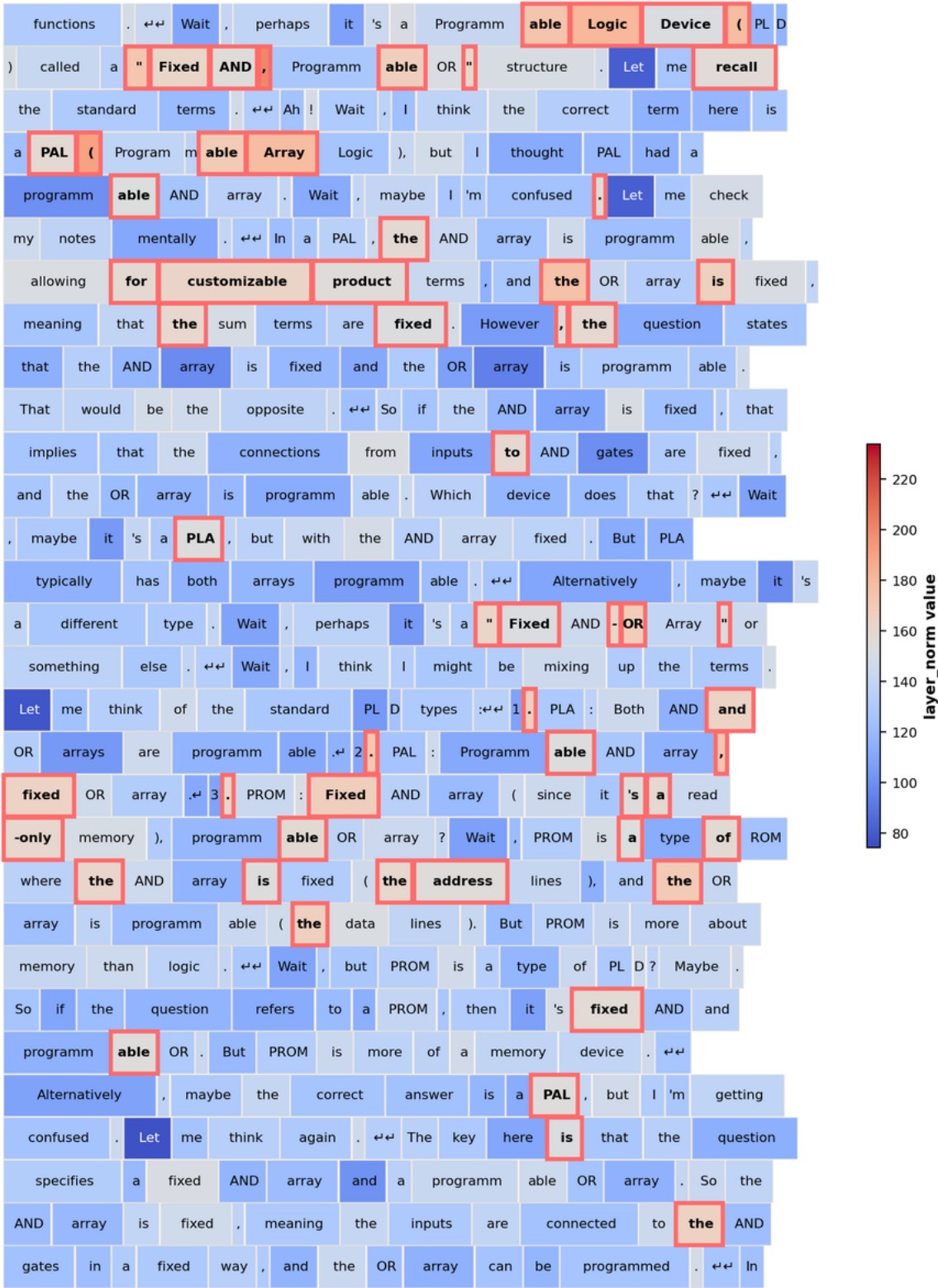

*Figure 40.* A reasoning example on Qwen3-8B

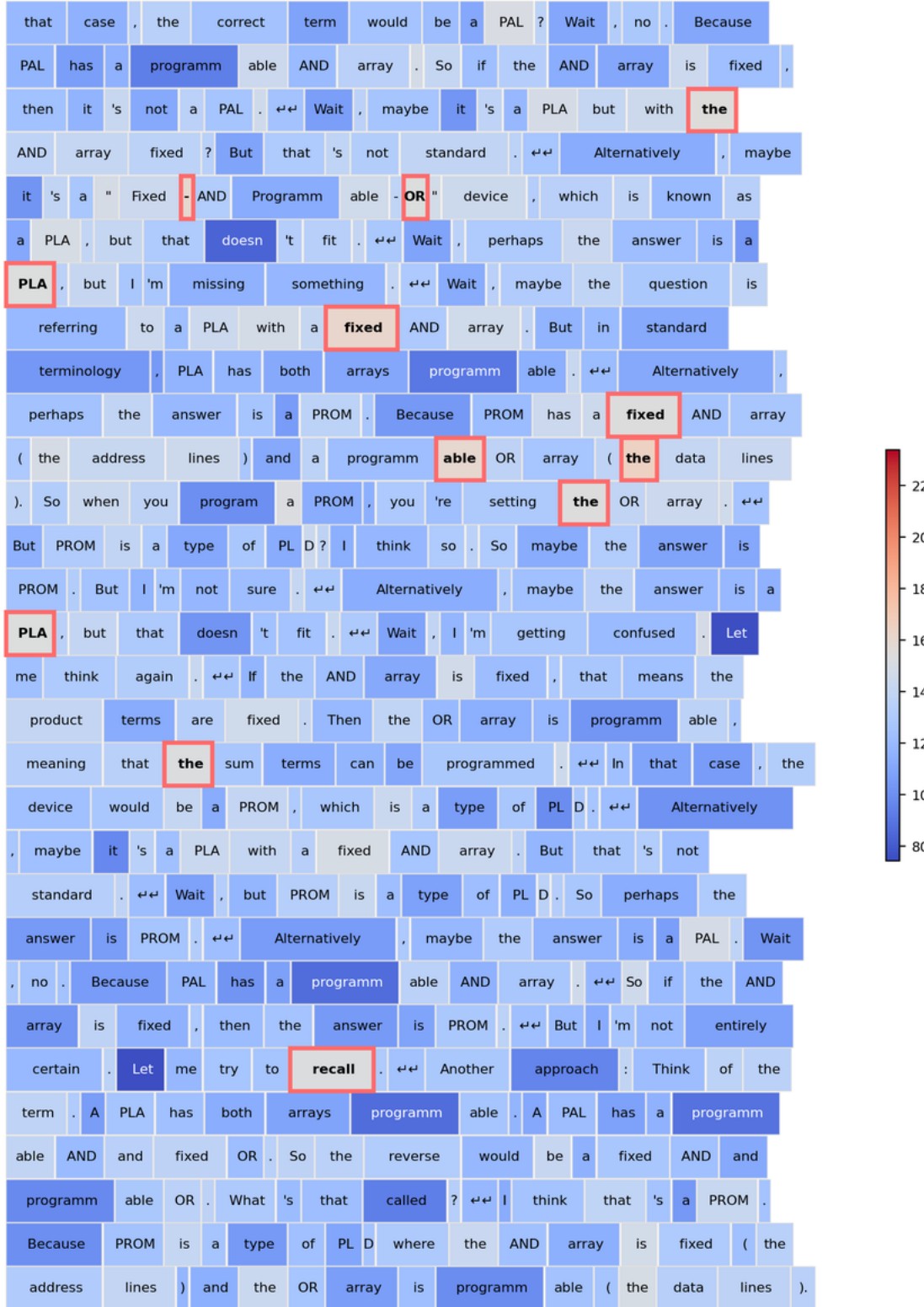

*Figure 41.* A reasoning example on Qwen3-8B

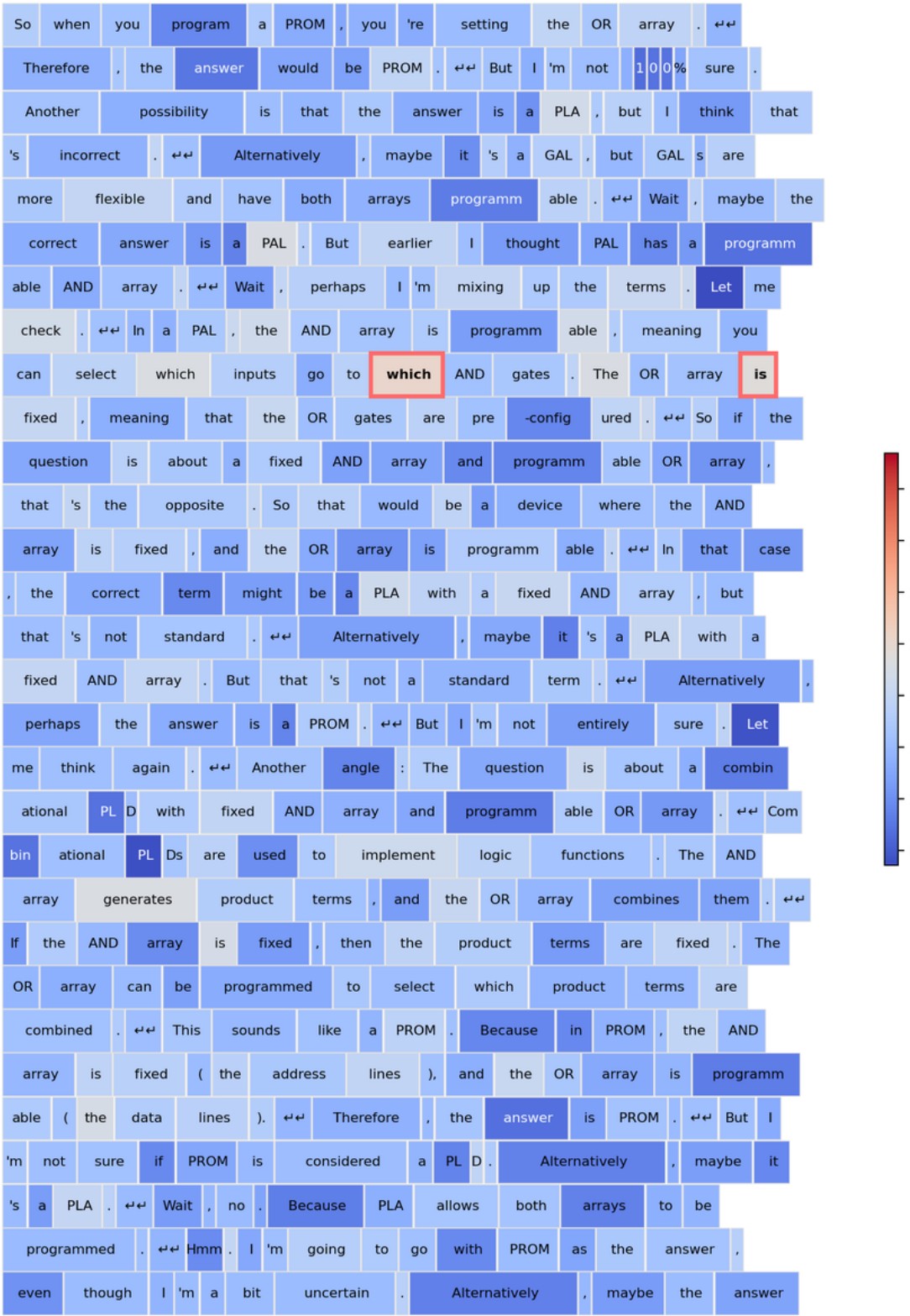

*Figure 42.* A reasoning example on Qwen3-8B

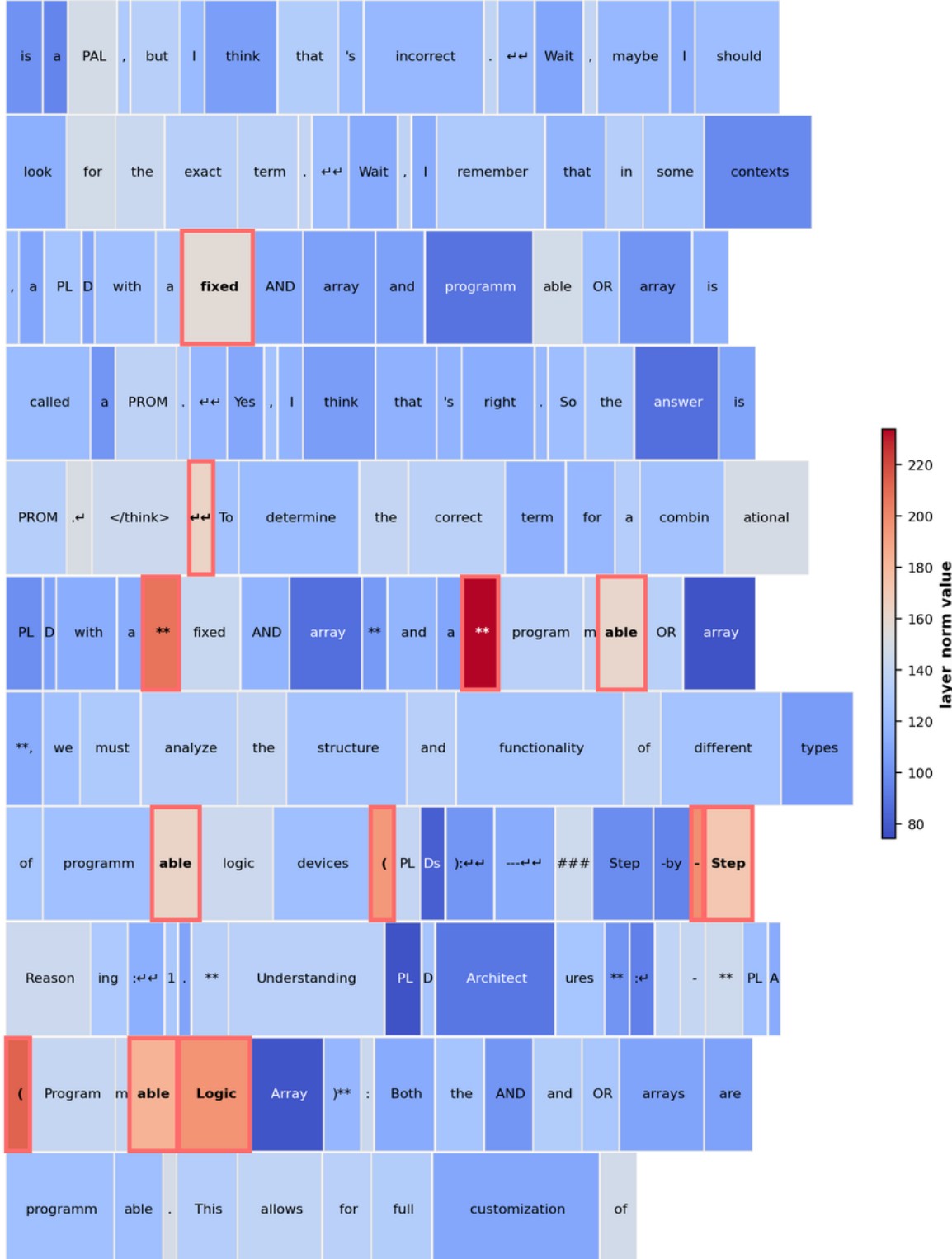

*Figure 43.* A reasoning example on Qwen3-8B

