# OpenReview forum: "The Tell-Tale Norm: $\ell_2$ Magnitude as a Signal for Reasoning Dynamics in Large Language Models"
_ICML.cc/2026/Conference — ICML 2026 spotlight_

### Official Review · Reviewer_HjMk · 2026-02-22

**Soundness:** 3
**Presentation:** 4
**Significance:** 3
**Originality:** 3
**Overall Recommendation:** 5
**Confidence:** 3

**Summary:**

This paper studies whether a simple, model-intrinsic signal can characterize reasoning dynamics in large language models (LLMs). The authors propose that the $\ell_2$ norm of hidden states serves as an endogenous indicator of reasoning intensity. Through layer-wise analysis, they show that reasoning-related activity concentrates in the final layers and is accompanied by a sharp increase in hidden-state magnitude.

The paper first uses Sparse Autoencoders (SAEs) as a diagnostic tool to identify reasoning features and demonstrates strong alignment between SAE activations and hidden-state $\ell_2$ norms. A theoretical analysis is provided to link SAE feature energy to the squared $\ell_2$ norm under standard assumptions. Empirically, the authors report consistent correlations between $\ell_2$ norms, reasoning difficulty across benchmarks, output entropy, and token-level control markers. A causal intervention study further shows that suppressing high-$\ell_2$ hidden states significantly degrades reasoning performance.

Building on these findings, the paper proposes three $\ell_2$-guided test-time techniques for reasoning enhancement, requiring no additional training. Experiments across multiple models and reasoning benchmarks demonstrate consistent performance improvements.

**Compliance With Llm Reviewing Policy:**

Affirmed.

**Final Justification:**

The authors’ rebuttal provides convincing empirical evidence and quantitative validation of the theoretical assumptions, as well as task-dependent boundary conditions, effectively addressing the key concerns raised.

**Key Questions For Authors:**

**Q1.** The paper provides strong empirical evidence that $\ell_2$ norm correlates with reasoning difficulty and mode. However, prior analyses indicate that hidden-state magnitudes may naturally increase with layer depth due to architectural scaling dynamics. Could the authors clarify whether the observed late-layer $\ell_2$ amplification during reasoning exceeds a baseline depth-dependent trend? A clear disentanglement between architectural scaling and reasoning-specific amplification would further solidify the causal interpretation and strengthen my confidence in the mechanism-level claim.

**Q2.** The theoretical analysis in Appendix B connects SAE feature energy to hidden-state $\ell_2$ magnitude under assumptions such as near-orthogonality and small reconstruction error. While the empirical correlations are convincing, could the authors provide quantitative diagnostics to assess how well these assumptions hold in practice? Such evidence would clarify how tightly the theoretical bounds reflect real model behavior and could elevate the contribution from strong empirical insight to stronger mechanistic validation.

**Q3.** Are there identifiable failure modes where high $\ell_2$ magnitude does not correspond to improved reasoning quality (e.g., hallucination-heavy tasks or adversarial prompts)? Clarifying boundary conditions would help delineate the scope of the proposed signal and better position its practical applicability. Evidence that the signal remains reliable outside structured reasoning benchmarks would further strengthen my evaluation.

**Limitations:**

Yes

**Strengths And Weaknesses:**

**Soundness.**

The paper is technically solid overall. The authors combine empirical analysis, theoretical arguments, and causal interventions to support their central claim that hidden-state $\ell_2$ norm reflects reasoning intensity. The theoretical section provides a careful derivation linking SAE feature energy to hidden-state magnitude under clearly stated assumptions, and the empirical results are extensive, covering multiple models and benchmarks. The intervention experiments (selective suppression of high-$\ell_2$ states) further strengthen the claim of functional relevance.

That said, several aspects could be further strengthened. First, while the empirical correlation between $\ell_2$ norms and SAE activations is convincing, some theoretical assumptions in Appendix B (e.g., near-orthogonality and small reconstruction error) are not quantitatively examined. Reporting reconstruction residual statistics or decoder orthogonality measures would help clarify how tightly the theoretical bounds reflect practical behavior.

Second, prior work analyzing normalization in Transformer architectures (e.g., Xiong R, Yang Y, He D, et al. On layer normalization in the transformer architecture[C]//International conference on machine learning. PMLR, 2020: 10524-10533.) has shown that Pre-Norm designs fundamentally alter gradient flow and activation scaling across layers. Related empirical observations in training-free layer selection studies indicate that hidden-state magnitudes can naturally increase with layer depth. The present work provides convincing evidence that deviations in $\ell_2$ magnitude correlate with reasoning difficulty and mode. However, it would further strengthen the argument to explicitly separate (i) baseline architectural norm growth from (ii) reasoning-induced amplification beyond that baseline. Controlled analyses on non-reasoning inputs or normalization-adjusted comparisons could clarify the extent to which the observed $\ell_2$ spikes exceed expected depth-driven scaling.

Overall, however, the claims are supported by substantial empirical evidence, and the identified limitations do not undermine the technical correctness of the work.

**Presentation.**

The paper is clearly written and well structured. The narrative progresses logically from empirical observation, to theoretical grounding, to intervention-based validation, and finally to practical test-time applications. The inclusion of extensive appendices improves transparency and reproducibility. The positioning with respect to SAE-based probing and reasoning analysis literature is generally appropriate. Minor improvements could include a more explicit discussion distinguishing architectural norm growth from reasoning-specific amplification to preempt potential confusion.

**Significance.**

The paper addresses an important and timely question: how to understand and enhance reasoning dynamics in large language models using intrinsic signals. The proposed $\ell_2$-based signal is simple, computationally cheap, and does not require training additional probes, which increases its practical appeal. The test-time techniques derived from this signal demonstrate consistent improvements without retraining, making the contribution relevant both for interpretability research and practical inference-time optimization.
While the empirical improvements are generally moderate rather than dramatic, the methodological simplicity and compatibility with existing inference pipelines make the contribution practically meaningful. The impact is likely to be incremental but broadly applicable, especially as a lightweight baseline or diagnostic tool for reasoning analysis.

**Originality.**

While the broader paradigm of analyzing internal representations and intervening at inference time has precedents, the identification of hidden-state $\ell_2$ norm as a probe-free, architecture-agnostic reasoning signal is a novel and elegant contribution. The work distinguishes itself by providing a cohesive framework that links representation analysis, theoretical interpretation, and practical test-time enhancement.

The originality lies less in introducing a fundamentally new modeling paradigm and more in recognizing and systematically validating a surprisingly simple intrinsic signal. Although elements of internal-state analysis and inference-time control have appeared in prior work, the unification of these components around a single scalar signal is well motivated and clearly articulated.

---

> ### Author Rebuttal · Authors · 2026-03-31
>
> We sincerely thank the reviewer for the insightful feedback.
>
> Rebuttal Tables and Figures: https://anonymous.4open.science/r/ICML2026-Rebuttal-6673/README.md
> # Q1: Architectural Scaling vs Reasoning Amplification
> Appendix J, Figure 9, 34, 35 show the LLM's layer-wise norms across benchmarks of varying reasoning difficulties. If the amplification were purely a result of architectural scaling, we would expect similar norm distributions across tasks. However, we observe that while $\ell_2$ increases with depth, the late-layer increase (L28–35) depends on reasoning intensity. For instance, at Layer 35, AIME (Olympiad math) norm reaches 2082.7 vs. 240.2 on GSM8K (~9× gap) within the same architecture.
>
> Rebuttal Fig.1 compares layer-wise $\ell_2$ across thinking vs. non-thinking modes on a same model and queries, showing that non-thinking follows a moderate upward drift closer to the architectural scaling, whereas thinking shows a much steeper escalation.
>
> While architectural scaling accounts for a fraction of the norm increase, the excess magnitude reflects reasoning amplification.
> # Q2: Validity of Theoretical Assumptions
> We empirically validate assumptions on Qwen3-4B and LLaMA3-8B (averaged on last 1/4 layers):
> 1. **B.1 Decoder Orthonormality** SAE decoder bases are near-orthonormal: diagonal ≈ 1.0000 (std $7.6×10^{-7}$), off-diagonal correlation $\epsilon_{orth}=0.0173\ll1$, consistent with [1].
> 2. **B.2 Reconstruction Quality** We observe low reconstruction error and high directional fidelity:
> - Reconstruction error: 8.7%
> - Cosine similarity of original and reconstructed representations: $cos(h_{think},\hat h_{think})=0.9932$, $cos(h_{nothink},\hat h_{nothink})=0.9831$
> 3. **B.3 & B.4 Semantic Orthogonality** For base semantics $h_{base}$ and reasoning signals $h_{reasoning}$:
> - $cos(h_{base},h_{reasoning}) = \mathbf{0.0570}$ indicates near-orthogonality and disentanglement between semantic and reasoning.
> - $|h_{base}|\approx 186.1,|h_{reasoning}|\approx 116.9$. Comparable norms prove that reasoning is a substantial component rather than noise.
> - Cosine drops from 0.3144(Layer 5) to 0.057(late). This transition indicates progressive disentanglement, consistent with Fig.2 dynamics.
> 4. **B.5 Feature Disentanglement** We reconstruct the reasoning direction using different feature subsets, confirming that reasoning is highly concentrated within $\mathcal{I}_{reasoning}$ and the separation of reasoning and base semantics is reliable.
> - Using the identified $k=668$ reasoning features, we faithfully reconstruct the reasoning direction: $cos(\sum_{i\in I_{reasoning}}\Delta z_i\cdot d_i,h_{reasoning}) = 0.9499$. Even in extreme sparsity (Top-5 features), the directional recovery remains high (0.8747).
> - Base Feature Independence: Base features show near-zero correlation with the reasoning direction: $cos(\sum_{i\in I_{base}}\Delta z_i\cdot d_i,h_{reasoning}) = 0.1443$
> 5. **Conclusion** Empirical results above show that the core geometric structures we rely on are supported, especially the near-orthogonality between semantic and reasoning directions and the concentrated reasoning features. Overall, while our assumptions do not hold in a strict equality in real-world scenarios, they are sufficiently satisfied for our framework.
> # Q3: Failure Modes and Boundary Conditions
> We extend experiments to non-reasoning-centric benchmarks, including commonsense (HellaSwag), instruction following (IFEval), open-domain QA (NQ-Open), contextual inference (PubMedQA), and truthfulness (TruthfulQA).
>
> **Results: Rebuttal Table 9 in the Link**
> 1. **Benchmark Performance and Failures**
>
> Our norm-guided interventions are generally effective but show degradation in some tasks.
> - NQ-Open: boosting reasoning may trigger hallucinations as LLMs attempt to "derive" facts that should be directly answered.
> - PubMedQA: over-analysis leads to excessive caution, replacing a definitive "Yes" by a "Maybe" due to the model over-evaluating fringe exceptions.
>
> These align with prior findings[2-4] that excessive reasoning can harm simple factual tasks.
>
> Interestingly, IFEval benefits, likely due to the model with boosted reasoning actively tracking and verifying constraints to better follow instructions.
>
> 2. **Boundary Conditions**
>
> $\ell_2$ reliably reflects computation intensity, but its benefit is task-dependent. We identify the boundary conditions as **System 1 vs System 2**. It serves as a powerful performance driver for System 2 tasks requiring extensive thinking (logical reasoning, constraint tracking), yet acts as a counterproductive overthinking signal for System 1 tasks (knowledge retrieval and intuitive commonsense).
> # References
> [1]Toy models of superposition.
>
> [2]Stop Overthinking: A Survey on Efficient Reasoning for Large Language Models TMLR
>
> [3]The Curse of CoT: On the Limitations of Chain-of-Thought in In-Context Learning TMLR
>
> [4]Mind Your Step (by Step): Chain-of-Thought can Reduce Performance on Tasks where Thinking Makes Humans Worse

---

> > ### Author Rebuttal · Reviewer_HjMk · 2026-04-02
> >
> > The authors’ rebuttal provides convincing empirical evidence and quantitative validation of the theoretical assumptions, I have updated my score.

---

> > > ### Author Response · Authors · 2026-04-03
> > >
> > > We **sincerely thank you for your insightful response** and for raising your score. We are **grateful** that our additional analyses on reasoning amplification, empirical validations of theoretical assumptions, and boundary conditions have addressed your concerns. **Your support is deeply appreciated, and we are thankful for the rigorous discussion that has significantly improved our paper.**

---

### Official Review · Reviewer_nk4U · 2026-03-10

**Soundness:** 3
**Presentation:** 4
**Significance:** 3
**Originality:** 3
**Overall Recommendation:** 5
**Confidence:** 3

**Summary:**

The paper proposes a new metric for quantifying reasoning intensity in language models, namely the l2 norm of the hidden vectors. Using Sparse Autoencoders (SAEs), the authors establish that reasoning mostly concentrates in the final layers of the models. Moreover, they prove both theoretically and empirically that the l2 norm bounds the activation strength of the SAE features. Building on these insights, the authors propose 3 different inference-time techniques that use the l2 norm to identify and amplify reasoning. Empirically, the authors observe consistent performance gains using these methods across multiple benchmarks and LLM model families.

**Compliance With Llm Reviewing Policy:**

Affirmed.

**Final Justification:**

The authors addressed all the concerns mentioned above.

**Key Questions For Authors:**

1. Have you explored any combination of the methods you proposed?

See also my questions from the weaknesses section.

**Limitations:**

Yes.

**Strengths And Weaknesses:**

Strengths:

1. The paper tackles an important and timely problem, namely understanding and controlling the reasoning dynamics in LLMs. The authors propose a very simple, yet elegant and efficient solution. Computing the l2 norm is computationally trivial, requires no additional training or data, but it is surprisingly powerful.

2. The paper makes multiple important contributions: an analysis establishing where reasoning concentrates in LLMs, establishes a theoretical and empirical link between the l2 norm and SAE reasoning features, and proposes three different inference-time techniques that can boost reasoning capabilities, proven on multiple models and benchmarks. All of these make the paper a very solid contribution to the field.

3. The paper is very well-written and structured. The authors present their contributions using the right amount of detail that makes it easy to follow, even for readers who are not super familiar with work in this area.


Weaknesses:
1. To identify reasoning-related features, the authors use paired responses: a thinking vs. a non-thinking response. This assumes that a non-thinking model does not try to do any internal reasoning, which to me seems like a strong assumption that should at least be discussed. I might be misunderstanding the reason why this is a good method. Could you elaborate on this?

2. The experiments focus only on two model families. Could you explain this choice? As far as I'm aware, these are similar architectures, trained similarly. This is minor but I think the paper would be stronger if you tested on slightly more diverse families.

---

> ### Author Rebuttal · Authors · 2026-03-31
>
> We sincerely thank the reviewer for the insightful feedback.
>
> Rebuttal Tables and Figures: https://anonymous.4open.science/r/ICML2026-Rebuttal-6673/README.md
> # W1: Clarification on Non-thinking
> 1. **Reasoning as a Spectrum, not a Binary State**
>
> We do not assume non-thinking models lack reasoning. We define reasoning as the logical transition from premises to conclusions following prior literature[1,2]. Thus all responses necessarily involve reasoning regardless of the generation mode.
>
> "Thinking" and "Non-thinking" modes refer to Qwen-3 configurations, trained via RL to exhibit different behaviors:
> - Non-thinking produces shallower, more straightforward, and less exploratory reasoning trajectories.
> - Thinking produces Long CoT, including deep logical branching and self-correction, representing a fundamentally more "intensive" version of the same cognitive process.
>
> By comparing these two modes, we are **not looking for the presence or absence of reasoning, but rather its intensity.**
>
> 2. **Empirical Consistency**
>
> Across experiments, $\ell_2$ reflects reasoning intensity rather than a binary distinction. Fig. 3 shows both modes have $\ell_2$ fluctuations (non-thinking may even spike at some steps), but thinking maintains a higher overall level. Rebuttal Fig. 1 shows both modes increase in later layers, with non-thinking consistently lower. Fig. 9 and Appendix J show norms scale with task difficulty: simpler GSM8K results in lower norms, while reasoning-intensive tasks (e.g., AIME) trigger significantly stronger signals.
>
> # W2: Generalization Across Diverse Model Families
> 1. **Diversity of the original model families**
>
> While both LLaMA3 and Qwen3 are transformer-based, their training paradigms and reasoning capabilities differ substantially. In particular:
> - Qwen3 incorporates RL post-training to explicitly enhance reasoning behaviors.
> - LLaMA3 (R1-distilled variants) rely primarily on SFT with distilled reasoning traces, without RL.
>
> Their differences have been discussed in prior work and our Appendix F. Our original setup covers two qualitatively different reasoning regimes.
>
> 2. **Additional results on more diverse model families**
>
> To further address the concern, we extend experiments to additional families with distinct characteristics:
> - Phi-4: trained on high-quality synthetic data; includes a reasoning-enhanced variant (Phi-4-Reasoning).
> - Gemma3: trained with multimodal data, representing a different data distribution and training objective.
>
> These models introduce diversity along data source (synthetic vs. multimodal), scale, and reasoning optimization, providing a broader testbed.
>
> **Results: Rebuttal Table 7 in the Link**
> 1. Our ALRR/ERSS/LRS consistently improve performance across all families.
> 2. The gains are particularly pronounced in reasoning-optimized models (Phi-4-Reasoning), suggesting our method effectively leverages stronger reasoning signals.
> 3. Even in weaker or non-reasoning-optimized models (Gemma3-4B-it), improvements remain stable, indicating robustness to model capability and training differences.
> # Q1: Combinations of $\ell_2$-based Methods
> We explore combinations of our proposed methods and summarize the findings below.
>
> **Results: Rebuttal Table 8 in the Link**
> 1. **ALRR + ERSS** (Refinement + Steering) are largely complementary. ALRR targets low-intensity moments for iterative refinement, while ERSS actively injects reasoning vectors into critical logical nodes. Combining them allows the model to both "think harder" at key steps and "re-think" shallow responses. This leads to new state-of-the-art results on several benchmarks (e.g., AIME25 for Qwen-3 and AIME24 for Llama-8B). However, because we used default hyperparameters for both, this method may over-perturb the original reasoning trajectory.
> 2. **ERSS + LRS** (Steering + Selection) largely performs worse than its individual components. Mechanistically, ERSS increases $\ell_2$ norms during low-activation steps, while LRS relies on relative norm differences across responses for selection. As a result, ERSS compresses the variance that LRS depends on, which leads to reduced discriminability between candidate responses and degraded reranking quality.
> 3. **ALRR + LRS** (Refinement + Selection). ALRR (looping) also tends to elevate the average norm (as discussed in Appendix G, Figure 29). While the interference is less severe than with ERSS, it still introduces some noise into the LRS ranking process.
> 5. **Conclusion**: Our investigation reveals that $\ell_2$-based interventions are most effective when they target distinct stages (e.g., ALRR for depth and ERSS for direction). However, when one method artificially alters the norm signal used as a metric by another (like ERSS vs. LRS), their joint application requires more careful signal calibration.
> # References
> [1]Thinking Machines: A Survey of LLM based Reasoning Strategies
>
> [2]Towards Reasoning Era: A Survey of Long Chain-of-Thought for Reasoning Large Language Models

---

> > ### Author Rebuttal · Reviewer_nk4U · 2026-04-01
> >
> > I thank the authors for their detailed response. It addresses all of my previous concerns; I have updated my score accordingly.

---

> > > ### Author Response · Authors · 2026-04-03
> > >
> > > We **sincerely thank you for your thoughtful feedback** and for raising your score. We are grateful that our clarifications of the "thinking" and the additional experiments on diverse model families and combined methods have addressed your concerns. Your support means a great deal to us, and we **truly appreciate the constructive discussion with you that has helped strengthen our work**.

---

### Official Review · Reviewer_ojhr · 2026-03-11

**Soundness:** 3
**Presentation:** 3
**Significance:** 3
**Originality:** 3
**Overall Recommendation:** 4
**Confidence:** 4

**Summary:**

This paper studies how to characterize and exploit reasoning dynamics inside LLMs, and proposes using the hidden-state $\ell_2$ norm as an intrinsic signal of reasoning intensity. The authors first analyze layer-wise reasoning behavior with sparse autoencoders and observe that reasoning-related activations become much stronger in later layers, then argue that this pattern is reflected by a corresponding increase in hidden-state $\ell_2$ magnitude. Building on this observation, they develop three training-free test-time methods guided by $\ell_2$ norms: adaptive layer-wise reasoning recursion, endogenous reasoning state steering, and $\ell_2$-guided response selection, all designed to amplify or better utilize internal reasoning activity during inference. On the theory side, the paper provides a formal connection showing that hidden-state $\ell_2$ norm bounds the activation strength of SAE reasoning features under several assumptions. Empirically, the authors support this claim with correlation and causal intervention analyses, and show that the proposed $\ell_2$-based inference strategies consistently improve reasoning performance across multiple models and benchmarks.

**Compliance With Llm Reviewing Policy:**

Affirmed.

**Final Justification:**

Most of my concerns have been addressed. Since my original score was already relatively high, and the concerns raised by the other reviewers are also reasonable, I have decided to keep my score unchanged.

**Key Questions For Authors:**

Please refer to the numbered items in the Weaknesses section above.

**Limitations:**

The authors do not appear to discuss the limitations of the proposed approach. For my suggestions, please see the Weaknesses section above.

**Strengths And Weaknesses:**

**Strengths:**
1. This paper studies an interesting and timely question: whether there exists a simple, model-intrinsic signal that tracks reasoning activity in LLMs, and it provides a fairly clear answer centered on the hidden-state $\ell_2$ norm.
2. The proposed methods are practically appealing because they are training-free and data-free, and the reported experiments suggest that all three variants can produce consistent gains across multiple models and reasoning benchmarks.
3. This paper is well written and easy to follow.

**Weaknesses:**

*If the authors could properly address them during the rebuttal phase, I am willing to raise my score.*
1. This paper is somewhat careless in notation. In several places, the authors use both $\ell_2$ and $l_2$ to denote the same norm. Although this is a minor presentation issue, inconsistency in mathematical notation hurts the paper’s professionalism and should be corrected throughout the manuscript.
2. The theoretical results rely on several fairly strong assumptions, such as approximate orthonormality, faithful SAE reconstruction, semantic orthogonality, and feature disentanglement. While these assumptions are clearly stated, it remains unclear to what extent they hold in practice for modern LLM hidden spaces, so the scope of the theory may be narrower than it first appears.
3. A substantial part of the paper’s motivation and validation depends on SAEs trained on model hidden states, yet SAEs themselves are known to be design-sensitive. The authors may want to provide a deeper discussion of how robust the conclusions are to SAE quality, sparsity choices, feature selection strategy, or alternative interpretability probes.
4. The practical methods are intuitive, but some of them still look somewhat heuristic. For example, peak detection uses an IQR-based threshold, ALRR repeats computation at detected layers, and LRS uses a summed $\ell_2$ score over middle-to-late layers. It would strengthen the paper to more carefully analyze sensitivity to these design choices and clarify which parts are essential.

---

> ### Author Rebuttal · Authors · 2026-03-31
>
> We sincerely thank the reviewer for the insightful feedback.
>
> Rebuttal Tables and Figures: https://anonymous.4open.science/r/ICML2026-Rebuttal-6673/README.md
> # W1: Notation Consistency
> We will unify notation to $\ell_2$.
> # W2: Validity of Theoretical Assumptions
> We validate on Qwen3-4B and Llama3-8B (averaged across the last 1/4 layers).
> 1. **B.1 Decoder Orthonormality** SAE decoder bases are near-orthonormal: diagonal ≈ 1.0000 (std $7.6×10^{-7}$), off-diagonal correlation $\epsilon_{orth}=0.0173\ll1$, consistent with [1].
> 2. **B.2 Reconstruction Quality** We evaluate reconstruction quality on both magnitude and direction. The high directional fidelity indicates that SAE preserves geometry, sufficient for our angle-based analysis.
> - Reconstruction error: 8.7%
> - Cosine similarity of original and reconstructed representations: $cos(h_{think},\hat h_{think})=0.9932$, $cos(h_{nothink},\hat h_{nothink})=0.9831$
> 3. **B.3 & B.4 Semantic Orthogonality** For base semantics $h_{base}$ and reasoning signals $h_{reasoning}$:
> - $cos(h_{base},h_{reasoning}) = \mathbf{0.0570}$ indicates near-orthogonality and strong disentanglement between semantic and reasoning.
> - $|h_{base}|\approx 186.1,|h_{reasoning}|\approx 116.9$. Comparable norms prove that reasoning is a substantial, independent component rather than negligible noise.
> - Layer-wise transition: cosine drops from 0.3144(Layer 5) to 0.057(late), indicating progressive disentanglement, consistent with the dynamics in Fig. 2.
> 4. **B.5 Feature Disentanglement** We assess the reconstruction error of the reasoning direction using reasoning feature subsets.
> - Using only the identified $k=668$ reasoning features, we faithfully reconstruct the reasoning direction: $cos(\sum_{i\in I_{reasoning}}\Delta z_i\cdot d_i,h_{reasoning}) = 0.9499$. Even in extreme sparsity (Top-5 features), the directional recovery remains high (0.8747).
> - Base Feature Independence: Base features show near-zero correlation with the reasoning direction: $cos(\sum_{i\in I_{base}}\Delta z_i\cdot d_i,h_{reasoning}) = 0.1443$
> This confirms that the reasoning signal is highly concentrated within $\mathcal{I}_{reasoning}$ and the separation of reasoning and base semantic signals is highly reliable.
> 5. **Conclusion** Direction separates reasoning from base semantics and magnitude reflects its strength. With semantic–reasoning disentanglement, $\ell_2$ increase directly measures reasoning energy. Empirical results above confirm our geometric assumptions hold approximately but sufficiently.
> # W3: Robustness to SAE Design and Alternative Probes
> 1. **SAE Configurations** We systematically vary key SAE designs on Qwen3-4B:
> - Architecture: Reduce feature dimensions from 4 to 2
> - Training Budget: Reduce epochs from 6 to 3
> - Sparsity: Increase the $L_1$ penalty ($\lambda_{L1}$) from $1e-3$ to $5e-3$
> - Selection Strategy: Use a relative ratio: $(think - nonthink) / (nonthink + \epsilon)$ instead of raw difference
> - Feature Dilution: Use only Top-40 features
>
> **Results**: $\ell_2$ norm increase in later layers remains invariant across all settings (**Link Rebuttal Fig. 2**), proving our findings are not tied to specific hyperparameters.
>
> 2. **Alternative Probes: MI/IG** We analyze mutual information (MI) and information gain (IG) patterns along the reasoning process, following [10] which identifies key thinking tokens via information theories. MI and IG are computed for tokens and cannot be visualized layer-wise.
>
> **Results**: $\ell_2$ peaks align with MI and IG spikes at key reasoning steps (**Link Rebuttal Fig. 3-4**). Some norm peaks align with negative IG, indicating exploratory and trial-and-error reasoning. Compared to MI (costly, requiring matrix inversions, GT-dependent), $\ell_2$ is near zero-cost and broadly applicable.
> # W4: Sensitivity of Application Designs
> We aim not to introduce highly-tuned methods, but to show that $\ell_2$ is broadly useful and robust under simple designs. We conduct extensive sensitivity analyses following the reviewer's suggestions. **Results: Rebuttal Table 5–6.**
> 1. **IQR threshold** For ALRR and ERSS: Dynamic (IQR-based) thresholds are essential; fixed thresholds (mean $\ell_2$ norm of MMLU-Pro) fail on hard tasks like AIME. Results are stable for reasonable $\tau$ (1.5, 3, 6), but too small $\tau$ triggers excessive, unnecessary interventions that degrade reasoning coherence.
> 2. **ALRR Loop Iterations** A single loop already yields significant gains. Excessive loops (Loop-8) degrade performance, indicating lightweight refinement is enough and the benefit does not rely on repeated computation.
> 3. **LRS Layer Aggregation Strategy** Aggregation rule (sum vs. peak count) has minor impact, but layer selection is critical: middle-to-late layers perform best, consistent with stronger late-layer reasoning signals.
> # References
> [1]Toy models of superposition
>
> [2]Demystifying Reasoning Dynamics with Mutual Information: Thinking Tokens are Information Peaks in LLM Reasoning NeurIPS 2025

---

> > ### Author Rebuttal · Reviewer_ojhr · 2026-04-01
> >
> > Thank you for the rebuttal. Most of my concerns have been addressed. Since my original score was already relatively high, and the concerns raised by the other reviewers are also reasonable, I have decided to keep my score unchanged.
> >
> > I encourage the other reviewers to read and respond to the authors' rebuttal, and I look forward to seeing the paper further improved.

---

> > > ### Author Response · Authors · 2026-04-03
> > >
> > > We sincerely thank you for confirming that our rebuttal has fully resolved your concerns and for **your encouraging words** regarding the other reviewers. We **deeply appreciate your time and the high standard** you have set for this work.
> > >
> > > Taking your valuable feedback to heart, we would like to clearly summarize how we have addressed your core concerns:
> > >
> > > 1. **On Theoretical Assumptions & SAE Robustness (W2 & W3)**: We acknowledge that geometric assumptions and the reliance on SAEs are strong. In our rebuttal, we provided systematic empirical diagnostics confirming that properties like orthonormality and semantic disentanglement hold approximately but sufficiently in practice. Furthermore, we demonstrated that our findings remain invariant across diverse SAE configurations and align with alternative information-theoretic probes (MI/IG). We believe this work serves as a foundational step in **bridging latent geometry and reasoning dynamics**. By establishing this formal link under specific conditions, we hope to inspire future work that relaxes these assumptions or explores alternative interpretability probes beyond SAEs.
> > >
> > > 2. **On Heuristic Application Designs (W4)**: We agree that practical interventions like IQR thresholds and ALRR loops are heuristic. To address this, we conducted extensive sensitivity analyses across thresholds, iteration counts, and layer aggregation rules, showing that our methods yield stable gains without fragile hyperparameter tuning. Our primary goal was to demonstrate that a **training-free, data-free** signal can effectively control reasoning even with such simple designs. We view these implementations as a proof of concept; future works could certainly benefit from more adaptive, learned strategies built upon this reliable baseline.
> > >
> > > **We are encouraged by the productive ongoing discussion** with all reviewers. We are grateful that other reviewers have acknowledged our efforts in addressing their concerns: we provided evidence clarifying the reasoning-intensity spectrum and architectural baselines; our experiments show that late-layer amplification exceeds depth-dependent trends and that theoretical assumptions are empirically supported; and we conducted additional analyses on extended model families and method combinations, further strengthening the manuscript's robustness and scope.
> > >
> > > We **truly cherish** this opportunity to exchange ideas with a reviewer of your expertise. Your insights have **significantly sharpened our arguments**, and we remain fully committed to refining the manuscript to the highest standards.
> > >
> > > We **fully respect your decision** given the broader discussion context. However, if the collective rebuttal and acknowledgement has addressed the reasonable concerns you noted and strengthened the manuscript's contribution, we would be **deeply honored by your reconsideration and support, should you find it appropriate**.
> > >
> > > Thank you again for your precious guidance.

---

### Official Review · Reviewer_j9yQ · 2026-03-12

**Soundness:** 3
**Presentation:** 3
**Significance:** 3
**Originality:** 2
**Overall Recommendation:** 4
**Confidence:** 5

**Summary:**

The authors suggest that reasoning intensity in large language models can be observed through the l2 norm of hidden states. Using SAE analysis, they show that reasoning-related features are associated with increases in hidden state norms. Based on this observation, they propose several inference-time applications using l2 norms: steering and response selection (ensemble). Experiments indicate that utilizing the l2 norm signal can improve model performance.

**Compliance With Llm Reviewing Policy:**

Affirmed.

**Final Justification:**

The authors have addressed my concerns, particularly on the utility of the L2 norm signal. I will raise my score accordingly.

**Key Questions For Authors:**

- What is “intensive thinking” or “thinking” in LRMs? These definitions are necessary to compare thinking and non-thinking states. (see W1)
- Could l2 norm values be influenced by special tokens such as <think> and </think>? Are the ℓ2 norm differences truly caused by thinking behavior? (see W1 and W2)
- How does Figure 2 change when the model does not think? (see W2)
- Is the l2 norm signal meaningful compared to other signals for applications (steering and selection)? (see W3)

**Limitations:**

The paper does not explicitly discuss limitations, but it seems there are no societal risks.
Suggestions regarding limitations are explained in the weakness and question.

**Strengths And Weaknesses:**

**Strengths**
- The authors identify that the l2 norm of hidden states correlates with the reasoning behavior of LRMs. This signal is simple to apply.
- The authors conduct extensive experiments on various models.
- The proposed methods are training-free and do not require additional computational resources to enhance model performance.

**Weaknesses**
- What is "thinking" of LRMs? Nothinking mode actually generates reasoning chains, although the length may be relatively short. It is unclear whether reasoning only counts as "thinking" when it appears inside <think> </think>. Because of this ambiguity, the paper should examine whether "thinking" actually corresponds to higher norm values with rigorous comparisons.
- More evidence is needed to establish that the observed l2 norm increase is specific to reasoning, rather than reflecting more general factors such as generation difficulty, activation scaling, or token uncertainty. It would also be helpful to examine whether the trend in Figure 2 changes in clearly non-reasoning settings.
- The authors need comparisons of applications to validate that the l2 norm signal is meaningful and strong. For example, in the case of LRS, the authors should compare this method with other methods such as DeepConf [1], which is a strong ensemble method and similar to the selection approach. Steering should also be compared with signals such as entropy, SAE signals, or log probability.

[1] Deep Think with Confidence, 2025

---

> ### Author Rebuttal · Authors · 2026-03-31
>
> We sincerely thank the reviewer for the insightful feedback.
>
> Rebuttal Tables and Figures: https://anonymous.4open.science/r/ICML2026-Rebuttal-6673/README.md
> # W1: Clarification on Thinking
> We distinguish thinking (process) from thinking mode (configuration).
> 1. **Definition** Following literature [1,2], we define "thinking/reasoning" as the logical transition from premises to conclusions. Thus, any response generation constitutes a thinking process, regardless of special tokens.
> 2. **Mode vs process** Thinking/non-thinking modes only refer to Qwen3 configurations trained via RL: the former encourages deeper, reflective exploration, while the latter favors more direct answers. Our comparisons highlights their **differences in thinking intensity, not the presence or absence of it.**
> 3. **Holistic Analysis.** We analyze the full response (including content outside <think> tags). Fig. 3 and Rebuttal Fig. 1 show both modes exhibit varying  norms over generation and layer-wise, but thinking mode has consistently higher $\ell_2$. Fig. 9 shows norms scale with task difficulty (e.g., AIME > GSM8K), supporting our claim.
> # W2: Specificity of $\ell_2$ Norm
> 1. **“Reasoning-specific” clarification.** We do not claim a one-to-one, exclusive mapping between $\ell_2$ norm and reasoning, but a strong, systematic association. Grounded in mechanistic interpretability [3,4], we propose $\ell_2$ norm as an efficient proxy for reasoning activity to replace training SAEs.
> 2. **Relation to other factors.** Signals like entropy, perplexity, and uncertainty are related and actually aligned with our findings. Prior work shows higher uncertainty induces more CoT reasoning[5,6]. We also observe correlation between $\ell_2$ and entropy (Table 1), suggesting a shared mechanism: increased compute allocation to difficult or uncertain steps.
> 3. **Activation scaling.** While layer scaling raises overall $\ell_2$, its magnitude and variance are much smaller on simple tasks. After removing this global trend, reasoning-related $\ell_2$ increases remain clear and systematic (hard reasoning benchmarks exhibit higher norms than simple tasks in Fig. 9, which is also discussed in Appendix J, rebuttal Fig. 1 shows higher norm trend for thinking).
> 4. **Cross-domain evidence.** Similar phenomenon appears in CV: feature norms correlate with confidence and salient signals [7], suggesting a general principle: models amplify representations for decision-critical features. In LLMs, this aligns with key reasoning steps.
> # W3: Comparison with other methods
> 1. **LRS vs DeepConf.** We compare LRS with DeepConf (Filter, Weighted) on 8 rollouts.
>
> **Results**: **Rebuttal Table 1 & Table 2**. LRS consistently achieves stronger, more stable gains, mostly outperforming DeepConf. DeepConf is competitive on easier tasks (e.g., BBH) but unstable on harder ones (MMLU-Pro, GPQA), likely due to uniformly low confidence that impedes accurate selection, while exploration (captured by higher norms) is more beneficial.
>
> 2. **Steering.** We compare TTTS with alternative signals:
> - Entropy: high entropy tokens as reasoning indicators
> - LogProb: low-confidence tokens as reasoning triggers
> - SAE: steering using high-activation SAE features
>
> **Results**: **Rebuttal Table 3 & Table 4**. TTTS achieves consistent gains overall, especially on reasoning-heavy tasks. Entropy/log-prob provide weak guidance. SAEs are competitive, but require substantial training, while L2 norm is cost-free.
> # Direct answers to questions
> **Q1**: Thinking is the logical transition from premises to conclusions (W1). Intensive thinking refers to RL-induced behaviors such as exploration, backtracking, and self-correction[2].
>
> **Q2**: $\ell_2$ is not affected by special tokens. We analyze full responses, and high-$\ell_2$ tokens extend beyond <think> tags (e.g., Fig. 3, 15, 25). Yes, the ℓ2 norm differences are directly driven by thinking behavior, as discussed above, including the relationship between norm and other relevant factors (W1 & W2).
>
> **Q3**: Rebuttal Fig. 1 shows the layer-wise norms for Non-thinking mode, which exhibits an upward trend but with lower magnitude compared to Thinking.
>
> **Q4**: $\ell_2$ is meaningful vs. other signals. Experiments (W3) show $\ell_2$-based methods are strong, robust, and efficient compared to alternatives.
> # References
> [1]Thinking Machines: A Survey of LLM based Reasoning Strategies
>
> [2]Towards Reasoning Era: A Survey of Long Chain-of-Thought for Reasoning Large Language Models
>
> [3]I Have Covered All the Bases Here: Interpreting Reasoning Features in Large Language Models via Sparse Autoencoders
>
> [4]How does Chain of Thought Think? Mechanistic Interpretability of Chain-of-Thought Reasoning with Sparse Autoencoding
>
> [5]Stepwise Perplexity-Guided Refinement for Efficient Chain-of-Thought Reasoning in Large Language Models ACL 2025 findings
>
> [6]CER: Confidence Enhanced Reasoning in LLMs ACL 2025
>
> [7]Understanding the Feature Norm for Out-of-Distribution Detection ICCV 2023

---

> > ### Author Rebuttal · Reviewer_j9yQ · 2026-04-04
> >
> > Thank you for your response.
> >
> > The authors have addressed my concerns, particularly on the utility of the L2 norm signal.
> > I will raise my score accordingly.

---

> > > ### Author Response · Authors · 2026-04-04
> > >
> > > **Thank you so much** for your thoughtful review and for taking the time to reconsider our responses. We are **truly grateful** that our clarifications regarding the L2 norm signal and its connection to reasoning behavior have addressed your concerns.
> > >
> > > Your constructive feedback has **significantly helped us improve the clarity and rigor of our work**. We **sincerely appreciate** your support and the opportunity to refine our manuscript through this discussion.

---

### Decision · Program_Chairs · 2026-04-30

**Decision:**

Accept (spotlight)

**Comment:**

This paper finds that the L2 norm of model representations across layers correlates with reasoning intensity in LLMs, and uses this technique to propose test-time methods that improve performance. The experiments are thorough, including a range of benchmarks and models, and the reviewers generally found their concerns were addressed by the rebuttal. I think this paper offers some intriguing observations that merit further exploration.